# A transposon insertion in the promoter of *OsUBC12* enhances cold tolerance during *japonica* rice germination

Chuanzhong Zhang ®[1,2], Hongru Wang ®[3], Xiaojie Tian[1], Xinyan Lin[1,4], Yunfei Han[1], Zhongmin Han[1], Hanjing Sha[1], Jia Liu[1], Jianfeng Liu[4], Jian Zhang ®[5], Qingyun Bu ®[1] & Jun Fang ®[1,6] ✉

Low-temperature germination (LTG) is an important agronomic trait for rice (*Oryza sativa*). *Japonica* rice generally has greater capacity for germination at low temperatures than the *indica* subpopulation. However, the genetic basis and molecular mechanisms underlying this complex trait are poorly understood. Here, we report that *OsUBC12*, encoding an E2 ubiquitin-conjugating enzyme, increases low-temperature germinability in *japonica*, owing to a transposon insertion in its promoter enhancing its expression. Natural variation analysis reveals that transposon insertion in the *OsUBC12* promoter mainly occurs in the *japonica* lineage. The variation detected in eight representative two-line male sterile lines suggests the existence of this allele introgression by *indica-japonica* hybridization breeding, and varieties carrying the *japonica OsUBC12* locus (transposon insertion) have higher low-temperature germinability than varieties without the locus. Further molecular analysis shows that OsUBC12 negatively regulate ABA signaling. OsUBC12-regulated seed germination and ABA signaling mainly depend on a conserved active site required for ubiquitin-conjugating enzyme activity. Furthermore, OsUBC12 directly associates with rice SUCROSE NON-FERMENTING 1-RELATED PROTEIN KINASE 1.1 (OsSnRK1.1), promoting its degradation. OsSnRK1.1 inhibits LTG by enhancing ABA signaling and acts downstream of OsUBC12. These findings shed light on the underlying mechanisms of UBC12 regulating LTG and provide genetic reference points for improving LTG in *indica* rice.

Seed germination is a complex trait influenced by many genes and environmental conditions[1,2]. Optimal rice (*Oryza sativa*) germination temperature ranges from 25 °C to 35 °C; temperatures below 17 °C cause cold stress, with low germination rates, germination delay, retarded growth and seedling mortality[3,4]. Moreover, low-temperature germinability is a prerequisite for modern direct-seeding cultivation, an alternative to conventional transplanting that effectively reduces rice production costs[5,6]. Although rice is generally sensitive to low

[1]Key Laboratory of Soybean Molecular Design Breeding, State Key Laboratory of Black Soils Conservation and Utilization, Northeast Institute of Geography and Agroecology, Chinese Academy of Sciences, 150081 Harbin, China. [2]Key Laboratory of Germplasm Enhancement, Physiology and Ecology of Food Crops in Cold Region, Ministry of Education, Northeast Agricultural University, Harbin 150030, China. [3]Shenzhen Branch, Guangdong Laboratory of Lingnan Modern Agriculture, Key Laboratory of Synthetic Biology, Ministry of Agriculture and Rural Affairs, Agricultural Genomics Institute at Shenzhen, Chinese Academy of Agricultural Sciences, Shenzhen, China. [4]Jilin Provincial Key Laboratory of Plant Resource Science and Green Production, Jilin Normal University, Siping, Jilin Province 136000, China. [5]State Key Lab of Rice Biology, China National Rice Research Institute, Hangzhou 311400, China. [6]Yazhouwan National Laboratory, Sanya 572024, China. ✉e-mail: fangjun@iga.ac.cn

temperatures, artificial human selection and cultivation in different geographic regions have given rise to two distinct rice varietal groups with significantly different chilling tolerances[7,8]. *Indica* subspecies are more sensitive to cold stress than *japonica* subspecies[9–11], and most cold-tolerant alleles identified in previous studies belong to *japonica*[12–14].

Ubiquitin (Ub) is a highly conserved 76-amino-acid polypeptide that can be conjugated to target proteins by the sequential action of three classes of enzymes: E1 ubiquitin-activating enzymes, E2 ubiquitin-conjugating enzymes and E3 ubiquitin-ligase enzymes[15]. Ubiquitin harbors seven lysine (K) residues: K6, K11, K27, K29, K33, K48 and K63. Polyubiquitin (polyUb) chains are formed when multiple Ub moieties are linked by one of the seven lysine residues in a ubiquitin molecule, or by the N-terminal methionine residue in the form of head-tail linear repeats[16–19]. PolyUb chains exhibit different topologies and are associated with diverse biological functions[20]. K48-linked polyubiquitination, the most common type, triggers the degradation of target proteins by the 26 S proteasome[21]. Atypical K63-linked polyubiquitination alters the activity of target proteins, primarily in a proteolysis-independent manner[22]. Much less is known about the other polyUb chain linkages. Ubiquitin-conjugating (UBC) E2 enzymes play important roles in protein ubiquitination by mediating the formation of the polyubiquitin chain and transferring it to the target protein[23]. In general, E2s have a conserved UBC domain that harbors the active-site cysteine residue required for enzyme-ubiquitin thioester bond formation[24]. UBC enzymes are widespread in eukaryotes. Rice has 48 UBC members, although only 39 are predicted to contain cysteine active sites[25].

Studies of plant E2 function are lacking and mainly relate to Arabidopsis (*Arabidopsis thaliana*). Among the 37 UBCs in Arabidopsis, ubiquitin-conjugating enzyme activity has been detected for 17, including AtUBC1, AtUBC2 and AtUBC32[26,27]. Further functional analyses reveal that AtUBCs play diverse roles. For instance, the double loss-of-function Arabidopsis mutant *atubc1-1 atubc2-1* shows a dramatically reduced number of rosette leaves and an early-flowering phenotype[28]. AtUBC32 regulates brassinosteroid (BR)-mediated salt tolerance[29], while AtUBC27 modulates ABA signaling and drought tolerance by promoting the degradation of the ABA co-receptor ABA INSENSITIVE 1 (ABI1)[30]. UBC13 is the only known E2 that can catalyze K63-linked polyubiquitination, which is closely related to iron metabolism[31], auxin signal transduction[32], and low-temperature and pathogen stress responses[33] in Arabidopsis. Overall, however, knowledge regarding E2s in plants is limited, with little in-depth analysis of the biological functions and regulatory mechanisms of E2s having been performed in rice and other crops.

The phytohormone abscisic acid (ABA) plays pivotal roles in seed germination. Accordingly, mutation or overexpression of genes involved in ABA biosynthesis or signaling often results in abnormal germination phenotypes[34–36]. For example, a loss-of-function mutation in the 14-3-3 family gene *GF14h* enhances ABA signaling and reduces seed germination rate in rice[37]. *Seed Dormancy 6* (*SD6*), encoding a basic helix-loop-helix (bHLH) transcription factor in rice, influences seed dormancy and germination by directly regulating the ABA catabolism gene *ABA8OX3*[38]. Moreover, we identified a series of regulatory factors controlling rice seed germination and dormancy by large-scale screening of mutant rice populations. Notably, the majority are related to ABA accumulation or signaling[39–45].

The core components of the ABA signaling network have been identified in recent decades. ABA is perceived by the ABA receptors PYRABACTIN RESISTANCE1 (PYR1)/PYR1-LIKE (PYL)/REGULATORY COMPONENTS OF ABA RECEPTORS (RCARs). The binding of ABA to these receptors promotes their interaction with the subclass A type 2C protein phosphatases (PP2Cs), which releases the SNF1-related protein kinase 2s (SnRK2s), members of the plant serine/threonine protein kinase family[46–48]. SnRK2s activate basic leucine zipper (bZIP) transcription factors, including ABSCISIC ACID INSENSITIVE 5 (ABI5), to regulate the expression of ABA-responsive genes[49,50]. Several lines of evidence also indicate a close interaction between SnRK1 and ABA signaling[51–55]. SnRK1.1 positively regulates ABA signaling in Arabidopsis; accordingly, its overexpression causes ABA hypersensitivity[56].

In this work, we examine *indica-japonica* rice chromosome segment substitution lines (CSSLs) and identify a transposon insertion in the *japonica OsUBC12* promoter that activates *OsUBC12* expression and increases cold tolerance during germination. Natural variation analysis reveals that the transposon in the *OsUBC12* promoter is found mainly in the *japonica* lineage and is absent in wild and *indica* accessions. And the variation detection analysis suggests the existence of this allele introgression by *indica-japonica* hybridization breeding, while varieties carrying the *japonica OsUBC12* locus (transposon insertion) have higher low-temperature germinability than varieties without the locus. Molecular and biochemical analysis show that OsUBC12 negatively regulate ABA signaling by promoting the proteasomal degradation of OsSnRK1.1, thus accelerating LTG. These results broaden our understanding of the regulatory functions of rice E2 UBC enzymes involved in ABA signaling and LTG, and provide a potential genetic locus that may be applied to improve the LTG of *indica* rice.

## Results

### A transposon insertion in the *OsUBC12* promoter promotes seed germination at low temperature

To identify genes regulating low-temperature germination, an advanced mapping population of CSSLs[57] derived from a cross between IR64 (*Oryza sativa* ssp. *indica*, cold-susceptible)[58] and Koshihikari (*Oryza sativa* ssp. *japonica*, cold-tolerant)[59] was generated and subjected to germination tests at two different immersion temperatures: 15 °C for 60 h and 30 °C for 24 h. Among the 36 CSSLs with Koshihikari introgression in an IR64 background, three (SL2116, SL2117 and SL2130) attracted our attention, as they consistently exhibited the highest increases in germination rates at both 15 °C and 30 °C when compared with IR64 (Supplementary Fig. 1a, b).

The introgression segments on chromosome 9 of Koshihikari in SL2130 contained a previously identified bZIP transcription factor-coding gene, *bZIP73* (*LOC_Os09g29820*). It has been reported that the *japonica* allele of bZIP73 (bZIP73$^{Jap}$) could improve rice cold stress tolerance[8]. Therefore, the more vigorous LTG of SL2130 may be partially attributable to the introgression of *bZIP73$^{jap}$*. Interestingly, the Koshihikari introgression segments of SL2116 and SL2117 include overlapping regions of Chr. 5 (Supplementary Fig. 2a, b), implying that these introgression segments, and especially the overlapping section, might control low-temperature germination.

To determine which genes contribute to the LTG trait, we carried out fine-mapping for a population obtained by crossing SL2117 and IR64 (Supplementary Fig. 2c). Genetic analysis showed that the LTG phenotype of SL2117 segregated as a semi-dominant trait (Supplementary Table 1). We generated a segregating F$_2$ population to screen for fixed homozygous recombinant individuals using twelve molecular markers covering the target region (specific primers are listed in Supplementary Data 1). Finally, progeny phenotyping of fixed homozygous recombinant individuals narrowed the candidate gene to a ~40-kb region between markers M5-22.59 and M5-22.63, containing five predicted open reading frames (ORFs) (Fig. 1a, upper panel). We found that the locus *LOC_Os05g38550*, encoding an E2 ubiquitin-conjugating enzyme OsUBC12, is localized to this region, and that although only a T/C synonymous mutation without changing the amino acid residue was detected in its ORFs between Koshihikari and IR64 (Supplementary Fig. 3), the −542 site of its promoter was inserted a transposon gene (LOC_Os05g38540; about 6 kb) in *japonica* Koshihikari background compared to that in *indica* IR64 (Fig. 1a, bottom panel) by referencing the genome sequence of Nipponbare (from Rice Genome

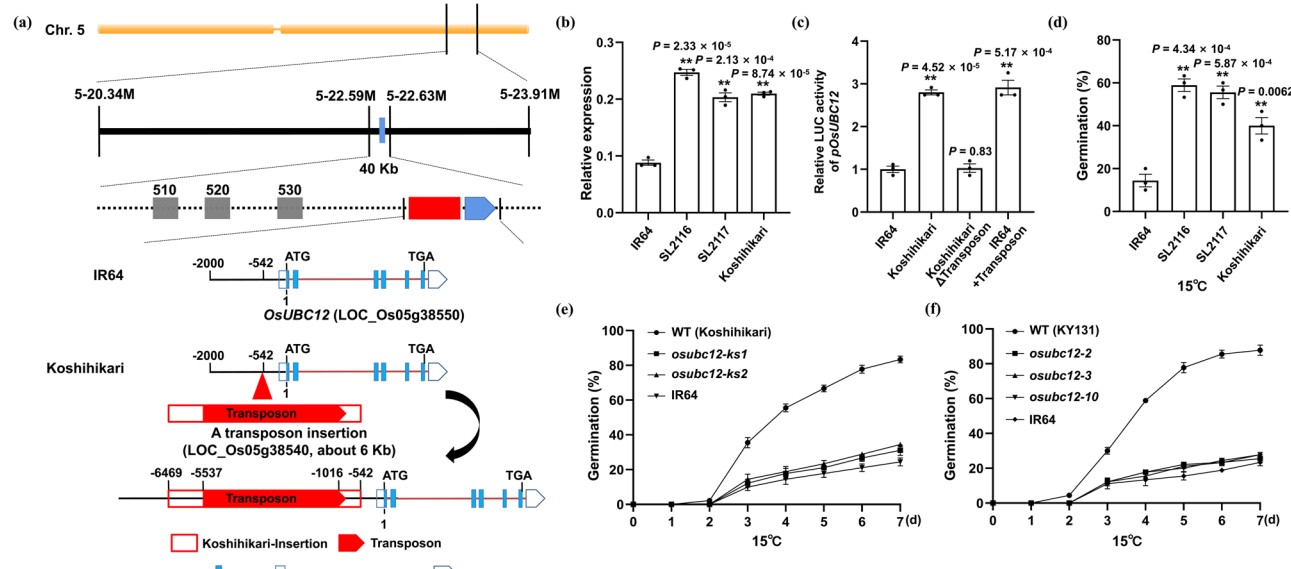

**Fig. 1 | A transposon insertion in the *OsUBC12* promoter promotes seed germination at low temperatures. a** Map-based cloning of *OsUBC12* and the transposon insertion in the *OsUBC12^Koshihikari^* promoter as compared to IR64. Specific primers used for mapping are listed in Supplementary Data 1. **b** Relative expression of *OsUBC12* in SL2116 and SL2117 seeds. Values are means ± SE from three individual replicates (*n* = 3). The housekeeping gene *OsUBQ5* was used as an internal control to normalize the data. The data were statistically analyzed using two-tailed Student's *t*-test (**P* < 0.05, ***P* < 0.01). **c** Relative LUC activity in rice protoplasts transformed with the recombinant *pOsUBC12^IR64^*, *pOsUBC12^Koshihikari^* *pOsUBC12^KoshihikariΔTransposon^*, or *pOsUBC12^IR64+Transposon^* vector. The relative LUC activity of *pOsUBC12^IR64^* was set to 1. Values are means ± SE from three individual replicates

(*n* = 3). The data were statistically analyzed by two-tailed Student's *t*-test (**P* < 0.05, ***P* < 0.01). **d** Germination rates of SL2116 and SL2117 after 60 h at 15 °C. Values are means ± SE from three individual biological replicates (30 seeds per biological replicate). The data were statistically analyzed using two-tailed Student's *t*-test (**P* < 0.05, ***P* < 0.01). **e** Time-course germination analysis of *osubc12-ks* mutants at 15 °C. The WT (Koshihikari) and IR64 were used as controls. Values are means ± SE from three individual biological replicates (30 seeds per biological replicate). **f** Time-course germination analysis of *osubc12* mutants at 15 °C. The WT (KY131) and IR64 were used as controls. Values are means ± SE from three individual biological replicates (30 seeds per biological replicate). Source data are provided as a Source Data file.

Annotation Project Database, http://rice.uga.edu/) and combining with pan-genome data of 33 genetically diverse rice accessions (including Koshihikari and IR64, from Rice Resource Center Database, http://ricerc.sicau.edu.cn/)[60]. Of the remaining three genes, *LOC_Os05g38510* and *LOC_Os05g38530* have 98.19% and 99.58%, genomic homologies between Koshihikari and IR64, respectively, with nearly no much variation (Supplementary Figs. 4 and 5), whereas the variation in *LOC_Os05g38520* was all in the introns (Supplementary Fig. 6). We therefore hypothesized that the transposon insertion in *pOsUBC12^Koshihikari^* may affect the expression level of *OsUBC12*, and thus cause the change of germinability.

To test our hypothesis, we examined the transcription level of *OsUBC12* in the SL2116 and SL2117 lines. Compared to IR64, the transcription of *OsUBC12* was significantly higher in SL2116 and SL2117 (Fig. 1b). Moreover, besides the presence or absence of the transposon, we also analyzed other polymorphisms in the promoter region of *OsUBC12* (Supplementary Data 2), and performed a dual-luciferase reporter assay. We cloned *pOsUBC12^IR64^* (2 kb) or *pOsUBC12^Koshihikari^* (7.9 kb, containing the inserted transposon) to drive firefly luciferase, and used Renilla luciferase as an internal reference for transfection efficiency (Supplementary Fig. 7). We also constructed *pOsUBC12^KoshihikariΔTransposon^* (*pOsUBC12^Koshihikari^* but lacking the transposon, the transposon was artificially removed) and *pOsUBC12^IR64+Transposon^* (*pOsUBC12^IR64^* but containing the transposon, the transposon was artificially inserted into the same position as the *pOsUBC12^Koshihikari^*) together to directly further elucidate whether the transposon insertion changes the transcriptional activity of the *OsUBC12* promoter rather than other polymorphisms (Supplementary Fig. 7). Compared with the rice protoplasts transfected with *p35S:REN–pOsUBC12^IR64^:LUC*, those transfected with *p35S:REN–pOsUBC12^Koshihikari^:LUC* and *pOsUBC12^IR64+Transposon^* displayed significantly increased promoter activity (greater LUC/REN

ratios), and the promoter activity of *pOsUBC12^KoshihikariΔTransposon^* has no significant difference only because of the loss of the transposon (Fig. 1c). These results confirm that the insertion of the transposon in *pOsUBC12^Koshihikari^* could up-regulate the expression of *OsUBC12*, while other polymorphisms in the promoter region of *OsUBC12* do not directly affect the expression. The Germination tests of SL2116 and SL2117 lines at 15 °C and 30 °C (Fig. 1d and Supplementary Fig. 8a) also showed that they had higher germination rates than the IR64 control, especially at low temperature (15 °C) (Fig. 1d). To determine the relationship between *OsUBC12* and cold tolerance during germination, and whether *OsUBC12* influences the germination, we generated *OsUBC12* homozygous knockout mutants (*osubc12-ks1* and *osubc12-ks2*) using a clustered regularly interspaced short palindromic repeat (CRISPR)/CRISPR-associated nuclease 9 (Cas9) system in Koshihikari background (Supplementary Fig. 9). When we tested the germination of these knockout mutants at 15 °C and 30 °C, the *osubc12-ks* mutant lines showed clearly decreased germination rates at low temperature (15 °C) (Fig. 1e), but only a slight reduction in germination at 30 °C (Supplementary Fig. 8b), confirming that OsUBC12 preferentially accelerates seed germination at low temperature, but fine-tunes seed germination at 30 °C.

To further determine the role of OsUBC12 in low-temperature germination, we generated *OsUBC12* homozygous knockout mutants (*osubc12-2*, *osubc12-3*, and *osubc12-10*) using CRISPR/Cas9 system in the Kongyu 131 (KY131) background (a high-quality rice with a large planting area in our cold *Japonica* rice area in Northeast China, and its genotype is consistent with Koshihikari, both of which carry the *japonica OsUBC12* locus) (Supplementary Fig. 10). Compared with the WT (KY131), the germination of *osubc12* mutant lines also showed strongly delayed at low temperature (15 °C) (Fig. 1f), but only a slight reduction in germination at 30 °C (Supplementary Fig. 8c). These

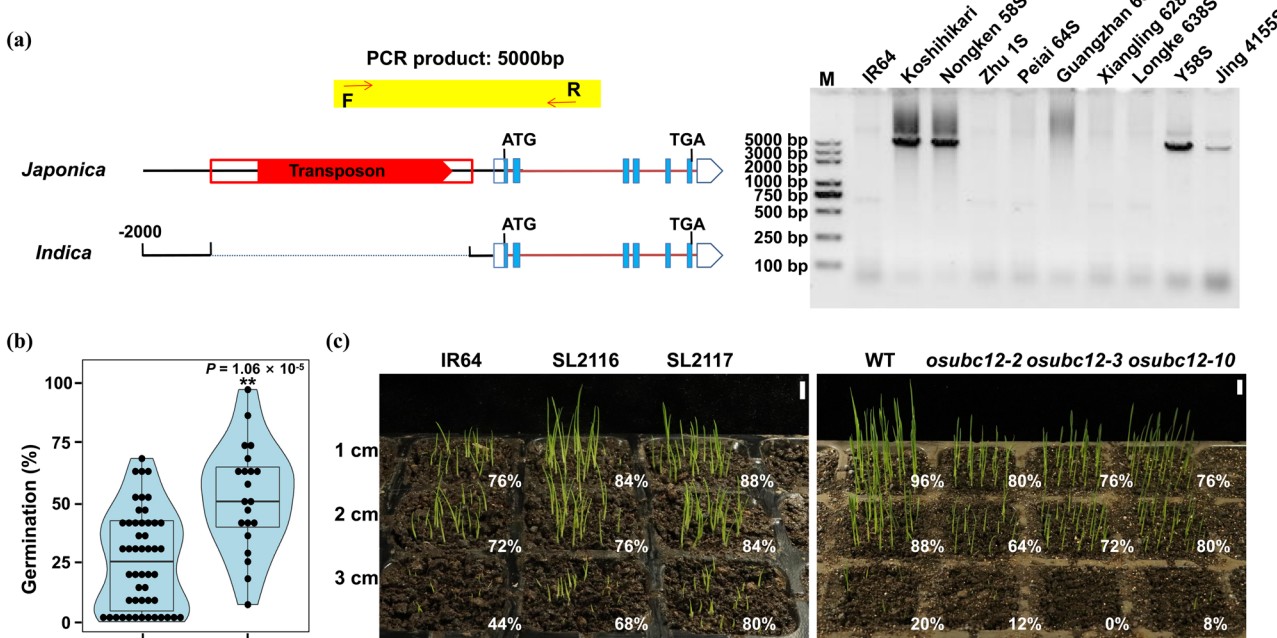

**Fig. 2 | Natural variations and potential application of *OsUBC12* in rice. a** The presence of the transposon insertion in eight representative two-line male sterile lines. M, DL2000 Plus DNA Marker. To detect the transposon insertion, we used forward (F) and reverse (R) primers located downstream of the transposon gene and upstream of *UBC12*, respectively; the PCR products are sequenced correctly. Specific primers for transposon insertion detection are listed in Supplementary Data 1. **b** Violin plots showing low-temperature germinability of 50 varieties without transposon insertion and 21 varieties with transposon insertion, which from the majority of the 69 cultivated accessions selected from nine rice subpopulations of the 3000 rice genomes as well as all varieties and sterile lines in Fig. 2a. $n = 50$ (without transposon insertion), 21 (with transposon insertion). Edges of box indicate 25 and 75 percentile points along with medians. Whiskers indicate minima and maxima. The data were statistically analyzed by two-tailed Student's *t*-test (*$P < 0.05$, **$P < 0.01$). **c** Left panel: seedlings of IR64 and the introgression lines SL2116 or SL2117, after 8 d at 1, 2 or 3 cm sowing depth under direct-seeding conditions at low temperature (18 °C). Right panel: seedlings of WT (KY131) and *osubc12* mutants, after 7 d at 1, 2 or 3 cm sowing depth under direct-seeding conditions at low temperature (18 °C). The value shown in the lower-right corner of each hole is the emergence rate. Scale bars, 1 cm. Source data are provided as a Source Data file.

results suggest that the transposon insertion in the *OsUBC12* promoter might promote low-temperature germination.

**Natural variations and potential application of *OsUBC12* in rice**

To study the natural variation in *pOsUBC12* among rice populations, we investigated a rice diversity panel consisting of 69 cultivated accessions selected from nine rice subpopulations of the 3000 rice genomes and six wild accessions[61]. We determined the presence/absence state for all the accessions and found that the transposon is absent in all wild and *indica* accessions in the panel, but show high frequency in *japonica* accessions (21/24) (Supplementary Table 2). The fixation index ($F_{st}$) among these subpopulations was 0.976, suggestive of significant genetic divergence among them (Supplementary Table 2). Furthermore, we extended our analyses using the high quality rice genome assembly dataset published in ref. 62. We called the presence absence variation of the transposon in 197 Asian domesticated rice accessions with subgroup information. The result is consistent with the result obtained with the dataset from ref. 61: the transposon is enriched in *japonica* rice (28/58), and absent in aus group, and only found in one accession of 135 accessions of *indica* variety (Supplementary Table 3). These results suggest that the transposon insertion occurred in the *japonica* lineage (Supplementary Tables 2 and 3).

Hybridization between different rice groups has been a common practice in rice breeding, and genomic locis conferring favorable agronomic traits are often introgressed across different rice subgroups[63-65]. Several recent studies have identified extensive introgressions between modern GJ (Geng/*japonica*) and XI (Xian/*indica*) rice varieties during modern rice breeding[66,67]. Since two-line hybrid rice has been used for large-scale grain production in China, benefiting from the continuous improvement of two-line male sterile lines, we checked for the transposon insertion in eight representative two-line male sterile lines, along with IR64 and Koshihikari as negative and positive controls. Nongken 58 S is the first two-line male sterile line developed from *japonica* rice; Peiai 64 S is the first commercialized *indica* two-line male sterile line, which was transferred from Nongken 58 S and then transferred to the second generation of the main *indica* two-line male sterile line Guangzhan 63 S; Zhu 1 S is a two-line male sterile line from *indica* rice, which was transformed into Xiangling 628 S and Longke 638 S and finally developed into the modern elite *indica* two-line male sterile lines Y58S and Jing 4155 S[68,69]. The results showed that the transposon insertion normally exits in two *japonica* lines, Koshihikari and Nongken 58 S (Fig. 2a). Interestingly, the *japonica OsUBC12* locus (transposon insertion) has been introgressed into Y58S and Jing 4155 S, from Nongken 58 S, Zhu 1 S, Peiai 64 S, Guangzhan 63 S, Xiangling 628 S, Longke 638 S (Fig. 2a), indicating that this *japonica*-derived locus (harboring the transposon insertion) has been introgressed into modern cultivars of an *indica* genomic background. To analyze the relationship between the *japonica OsUBC12* locus (transposon insertion) and LTG, we investigated the low-temperature germinability of 50 varieties without transposon insertion and 21 varieties with transposon insertion, which from the majority of the 69 cultivated accessions selected from nine rice subpopulations of the 3000 rice genomes as well as all varieties and sterile lines in Fig. 2a, confirming that varieties carrying the *japonica OsUBC12* locus (transposon insertion) have higher low-temperature germinability than varieties without the locus (Fig. 2b). Meanwhile, we divided *japonica* into two groups (transposons + vs −) for LTG

comparison. The results showed that the low-temperature germinability of varieties carrying the *japonica* *OsUBC12* locus in *japonica* was indeed higher than that of varieties without the locus (Supplementary Fig. 11a). We also tested the low-temperature germinability of SL2016[57], the CSSL with IR64 introgression (IR64 introgression segments containing the IR64 *OsUBC12* locus) in a Koshihikari background and whose *UBC12* transcript levels were significantly lower than those of the control Koshihikari (Supplementary Fig. 11b, c). Compared to Koshihikari, the low-temperature germinability was significantly lower in SL2016 (Supplementary Fig. 11d). Furthermore, a simulated rice direct seeding experiment at low temperature (18 °C) at soil depths of 1, 2 and 3 cm showed that the seedlings of introgression lines SL2116 and SL2117 emerged faster than those of IR64, while *osubc12* mutants emerged more slowly than WT (KY131) seedlings (Fig. 2c). These results suggest that *japonica* *OsUBC12* locus introgressions exists in *indica-japonica* hybridization breeding, and may be used to improve the low-temperature germinability of *indica* rice.

### OsUBC12 accelerates low-temperature germination by negatively regulating ABA signaling

The role of ABA in cold tolerance and seed germination has been well-documented by numerous genetic and physiological studies[70–75]. To explore the molecular mechanism by which OsUBC12 regulates LTG, we first analyzed the expression kinetics of OsUBC12 in response to cold and ABA. As shown in Fig. 3a, the transcription of *OsUBC12* was induced by both cold and ABA, implying that OsUBC12 may be involved in responding to ABA and cold stress. Then, we examined differences in gene expression of both WT and *osubc12* mutant seeds germinated at low temperature (15 °C) using transcriptome deep sequencing (RNA-seq). We identified 3526 differentially expressed genes (DEGs) (2063 up-regulated and 1463 down-regulated) in *osubc12* mutants compared to the WT using a 1.2-fold change in expression and $p < 0.05$ as the threshold (Supplementary Fig. 12a and Supplementary Data 4). Gene ontology (GO) enrichment analysis revealed that OsUBC12 may be involved in a variety of biological and physiological processes, such as response to abiotic stimulus, response to temperature stimulus, response to cold, abscisic acid-activated signaling pathway, response to abscisic acid, protein binding ect (Supplementary Fig. 12b and Supplementary Data 5). Among the DEGs, *OsUBC12* was significantly down-regulated in *osubc12* mutants compared with WT (Supplementary Fig. 12c). Moreover, many ABA-related genes showed differential expression between WT and *osubc12* mutants (Supplementary Fig. 12c and Supplementary Data 6). Notably, the expression of *OsABI5*, a core components of ABA signaling which also functions as a negative regulator of seed germination[76], was up-regulated in *osubc12* mutants (Supplementary Fig. 12c and Supplementary Data 6). We then examined the expression of *OsABI5* as well as *OsRAB21* which is a commonly used marker gene for ABA response[77] in WT and *osubc12* mutants by qRT-PCR analysis. Both *OsABI5* and *OsRAB21* expression was significantly higher in *osubc12* mutants compared to WT (Fig. 3b, c).

We also studied the effect of ABA on the germination of WT and *osubc12* mutant seeds by quantifying germination rates in response to 0, 1 or 2 μM ABA (Fig. 3d–f). In the absence of ABA, the *osubc12* mutant lines exhibited delayed germination (Fig. 3d). Treatment with 1 or 2 μM exogenous ABA produced a more pronounced inhibitory effect on *osubc12* mutants than on WT (Fig. 3e, f). To better quantify the ABA sensitivity of WT and *osubc12* mutants, we analyzed their ABA-mediated germination inhibition rates. After 1 or 2 μM ABA treatment, the germination inhibition rate was significantly higher in *osubc12* mutants in WT, indicating that *OsUBC12* knockdown increases ABA sensitivity (Fig. 3g, h). Taken together, these findings indicate that knockdown of *OsUBC12* enhances ABA signaling.

### OsUBC12's role in regulating LTG mainly depends on its conservated ubiquitination function

The protein structure and sequence alignment of OsUBC12 showed that OsUBC12 contains a highly conserved UBC domain harboring an active-site cysteine residue (Supplementary Fig. 13a, b). In general, the active site cysteine residue is required for enzyme-ubiquitin thioester bond formation[24]. To determine whether OsUBC12 has ubiquitin-conjugating enzyme activity, we mutated its active-site cysteine residue to alanine (OsUBC12$^{C92A}$) and performed an in vitro ubiquitin thioester formation assay. In the presence of the E1 and Ub, OsUBC12-His generated abundant poly-Ub conjugates (Fig. 4a, lane 4), while the mutated OsUBC12$^{C92A}$-His variant did so very slightly (Fig. 4a, lane 5). These results indicate that OsUBC12 has ubiquitin-conjugating enzyme activity, which requires the conserved cysteine residue in its UBC domain.

To analyze the relationship between OsUBC12-mediated ubiquitination and its biological function, we produced transgenic rice expressing *p35S:OsUBC12* (*OsUBC12-OE*) or *p35S:OsUBC12$^{C92A}$* (*OsUBC12$^{C92A}$-OE*), using enhanced green fluorescent protein (EGFP) as a selectable marker for germination assays at 15 °C and 30 °C. Fluorescence screening (Supplementary Fig. 14a, b) and RT-qPCR (Fig. 4c) confirmed the overexpression of *OsUBC12* and *OsUBC12$^{C92A}$* in three independent T$_3$ *OsUBC12-OE* lines (*OsUBC12-OE1–3*) and T$_3$ *OsUBC12$^{C92A}$-OE* lines (*OsUBC12$^{C92A}$-OE1–3*), respectively. Compared with WT, *OsUBC12-OE* lines showed early germination at 15 °C. In contrast, the germination of *OsUBC12$^{C92A}$-OE* lines at 15 °C was more similar to that of WT (Fig. 4b). Moreover, *OsABI5* and *OsRAB21* expression was significantly lower in *OsUBC12-OE* lines compared to WT. However, the reduction was less pronounced in the *OsUBC12$^{C92A}$-OE* lines (Supplementary Fig. 15). Additionally, *OsUBC12-OE* lines exhibited early germination in the absence of ABA (Fig. 4d) and were less sensitive to ABA compared to WT. In contrast, this reduction in ABA sensitivity was less pronounced in the *OsUBC12$^{C92A}$-OE* lines (Fig. 4e–h). These results suggest that the conserved ubiquitination function of OsUBC12 is required to regulate LTG and ABA responses in rice.

### OsUBC12 interacts with OsSnRK1.1

To further understand the molecular regulatory mechanism of OsUBC12 protein, we attempted to identify its ubiquitination targets, using OsUBC12 as bait to screen a yeast two-hybrid (Y2H) library generated from rice seed cDNA, and isolated 67 candidate clones. Expressed sequence tags (ESTs) from partial candidate genes encoding proteins are listed in Supplementary Table 4. Among them, RING-BOX1 (OsRBX1), OsSnRK1.1, OsUBC12 and OsWRKY42 were verified in a point-to-point yeast two-hybrid system. Among the different combinations tested, only yeast cells co-expressing *pGBD-OsUBC12* and *pGAD-OsSnRK1.1* grew well on screening medium (QDO) and showed α-galactosidase activity (Fig. 5a), indicating that OsUBC12 may interact with OsSnRK1.1 in yeast cells.

We performed an in vitro pull-down assay to validate the interaction between OsUBC12 and OsSnRK1.1. OsSnRK1.1-His, OsUBC12-GST and GST alone were all detected in whole-cell lysates (Input). OsSnRK1.1-His was pulled down by OsUBC12-GST, but not by the GST-only control, suggesting that OsUBC12 directly interacts with OsSnRK1.1 (Fig. 5b). We further confirmed the OsUBC12–OsSnRK1.1 interaction *in planta* using firefly luciferase complementation imaging (LCI) (Fig. 5c). Co-IP assays indicated that OsSnRK1.1-Myc fusion proteins could be immunoprecipitated with OsUBC12-Flag in *N. benthamiana* transiently expressing OsSnRK1.1-Myc and OsUBC12-Flag, while OsSnRK1.1-Myc could not be immunoprecipitation with negative control GFP-Flag (Fig. 5d), also confirmed the OsUBC12-OsSnRK1.1 interaction *in planta*. Collectively, the four independent assays demonstrate that OsUBC12 directly interacts with OsSnRK1.1 both in vitro and *in planta*.

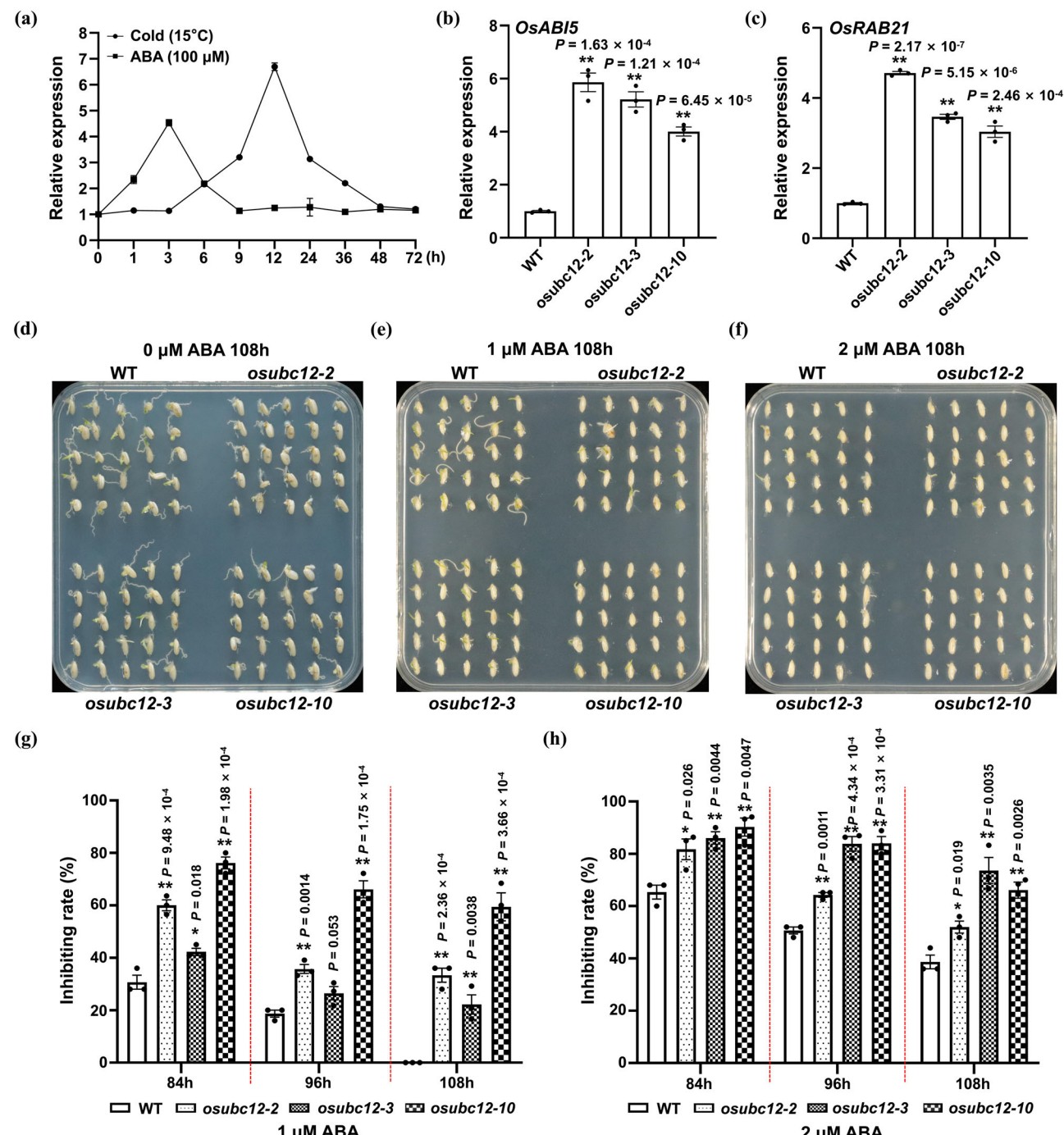

**Fig. 3 | OsUBC12 accelerates low-temperature germination by negatively regulating ABA signaling. a** Relative expression of *OsUBC12* in WT (KY131) seeds at cold (15 °C) and ABA (100 μM) treatment during germination. The relative expression levels of *OsUBC12* were compared with those of mock-treated seeds (seeds treated with sterile water) at the same time point. Values are means ± SE from three individual replicates ($n = 3$). The housekeeping gene *OsUBQ5* was used as an internal control to normalize the data. Relative expression of *OsABI5* (**b**) and *OsRAB21* (**c**) in seeds of *osubc12* mutants. The expression level of the control sample (WT, KY131) was set to 1. Values are means ± SE from three individual replicates ($n = 3$). The housekeeping gene *OsUBQ5* was used as an internal control to normalize the data. The data were statistically analyzed using two-tailed Student's *t*-test (*$P < 0.05$, **$P < 0.01$). Germination performance of WT (KY131) and *osubc12* mutant seeds after 108 h on ½-MS agar medium containing 0 μM (**d**), 1 μM (**e**) or 2 μM (**f**) ABA. The germination inhibition rate of WT (KY131) and *osubc12* mutant seeds under 1 μM (**g**) or 2 μM (**h**) ABA at three timepoints. Values are means ± SE from three individual biological replicates (25 seeds per biological replicate). The data were statistically analyzed using two-tailed Student's *t*-test (*$P < 0.05$, **$P < 0.01$). Source data are provided as a Source Data file.

## OsUBC12 mainly catalyzes K48-linked polyubiquitination and promotes OsSnRK1.1 degradation

It should be noted that E2s exert different effects on target proteins by mediating different polyubiquitination modifications[23]. We then explored the type of polyubiquitination mediated by OsUBC12. We detected polyUb conjugates when using a typical Ub or a Ub-K63R

variant that lacks K63 (Supplementary Fig. 16, lanes 2 and 3), but not a Ub-K48R variant lacking K48 (Supplementary Fig. 16, lane 4), indicating that OsUBC12 mainly catalyzes K48-linked polyubiquitination. Therefore, we examined whether OsUBC12 mediates the proteasomal degradation of OsSnRK1.1. To this end, we performed a cell-free degradation assay with immunoblot analysis to measure OsSnRK1.1-

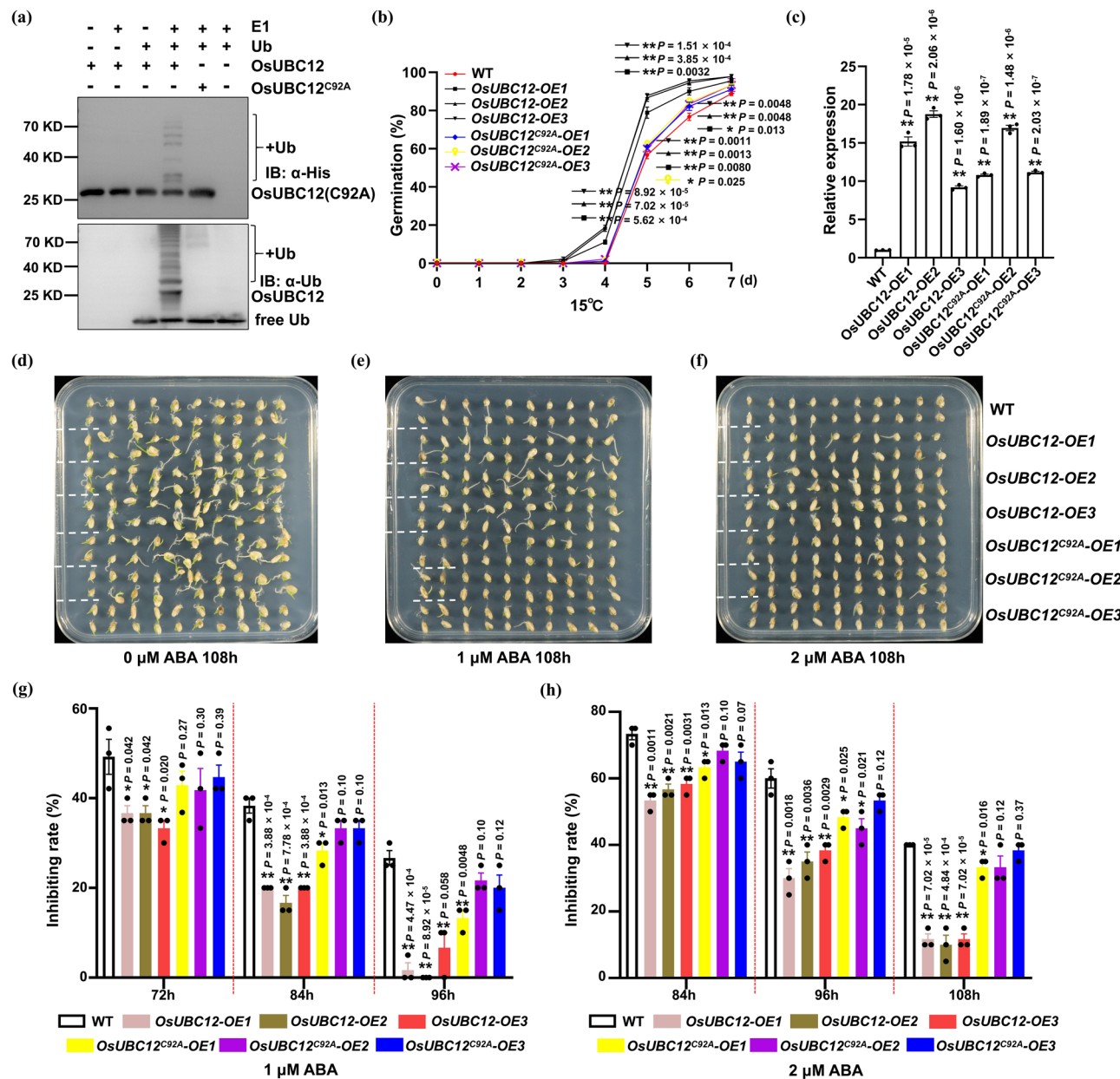

**Fig. 4 | OsUBC12's role in regulating LTG mainly depends on its conservative ubiquitination function. a** Formation of the ubiquitin thioester bond was analyzed for OsUBC12 and OsUBC12^C92A using anti-His and anti-Ub antibodies. **b** Time-course germination analysis of WT (KY131), *OsUBC12-OE* and *OsUBC12^C92A-OE* lines over 7 days at 15 °C. Values are means ± SE from three individual biological replicates (30 seeds per biological replicate). The data were statistically analyzed by two-tailed Student's *t*-test (**P* < 0.05, ***P* < 0.01). **c** Relative expression of *OsUBC12* in seeds of *OsUBC12-OE* and *OsUBC12^C92A-OE* lines. The expression level of the control samples (WT, KY131) was set to 1. Values are means ± SE from three individual replicates (*n* = 3). The housekeeping gene *OsUBQ5* was used as an internal control to normalize the data. The data were statistically analyzed using two-tailed Student's *t*-test (**P* < 0.05, ***P* < 0.01). Germination performance of WT (KY131), *OsUBC12-OE* and *OsUBC12^C92A-OE* seeds after 108 h on ½-MS agar medium containing 0 μM (**d**), 1 μM (**e**) or 2 μM (**f**) ABA. The germination inhibition rates of WT (KY131), *OsUBC12-OE* and *OsUBC12^C92A-OE* seeds under 1 μM (**g**) or 2 μM (**h**) ABA at three timepoints. Values are means ± SE from three individual biological replicates (20 seeds per biological replicate). The data were statistically analyzed using two-tailed Student's *t*-test (**P* < 0.05, ***P* < 0.01). Source data are provided as a Source Data file.

His abundance. Compared to WT extract, the degradation rate of OsSnRK1.1 was decreased in *osubc12* extract (Fig. 6a). Notably, OsSnRK1.1 degradation was inhibited by the proteasome inhibitor MG132, irrespective of whether protein extracts from WT or *osubc12* mutants were assayed (Fig. 6b). These data indicated that OsUBC12 promotes the degradation of OsSnRK1.1, possibly via the 26 S proteasome pathway.

Furthermore, overexpression of *OsUBC12*, but not of a *OsUBC12^C92A* mutant, enhanced the degradation of OsSnRK1.1 (Fig. 6c). In protoplasts degradation experiment also suggested that OsUBC12-Flag obviously promoted the degradation of OsSnRK1.1-Flag compared to empty-Flag (EV) and OsSnRK1.1-Flag co-expression, in contrast, this degradation was less pronounced in the presence of OsUBC12^C92A-Flag (Fig. 6d). To further verify whether OsUBC12 could affect the OsSnRK1.1 protein level in vivo, we performed western blotting to measure OsSnRK1.1 protein levels in WT, *osubc12* mutants, *OsUBC12-OE* and *OsUBC12^C92A-OE* transgenic lines using anti-OsSnRK1.1. The results showed that OsSnRK1.1 protein levels were clearly increased in *osubc12* mutants and obviously decreased in *OsUBC12-OE* lines compared to WT, whereas the reduction of OsSnRK1.1 protein levels in

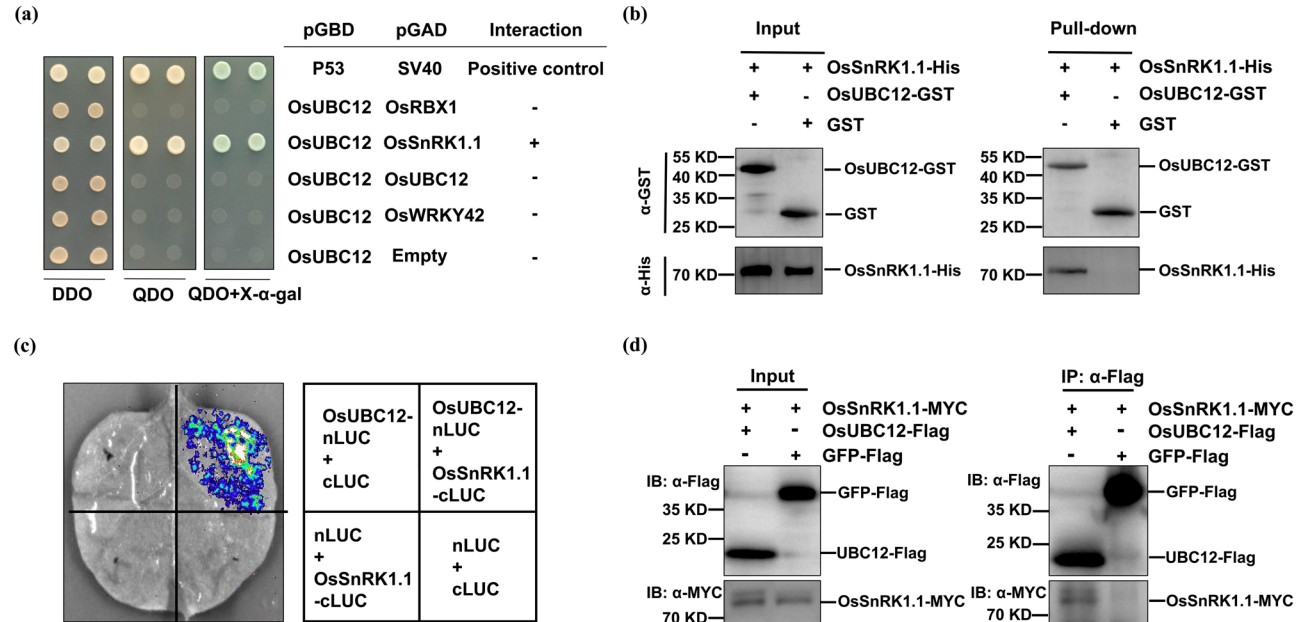

**Fig. 5 | OsUBC12 interacts with OsSnRK1.1. a** OsUBC12 interacts with OsSnRK1.1 in a yeast two-hybrid system. The yeast cells harboring the bait and prey vectors were selected on SD medium lacking Leu and Trp (DDO). Interaction was assessed based on the growth of yeast cells on selective SD medium lacking Leu, Trp, His and Ade (QDO) or on SD medium lacking Leu, Trp, His and Ade (QDO) but containing X-α-Gal for 3 days at 30 °C. The pGBD-P53 + pGAD-SV40 combination was used as a positive control. X-α-Gal represents 5-bromo-4-chloro-3-indolyl-α-D-galactoside. **b** In vitro pull-down assays showing the interaction of OsUBC12 with OsSnRK1.1. His-tagged SnRK1.1 was incubated with immobilized GST or GST-tagged UBC12, and immunoprecipitated fractions were detected using an anti-His antibody. **c** Interaction between OsUBC12 and OsSnRK1.1 in luciferase complementation imaging (LCI) assays. Co-transformation of OsUBC12-nLUC and OsSnRK1.1-cLUC led to reconstitution of the LUC signal, whereas no signal was detected upon co-expression of OsUBC12-nLUC and cLUC, nLUC and OsSnRK1.1-cLUC, or cLUC and nLUC. **d** Co-IP assay indicates that OsUBC12 interacts with OsSnRK1.1 in planta. OsUBC12-Flag and OsSnRK1.1-Myc were co-expressed in *N. benthamiana*. Co-transformation of GFP-Flag and OsSnRK1.1-Myc was used as the control. Protein extract was immunoprecipitated with anti-Flag-Tag Mouse mAb (Agarose Conjugated) and detected with anti-MYC (HRP) or anti-FLAG (HRP) antibodies, respectively. Source data are provided as a Source Data file.

*OsUBC12$^{C92A}$-OE* lines was not pronounced (Fig. 6e). These results confirm that OsUBC12 promotes OsSnRK1.1 degradation in vitro and in vivo, and requires its active-site cysteine residue to do so. We also found that OsSnRK1.1 expression was not obviously altered in *osubc12* knockout mutants and transgenic *OsUBC12-OE* lines (Supplementary Fig. 17a, b), indicating that the regulatory effects of OsUBC12 on OsSnRK1.1 likely occur mainly at the post-translational rather than the transcriptional level.

## OsSnRK1.1 acts downstream of OsUBC12 in controlling LTG

We then explored the possible role of OsSnRK1.1 in controlling LTG by analyzing the germination rates of *OsSnRK1.1-OE* lines at 15 °C and 30 °C. The *OsSnRK1.1-OE* transgenic lines were examined by fluorescence screening (Supplementary Fig. 14c) and qRT-PCR (Supplementary Fig. 14d). Compared with WT, the germination of *OsSnRK1.1-OE* lines was strongly delayed at 15 °C (Fig. 7a) but only slightly delayed at 30 °C (Supplementary Fig. 8e). Accordingly, *OsABI5* and *OsRAB21* expression levels were higher in *OsSnRK1.1-OE* lines than in WT (Supplementary Fig. 18a, b). Moreover, ABA sensitivity assays showed that *OsSnRK1.1-OE* lines exhibited delayed germination in the absence of ABA (Supplementary Fig. 19a) and were more sensitive than WT to ABA (Supplementary Fig. 19b, c). The germination inhibition rates of *OsSnRK1.1-OE* lines were also significantly higher than in WT after 1 or 2 μM ABA treatment (Supplementary Fig. 19d, e), indicating that *OsSnRK1.1* overexpression increased seed sensitivity to ABA. These results indicate that OsSnRK1.1 inhibit LTG by enhancing ABA signaling, a functionality opposite to that of OsUBC12.

To further reveal the functionality of OsUBC12 and the negative role of UBC12 in SnRK1-dependent ABA signaling during LTG, we performed the genetic interaction analysis by generating *OsSnRK1.1* homozygous knockout mutants (*ossnrk1.1-1*, *ossnrk1.1-2*, and

*ossnrk1.1-3*) (Supplementary Fig. 20) and *osubc12snrk1.1* double mutants (*osubc12snrk1.1-1*, *osubc12snrk1.1-2*, and *osubc12snrk1.1-3*) in the KY131 background (Supplementary Fig. 21). Germination tests displayed that *ossnrk1.1* mutants showed clearly increased germination rates at low temperature (15 °C) (Fig. 7b), but only a slight increment in germination at 30 °C (Supplementary Fig. 8f), and knockout of *OsSnRK1.1* rescued the increased cold sensitive phenotype of *osubc12* during LTG (Fig. 7b). The genetic relationship between *OsUBC12* and *OsSnRK1.1* in ABA sensitivity was analyzed too. Similarly, knockout of *OsSnRK1.1* rescued the increased ABA sensitive phenotype of *osubc12* (Fig. 7b–g), indicating that OsSnRK1.1 acts downstream of OsUBC12 to regulate LTG and ABA responses. Taken together, our results suggest that OsUBC12 negatively regulate ABA signaling by promoting the degradation of OsSnRK1.1, thus accelerating LTG.

## Discussion

Although the control of seed germination is important for plant adaptability, knowledge of the genetic basis and molecular mechanisms regulating this trait remains limited. In the present study, OsUBC12 is genetically and molecularly identified as a previously unrecognized regulator of seed germination, preferentially functioning at 15 °C, but performing a fine-tuning at 30 °C. We also revealed that OsUBC12-mediated ubiquitination controls LTG by affecting ABA signaling, and provided a genetic reference for improving the low-temperature germinability of *indica* rice.

Arabidopsis encodes 37 E2 ubiquitin-conjugating enzymes (UBCs)[26], many of whose functions have been elucidated. However, rice encodes 48 UBCs[25], and in-depth studies of their functions are lacking. Here, we found that OsUBC12 accelerated seed germination at low temperature. *osubc12* mutants generated via CRISPR/Cas9-

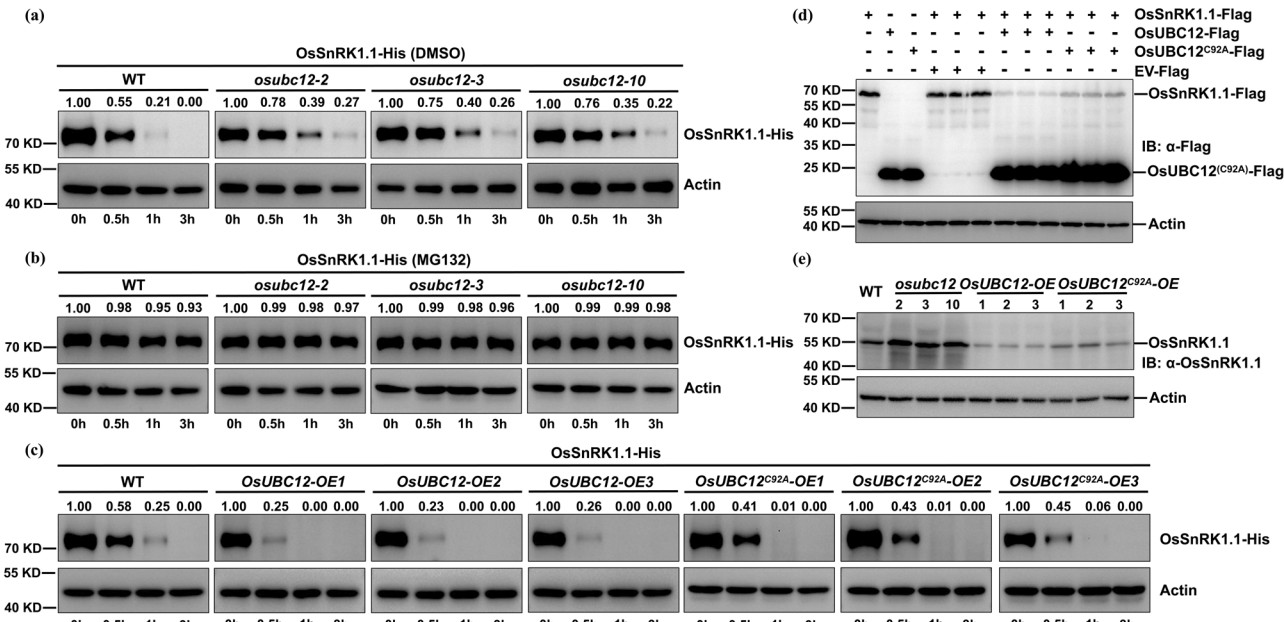

**Fig. 6 | OsUBC12 promotes OsSnRK1.1 degradation.** Cell-free degradation assays of OsSnRK1.1-His incubated with protein extracts from WT (KY131) or *osubc12* mutants following a 1-h treatment with DMSO (**a**) or 100 μM MG132 (**b**). DMSO is the solvent used for MG132 preparation. Protein extracts from WT (KY131) and *osubc12* mutants were incubated with OsSnRK1.1-His for the indicated durations. OsSnRK1.1-His levels were visualized by immunoblotting using an anti-His antibody. OsActin was used as the loading control. To quantify relative protein band intensity, the protein band at 0 h was set to 1.00. **c** Cell-free degradation assays of OsSnRK1.1-His incubated with protein extracts from WT (KY131) and *OsUBC12-OE* or *OsUBC12^{C92A}-OE* transgenic rice for the indicated durations. OsSnRK1.1-His levels were visualized by immunoblotting using an anti-His antibody. OsActin was used as the loading control. To quantify relative protein band intensity, the protein band at 0 h was set

to 1.00. **d** In protoplasts degradation experiment of OsUBC12-Flag and OsSnRK1.1-Flag co-expression in rice protoplasts. The co-expression of empty-Flag (EV) and OsSnRK1.1-Flag was used as the control. OsActin was used as the loading control. **e** OsSnRK1.1 protein levels of WT, *osubc12* mutants, *OsUBC12-OE* and *OsUBC12^{C92A}-OE* transgenic lines. Total proteins extracted from the seeds of WT, *osubc12* mutants, *OsUBC12-OE* and *OsUBC12^{C92A}-OE* transgenic lines, were subjected to precipitates with anti-OsSnRK1.1 or anti-OsActin respectively by western blotting analysis. OsActin was used as the loading control. OsSnRK1.1 protein levels were visualized by western blotting using an anti-OsSnRK1.1 antibody. The anti-OsSnRK1.1 antibody was prepared by Abmart Shanghai Co.,Ltd. Source data are provided as a Source Data file.

---

mediated genome editing showed considerably decreased germinability under low temperature (15 °C) (Fig. 1e, f), but only slightly slower germination at 30 °C (Supplementary Fig. 8b, c), implying that OsUBC12 preferentially accelerates seed germination at low temperature, but fine-tunes it at 30 °C. Considering the temperature-dependent function of OsUBC12 and its biochemical properties as an E2 ubiquitin-conjugating enzyme, we analyzed the effect of low temperature on the post-translational level of UBC12. The results showed that low temperature could promote the accumulation of OsUBC12 protein in vitro and in protoplasts (Supplementary Fig. 22a, b). At the transcriptional level, *OsUBC12* was induced by both cold and ABA. (Fig. 3a). These observations further indicate the priority regulatory mechanism of UBC12 for low-temperature germination. Moreover, Go enrichment analysis of DEGs in *osubc12* mutants vs. WT not only showed the enrichment of low-temperature related genes (Supplementary Fig. 12b), but also the enrichment of ABA signaling genes (Supplementary Fig. 12b). The expression levels of *OsABI5* and *OsRAB21* were higher in *osubc12* mutants than in WT (Fig. 3b, c). Knockdown of *OsUBC12* also increased ABA sensitivity (Fig. 3d–h). These findings indicate that OsUBC12-promoted LTG is associated with suppression of ABA signaling. The results obtained by overexpression analysis of transgenic rice lines showed an agreement with that of *osubc12* mutants, and blocking the 92th cysteine site of OsUBC12 could significantly suppress its regulatory effect on germination (Fig. 4b–h). Our results thus provide evidence that OsUBC12 accelerates LTG by repressing ABA signaling via its conserved ubiquitination function.

Several E2s play specific roles in ABA signaling in Arabidopsis. For example, AtUBC26 forms complexes with the ABA receptors PYR1 and PYL4 to negatively regulate ABA signaling[78], while AtUBC32, AtUBC33 and AtUBC34 negatively regulate ABA-mediated stomatal closure and drought tolerance[79]. Our findings broaden the knowledge of E2-regulated ABA signaling. More importantly, we revealed that OsUBC12 regulates LTG, a key trait in rice, by negatively regulating ABA signaling.

Moreover, E2s directly bind target proteins to regulate specific functions. In Arabidopsis, the E2 conjugase PHOSPHATE 2 (PHO2) directly binds and modulates the stability of the phosphate transporter PHO1 in phosphate homeostasis[80]. In addition, UBC27 promotes the degradation of the ABA co-receptor ABI1 via the 26 S proteasome, likely through K48-linked polyubiquitination, to regulate ABA signaling and drought tolerance[30]. Similarly, we demonstrated that OsUBC12 interacts with OsSnRK1.1in vitro and *in planta* (Fig. 5). OsUBC12 mainly catalyzes K48-linked polyubiquitination and promotes the degradation of OsSnRK1.1, with the latter process requiring the OsUBC12 active-site cysteine residue (Fig. 6 and Supplementary Fig. 15). SnRK1s and SnRK2s, core components of ABA signaling, both belong to the plant serine/threonine protein kinase family[81]. In Arabidopsis, SnRK1.1 positively regulates ABA signaling, and its overexpression delays germination and growth[82] and causes ABA hypersensitivity[56]. Consistent with these results, OsSnRK1.1 inhibits LTG by enhancing ABA signaling (Fig. 7a, Supplementary Figs. 18a, b, and 19a–e). Further genetic interaction analysis indicated that OsSnRK1.1 acts downstream of OsUBC12 to regulate LTG and ABA responses (Fig. 7b–g). Thus, OsUBC12 negatively regulates ABA signaling by promoting OsSnRK1.1 degradation, thereby accelerating LTG. Although it has been reported that SnRK1 positively regulates ABA signaling which is well known in the role of seed germination, and one of the most common 'brakes' that weaken the ABA signaling is degradation of the core signaling

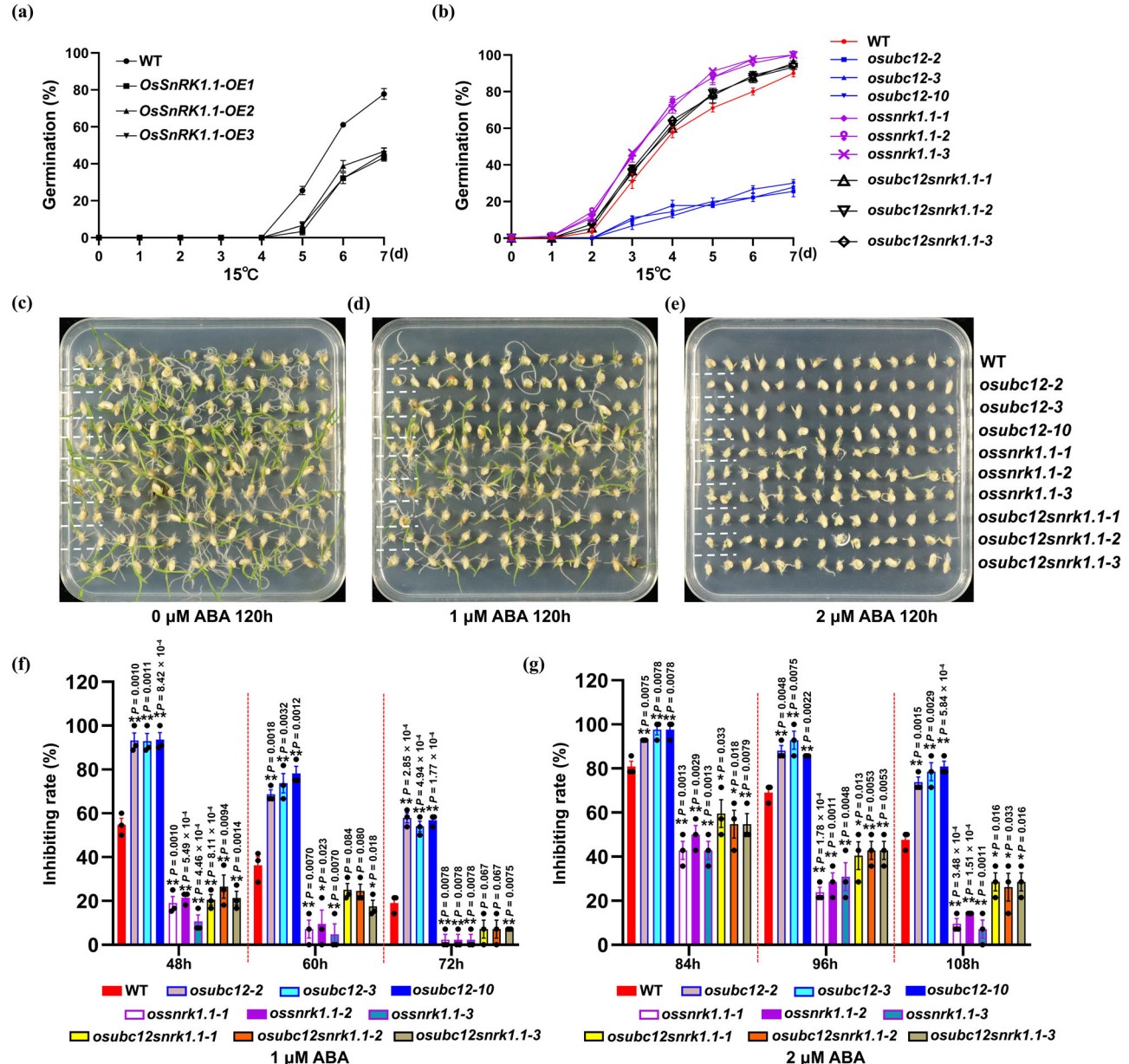

**Fig. 7 | OsSnRK1.1 acts downstream of OsUBC12 in controlling LTG. a** Time-course germination analysis of WT (KY131) and *OsSnRK1.1-OE* lines at 15 °C. Values are means ± SE from three individual biological replicates (30 seeds per biological replicate). **b** Time-course germination analysis of WT (KY131), *osubc12* mutants, *ossnrk1.1* mutants and *osubc12snrk1.1* double mutants at 15 °C. Values are means ± SE from three individual biological replicates (30 seeds per biological replicate). Germination performance of WT (KY131), *osubc12* mutants, *ossnrk1.1* mutants and *osubc12snrk1.1* double mutants seeds after 120 h on ½-MS agar medium containing 0 μM (**c**), 1 μM (**d**) or 2 μM (**e**) ABA. The germination inhibition rates of WT (KY131), *osubc12* mutants, *ossnrk1.1* mutants and *osubc12snrk1.1* double mutants seeds under 1 μM (**f**) or 2 μM (**g**) ABA at three timepoints. Values are means ± SE from three individual biological replicates (14 seeds per biological replicate). The data were statistically analyzed using two-tailed Student's *t*-test (*P < 0.05, **P < 0.01). Source data are provided as a Source Data file.

components by ubiquitin-mediated degradation pathways. However, the function and mechanism of E2 ubiquitin-conjugating enzymes, the key enzyme of ubiquitin-proteasome system (UPS), in regulating LTG through ABA signaling have not been reported yet. Compared to our known or easily deductive knowledge, the exploration of the module E2 (OsUBC12)-OsSnRK1 and its ubiquitination-regulatory mechanism in LTG are the key findings in terms of mechanistic novelty of this study. Moreover, in addition to regulating ABA signaling, SnRK1 also functions as an energy regulator integrating carbohydrate, starch, and lipid metabolism ect[83–85], while Go enrichment analysis of DEGs in *osubc12* mutants vs. WT revealed that OsUBC12 may also be involved in some metabolic pathways, including carbohydrate metabolic process, sucrose metabolic process, starch metabolic process, and lipid

metabolism process ect (Supplementary Fig. 12b). This also suggests the connection between OsUBC12 and OsSnRK1.1 from the other side, and OsSnRK1.1 may be the important target protein of OsUBC12.

Notably, we found that a transposon insertion in the *OsUBC12* promoter promotes seed germination at low temperatures (Fig. 1). Natural variation analysis revealed that transposon insertion in the *OsUBC12* promoter mainly occurred in the *japonica* varieties which are planted in both tropical and cooler temperate regions globally[63], and that the transposon is absent in wild and *indica* accessions which are mostly cultivated in tropical regions (Supplementary Table 2). Indeed, the *japonica* subpopulation in general has a greater low-temperature germinability than *indica*, and most alleles with increased low-temperature germinability were found in *japonica*[4,13,14,86,87]. This

finding is consistent with the notion that the *japonica* allele with the transposon insertion conferring low-temperature germinability may have facilitated *temperate japonica* varieties to adapt to cooler climate[11], and the *japonica* allele could be useful to enhance the low-temperature germinability trait of *indica*.

Furthermore, extensive introgressions have occurred between modern GJ (Geng/*japonica*) and XI (Xian/*indica*) rice varieties during modern rice breeding, and introgression from GJ into XI was much greater than introgression from XI into GJ[66,67,88]. Interestingly, we demonstrated that the *japonica OsUBC12* locus (transposon insertion) has been introgressed into the modern elite *indica* two-line male sterile lines Y58S and Jing 4155 S (Fig. 2a). We also investigated the low-temperature germinability of 50 varieties without transposon insertion and 21 varieties with transposon insertion, which from the majority of the 69 cultivated accessions selected from nine rice subpopulations of the 3000 rice genomes as well as all varieties and sterile lines in Fig. 2a, confirming that varieties carrying the *japonica OsUBC12* locus (transposon insertion) have higher low-temperature germinability than varieties without the locus (Fig. 2b). Under direct-seeding conditions at low temperature (18 °C), seedlings of the introgression lines SL2116 and SL2117 emerged faster than those of IR64, while *osubc12* mutants emerged slower than WT (Fig. 2c). These results suggest the potential applicability of *japonica OsUBC12* in improving the low-temperature germinability of *indica* rice. Our study not only sheds light on the possible genomic contributions of *japonica OsUBC12* locus introgressions to trait improvements of *indica* rice cultivars, but also provides a genetic reference for improving the LTG of *indica* rice.

Taking these results together, we propose a model to explain how OsUBC12 regulates LTG (Supplementary Fig. 23). According to our model, OsSnRK1.1 functions as a downstream key regulator to enhance ABA responses by up-regulating the expression of ABA-signaling-related genes such as *OsABI5* and *OsRAB21*, thus inhibiting LTG. OsUBC12, an E2 enzyme for K48-linked polyubiquitination, recruits and degrades OsSnRK1.1. Compared with *indica* rice, a transposon insertion in the *japonica OsUBC12* promoter activates its expression. Increased OsUBC12 levels further promote the degradation of OsSnRK1.1, thereby weakening OsSnRK1.1-regulated ABA signaling and enhancing the low-temperature germinability of *japonica* rice. Although we provided valuable information for the fields of plant biology and rice breeding, more interesting problems and possible details need to be uncovered in the future study. For example, in addition to directly recruiting targets, E2s interact with E3 ubiquitin ligases, which specifically bind substrates[23]. It remains unclear whether specific E3 ligases participate in the OsUBC12-mediated degradation of OsSnRK1.1. Further studies are needed to identify other OsUBC12 ubiquitination targets and interacting partners, which would reveal its substrate diversity and functional specificity. SnRK1s are serine/threonine protein kinases which requires activation of the T-loop by SnAK/GRIK kinases to increase its phosphorylation activity and phosphorylate targets[89,90], it would be interesting to further study the cross-talk between OsSnRK1-mediated phosphorylation and OsUBC12-mediated ubiquitination. Moreover, given the complex nature of the LTG trait in rice, a near-isogenic line (NIL) with the strong allele of *OsUBC12* introduced to an *Indica* background, or a NIL with the strong allele of *OsUBC12* introduced to a *Japonica* background with the weak allele will be constructed for testing LTG to explore the application prospect of this allele in LTG breeding, and determining how to use the *japonica OsUBC12* locus to efficiently improve the LTG of *indica* rice remains the major goals.

## Methods
### Plant materials and growth conditions
The chromosome single-segment substitution lines (CSSLs) with Koshihikari introgression in IR64 were grown in a growth chamber (white fluorescent tubes, 200–300 µmol m-2 s-1) under a 10 h light/14 h dark cycle at 30 °C. *OsUBC12* homozygous knockout mutants

(*osubc12-ks1* and *osubc12-ks2*) in Koshihikari background were grown in a growth chamber (white fluorescent tubes, 200–300 µmol m-2 s-1) under a 10 h light/14 h dark cycle at 30 °C. The F$_2$ population was grown in Hainan province, China (winter), under natural conditions. Transgenic *OsUBC12* in the KY131 background Transgenic *OsUBC12* and *OsSnRK1.1* plants in the KY131 background were grown in the field at the experimental stations of the Northeast Institute of Geography and Agroecology, Chinese Academy of Sciences, in Harbin, Heilongjiang province, China. All varieties and sterile lines used in the analysis of the relationship between the *japonica OsUBC12* locus (transposon insertion) and LTG were grown in a growth chamber (white fluorescent tubes, 200–300 µmol m-2 s-1) under a 10 h light/14 h dark cycle at 30 °C. The growth conditions and sampling times of the control lines involved in each experiment in this study were consistent with their related CSSLs, mutants, transgenic lines or varieties.

### Evaluation of germination rate
Germination rate was evaluated as described by Yoshida et al.[37] and Fujino et al.[3] with minor modifications. Seeds of the control and experimental groups were grown under the same conditions and collected at 45 days after flowering, air-dried, and stored at 45 °C for 3 days to break dormancy. To evaluate the seed germination of the CSSLs, 30 seeds per line were placed on filter paper in a 9-cm Petri dish, 10 mL of distilled water was added, and plates were incubated at 30 °C for 24 h or 15 °C for 60 h under dark conditions to induce germination. To evaluate *OsUBC12* and *OsSnRK1.1* transgenic lines, 30 seeds per line were incubated for 7 days at 30 °C or 15 °C under dark conditions. Germination was considered to have occurred when the epiblast was broken and the white embryo had emerged to a certain length[37]. Germinated seeds were counted and the germination rates (%) were calculated by dividing them with those germinated at 30 °C for 48 h, under which all viable seeds were thought to have germinated.

### Cloning of *OsUBC12*
To clone *OsUBC12*, SL2117 was backcrossed with IR64 to construct an F$_2$ segregating population. Fixed homozygous recombinant plants were screened by molecular markers covering the target genomic region. The target gene was then identified by repeated phenotypic characterization of these fixed recombinants. The primer sequences are listed in Supplementary Data 1.

### RNA extraction and quantitative reverse transcription PCR (RT-qPCR)
Total RNA was isolated from seeds using TRIzol reagent and reverse transcription was performed using the ReverTra Ace Kit (Toyobo) according to the manufacturer's instructions. RT-qPCR was employed to measure gene expression levels using the SYBR qPCR Mix kit and a LightCycler® 96 System (Roche). The gene expression levels were calculated by the $2^{-\Delta\Delta Ct}$ method with *OsUBQ5* (LOC_Os01g22490) as the internal control. The primers used for expression analysis are shown in Supplementary Data 1.

### Transient dual-luciferase (dual-LUC) assay
The *OsUBC12^{IR64}* (2 kb) or *OsUBC12^{Koshihikari}* (7.9 kb, containing the inserted transposon) promoter sequence was cloned into pGreenII-0800-LUC, and subsequently transformed into rice protoplasts. Moreover, *pOsUBC12^{KoshihikariΔTransposon}* (*pOsUBC12^{Koshihikari}* but lacking the transposon) and *pOsUBC12^{IR64+Transposon}* (*pOsUBC12^{IR64}* but containing the transposon, the transposon was artificially inserted into the same position as the *pOsUBC12^{Koshihikari}*) were constructed by ClonExpressTM MultiS One Step Cloning Kit (vazyme, C113). The primer sequences are listed in Supplementary Data 1. The Renilla luciferase (REN) gene directed by the 35 S promoter in the pGreenII 0800-LUC vector was used as an internal control. Firefly LUC and REN activities were measured using the Dual-Luciferase reporter assay kit (Beyotime) and a

GloMax 20/20 luminometer (Promega). LUC activity was normalized to REN activity and LUC/REN ratios were calculated. The data presented are the averages of at least three independent replicates.

## Plasmid construction and genetic transformation

Knockout *osubc12-ks* mutants, *osubc12* mutants, *ossnrk1.1* mutants and *osubc12snrk1.1* double mutants were generated by CRISPR/Cas9-mediated genome editing[91]. The *OsUBC12* or OsSnRK1.1 guide RNA sequence (Supplementary Data 1) was introduced into the CRISPR/Cas9 binary vector pYLCRISPR/Cas9P$_{ubi}$-H, respectively. To generate the *osubc12snrk1.1* double mutants, the target regions of *OsUBC12* and *OsABI5* (Supplementary Data 1) were introduced into pYLCRISPR/Cas9P$_{ubi}$-H. The recombinant vectors were transformed into *Agrobacterium tumefaciens* strain EHA105-pSOUP for rice genetic transformation, respectively. To generate overexpression lines, the coding sequences (CDS) of *OsUBC12* and *OsSnRK1.1* were amplified from *japonica* cv. Nipponbare cDNA. The active-site cysteine residue of *OsUBC12* was mutated by site-directed mutagenesis to generate *OsUBC12*$^{C92A}$, which was subsequently cloned into modified pCAMBIA2300 with the GFP gene (as the selectable marker) under the control of the *Actin1* promoter. Except that *osubc12-ks* mutants were Koshihikari background, KY131 was used for *Agrobacterium*-mediated transformation to generate *osubc12* mutants, *ossnrk1.1* mutants, *osubc12snrk1.1* double mutants and overexpression lines. Primers used for gene editing and plasmid construction are listed in Supplementary Data 1.

## Natural variations of *OsUBC12*

To determine whether the transposon was present in the promoter, we took the 500-bp sequence in the promoter region from the IR64 genome which do not contains the transposon insertion as a probe. And in comparison, the transposon found in the Nipponbare genome is inserted in the 100-bp position of the probe. The 500-bp probe was blast-searched in the target genome assemblies. If the probe sequence can be mapped to the assembly in full length, the assembly is called as not having the transposon; if the first 100-bp is missing from the alignment, the assembly is called as having the transposon. The assemblies called as having the transposon was further confirmed by blast search using the Nipponbare haplotype.

## RNA sequencing (RNA-seq) analysis

Three independent *osubc12* mutant and wild-type (WT, KY131) seeds were grown and harvested from the field at the experimental stations of the Northeast Institute of Geography and Agroecology, Chinese Academy of Sciences, in Harbin, Heilongjiang province, China. Seeds were soaked in sterile water at 15 °C for 12 h and then drained, followed by RNA extraction for RNA-seq analysis. The extraction and examination of total RNA, library preparation and Illumina sequencing were performed by Majorbio Bio-Pharm Technology Co., Ltd. (Shanghai, China). An Illumina HiSeq™ 2500 was used to conduct high-throughput transcriptome analysis of *osubc12* mutant and WT (KY131) seeds, and 40,444,345 and 41,135,767 clean reads were obtained, respectively. The lengths of most genes were distributed about 1800 bp. Transcriptome analysis was conducted using the complete genomic sequence of Nipponbare rice as a reference downloaded from IRGSP-1.0 (http://rapdb.dna.affrc.go.jp/download/irgsp.html). The Poisson-dispersion model of fragments was used to conduct statistical analysis (false discovery rate (FDR) < 0.05) and gene expression levels were estimated by fragments per kilobase of transcript per million fragments mapped (FPKM). Differentially expressed genes (DEGs) between samples were defined by DESeq, based on fold change |log$_2$ ratio | > 1.2 and an adjusted *P* < 0.05. Gene Ontology (GO) enrichment analysis of the DEGs was performed using the GOseq R package based on the Wallenius noncentral hypergeometric distribution, which adjusts for gene length bias in DEGs.

## ABA sensitivity assay

Rice seeds were placed on half-strength Murashige and Skoog (½-MS) agar medium containing 0, 1, and 2 µM ABA and incubated at 30 °C. Germination rates were assessed every 12 h for 120 h. The formula for calculating the germination inhibition rate by ABA at designated timepoints is as follows:

$$\text{Inhibition rate (\%)} = [(\text{Number of germinated seeds at 0}\mu\text{M ABA} \\ - \text{Number of seeds germinated at 1 or 2}\mu\text{M ABA})/ \quad (1) \\ \text{Number of germinated seeds at 0}\mu\text{M ABA}] * 100\%$$

## Ubiquitin-conjugating enzyme activity assay

The E2 ubiquitin-conjugating activity assay was conducted as described by Zhao et al. [92] with minor modifications. *OsUBC12* and *OsUBC12*$^{C92A}$ were individually cloned into the pET29b(+) expression vector. The fusion proteins OsUBC12-His and OsUBC12$^{C92A}$-His were purified at 4 °C and quantified according to the pET System Manual. Buffer, ubiquitin, E1 enzyme, OsUBC12-His or OsUBC12$^{C92A}$-His and other components were combined in the reaction tube to prepare the reaction system. The reaction was performed at 37 °C for 3 h. Subsequently, in vitro ubiquitin thioester bond formation was detected using anti-His (Abmart, code number M20001S) and anti-Ub (Beyotime, code number AF1705) antibodies.

## Yeast two-hybrid assay

The *OsUBC12* CDS was cloned into the pGBKT7 (bait) vector, and the recombinant plasmid and pGADT7 (prey) empty vector were co-transformed into the Y2HGold yeast strain to test for self-activation. Subsequently, OsUBC12 was used as the bait protein to screen against a yeast two-hybrid cDNA library, and candidate clones were obtained through sequencing and alignments. The CDS of candidate clones was independently cloned into the pGADT7 vector. These prey plasmids and the *OsUBC12* bait vector were co-transformed, and interactions were verified by a point-to-point yeast two-hybrid system. The yeast cells were selected on SD/-Leu/-Trp (DDO), and interaction was assessed based on their ability to grow on SD/-Leu/-Trp/-His/-Ade (QDO) or SD/-Leu/-Trp/-His/-Ade/+X-α-Gal for 3 days at 30 °C.

## In vitro pull-down assay

The *OsSnRK1.1* CDS was cloned into the pET29b(+) expression vector. The recombinant protein was purified at 4 °C and quantified according to the pET System Manual. The *OsUBC12* CDS was inserted into the pGEX-4T-1 expression vector and expressed in Rosetta (DE3) *Escherichia coli* cells. The target protein OsUBC12-GST was purified with GST resin (GE Healthcare). The pulled-down proteins were eluted and detected by immunoblotting using anti-GST (Abmart, code number M20007S) and anti-His (Abmart, code number M20001S) antibodies, respectively.

## Firefly luciferase complementation imaging (LCI) assay

*OsUBC12* and *OsSnRK1.1* were cloned in-frame with the N-terminal and C-terminal fragments of the luciferase reporter gene to generate pCAMBIA1300-*OsUBC12*nLUC and pCAMBIA1300-*OsSnRK1.1*cLUC, respectively. *Agrobacteria* harboring these constructs were co-infiltrated into *Nicotiana benthamiana* leaves, which were subsequently sprayed with luciferin (1 mM luciferin and 0.01% Triton X-100) and photographed using Chemiluminescence imaging (Tanon 5200) at 72 h after infiltration.

## Co-IP assay

The *OsUBC12* CDS was cloned into pCAMBIA1300-Flag expression vector to generate pCAMBIA1300-*OsUBC12*-Flag. To generate *OsSnRK1.1*-Myc recombinant vector, *OsSnRK1.1* in pENTRY vector was cloned into pGWB18 through LR recombination reaction. *Agrobacteria*

harboring these constructs were co-infiltrated into *Nicotiana benthamiana* leaves. The total proteins were extracted using the IP lysis buffer (Beyotime, P0013). Co-transformation of GFP-Flag and OsSnRK1.1-Myc was used as the control. Protein extract was immunoprecipitated with anti-Flag-Tag Mouse mAb (Agarose Conjugated) (Abmart, code number M20018S) and detected with anti-MYC (HRP) (Abmart, code number M20002S) or anti-FLAG (HRP) (Abmart, code number M20008M) antibodies, respectively.

### In protoplasts degradation experiment
The *OsUBC12$^{C92A}$* and *OsSnRK1.1* CDS were cloned into pCAMBIA1300-Flag expression vector to generate pCAMBIA1300-*OsUBC12$^{C92A}$*-*Flag* and pCAMBIA1300-*OsSnRK1.1-Flag*, respectively. The combinations of Empty-Flag (EV) + pCAMBIA1300-*OsSnRK1.1-Flag*, pCAMBIA1300-*OsUBC12-Flag* + pCAMBIA1300-*OsSnRK1.1-Flag* and pCAMBIA1300-*OsUBC12$^{C92A}$*-*Flag* + pCAMBIA1300-*OsSnRK1.1-Flag* were co-transfected into rice protoplasts. Different combinations of plasmid were transiently expressed in the protoplasts by PEG-mediated transfection. Following overnight incubation in the dark at 28 °C, total proteins were isolated from the protoplasts with SDS Lysis Solution (Beyotime, P0013G). The protein extracts were detected by immunoblotting using anti-Flag antibody (Abmart, code number M20008M). OsActin was used as the loading control.

### Analysis of polyubiquitination
Ubiquitin and the Ub-K63R and Ub-K48R variants were used as basic components to prepare reaction systems with E1, OsUBC12-His, etc. The polyubiquitination type of OsUBC12 was assessed by immunoblotting using anti-His (Abmart, code number M20001S) and anti-Ub (Beyotime, code number AF1705) antibodies.

### Cell-free protein degradation assays
Cell-free protein degradation assays were performed as described by ref. 93 with some modifications. Total proteins were extracted from transgenic lines and WT seeds with degradation buffer. Each cell-free protein degradation reaction contained 500 μg total protein and 100 ng of OsSnRK1.1-His purified from *E. coli* Rosetta (DE3) cells. For the proteasome inhibitor experiments, 100 μM MG132 was added to the total proteins 60 min prior to the cell-free degradation experiment. The reactions were incubated at 22 °C. The mixed solutions were collected at designated time points (0, 0.5, 1, and 3 h) and examined by immunoblotting using an anti-His antibody (Abmart, code number M20001S). Results were quantified using ImageJ software (https://imagej.nih.gov/ij/index.html).

### Statistics and reproducibility
All experiments were repeated at least three times independently with similar results. Numbers (*n*) of samples or replicates are indicated in figure legends. The data are presented as mean ± standard error (SE). Data were statistically analyzed using two-tailed Student's *t*-test. A difference was considered statistically significant when *$P < 0.05$ or **$P < 0.01$.

### Reporting summary
Further information on research design is available in the Nature Portfolio Reporting Summary linked to this article.

## Data availability
Raw RNA sequencing data are available at the NCBI Sequence Read Archive (SRA) under accession PRJNA1050330. Genes sequence data from this study can be accessed from the Rice Genome Annotation Project website under the following accession numbers: OsUBC12/LOC_Os05g38550, OsSnRK1.1/LOC_Os03g17980, OsABI5/LOC_Os01g64000, OsRAB21/LOC_Os11g26790, and OsUBQ5/LOC_Os01g22490. Source data are provided with this paper.

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

## Acknowledgements

We thank Dr He Gao (Max Planck Institute for plant breeding) for revising the manuscript. We thank researcher Yuanzhu Yang of Long Ping High-Tech for providing the Zhu 1S, Xiangling 628S, Longke 638S and Jing 4155S lines, researcher Qiyun Deng of State Key Laboratory of Hybrid Rice, the Hunan Hybrid Rice Research Center for providing the Y58S line, and researcher Dahu Ni of Rice Research Institute, Anhui Academy of Agricultural Sciences, for providing other two-line male sterile lines in this study. We also thank researcher Wensheng Wang of Chinese Academy of Agricultural Sciences for providing the varieties of the 69 cultivated accessions selected from nine rice subpopulations of the 3000 rice genomes. This work was supported by the Strategic Priority Research Program of the Chinese Academy of Sciences (XDA28100301), National Natural Science Foundation of China (U22A20456), Heilongjiang Provincial Natural Science Foundation of China (YQ2021C035), Chinese Postdoctoral Science Foundation (2021M693157), Young Scientist Group Project of Northeast Institute of Geography and Agroecology, Chinese Academy of Sciences (2023QNXZ02) and the Opening Foundation of Key Laboratory of Germplasm Enhancement, Physiology and Ecology of Food Crops in Cold Region, Ministry of Education, Northeast Agricultural University (CXSTOP2021006).

## Author contributions

J.F. and C.Z. designed the experiments. C.Z., X.T., H.W., J.F., and X.L. performed the experiments. C.Z., Y.H., Z.H., H.S., and J.L. analyzed the data. C.Z. wrote the manuscript. X.T., J.F., J.F.L., J.Z., and Q.B. revised the manuscript. All authors read and approved the final manuscript.

## Competing interests

The authors declare no competing interests.
