## [Peer Review File · Nature Communications]

A transposon insertion in the OsUBC12 promoter enhances cold tolerance during germination in japonica riceReviewers' Comments:

Reviewer #1:

Remarks to the Author:

This is an excellent manuscript that provides a molecular-genetic explanation for the improved germination rates of japonica rice varieties at cold temperatures, compared to indica varieties that require higher germination temperatures. As such, this work reveals an important step in the domestication of rice varieties in colder agricultural regions. The results are of high quality and presented well. I have some minor comments.

1) 253: "Among the DEGs, OsUBC12 was significantly down-regulated in *osubc12* mutants compared with WT (Fig. 3b), supporting the reliability of the transcriptome data."

These mutations are predicted to impact OsUBC12 protein synthesis, not the OsUBC12 transcript (at least not in theory). Thus, the downregulation of OsUBC12 transcript in these mutants could be because OsUBC12 promotes OsUBC12 expression (probably indirectly). Is there anything known about ABA-regulation of this gene? If ABA represses OsUBC12 expression, then loss of OsUBC12 function may enhance this repression. These data are probably already available and should be incorporated or mentioned if possible.

2) Some rephrasing suggestions:

274: "Taken together, these findings indicate that knockdown of OsUBC12 enhances ABA signaling, and thus that knockdown of OsUBC12 enhances ABA signaling."

Suggestion: remove the part after the comma.

303: "In contrast, OsUBC12C92A-OE lines showed decreased insensitivity combining ABA sensitivity phenotypes (Fig. 4h, i) and germination inhibition rate analysis (Fig. 4j, k)."

Suggestion: In contrast, this reduction in ABA sensitivity was less pronounced in the OsUBC12C92A-OE lines (Fig. 4h, i, j and k).

321: "OsSnRK1.1-His was not detected in the control sample (GST protein alone) but was pulled down by OsUBC12-GST, suggesting that OsUBC12 directly interacts with OsSnRK1.1 (Fig. 5b)."

Suggestion to rephrase to: OsSnRK1.1-His was pulled down by OsUBC12-GST, but not by the GST-only control, suggesting that OsUBC12 directly interacts with OsSnRK1.1 (Fig. 5b).

356: "We also analyzed OsSnRK1.1 expression in *osubc12* knockout mutants and transgenic OE lines, and showed that the presence of the functional OsUBC12 allele does not significant affect OsSnRK1.1 transcript levels (Fig. S10a, b), indicating that the regulatory effects of OsUBC12 on OsSnRK1.1 likely occur mainly at the post-translational rather than the transcriptional level."

Suggestion to rephrase to: We also found that OsSnRK1.1 expression was not significantly altered in *osubc12* knockout mutants and transgenic OsUBC12-OE lines, indicating that the regulatory...

398: "These findings indicate that OsUBC12-regulated LTG may be associated with enhanced ABA signaling."

Suggestion to rephrase to: These findings indicate that OsUBC12-promoted LTG is associated with suppression of ABA signaling.

297: "In contrast, the germination of OsUBC12C92A-OE lines at 15 °C was more similar to that of WT

(Fig. 4b), whereas only a slight effect was observed at 30 °C (Fig. S5c)."

Suggestion: delete part after the comma.

3) Some phrasing errors:

281: "To determine whether OsUBC12 has ubiquitin-conjugating enzyme activity and its active site, we mutated its active-site cysteine residue to alanine (OsUBC12C92A) and performed an in vitro ubiquitin thioester formation assay."

Delete: ...and its active site...

348: "Notably, OsSnRK1.1 degradation was significantly inhibited the proteasome inhibitor MG132, irrespective of whether protein extracts from WT or osubc12 mutants were assayed (Fig. 6c, bottom panel)."

...inhibited by the proteasome inhibitor...

4) Throughout the manuscript the authors have used unnecessary cautionary phrasing. Some examples are copied below:

132: "Molecular and biochemical analyses showed that OsUBC12 may negatively regulate ABA signaling by promoting the proteasomal degradation of OsSnRK1.1, thus accelerating LTG."

...OsUBC12 negatively regulates ABA signaling....

243: "These results suggest that japonica OsUBC12 locus introgressions into indica might have been used to improve the low-temperature germinability of indica rice."

.... introgressions into indica have been used to....

375: "Taken together, our results suggest that OsUBC12 may negatively regulate ABA signaling by promoting the degradation of OsSnRK1.1, thus accelerating LTG."

...our results show that OsUBC12 negatively regulates...

Reviewer #2:

Remarks to the Author:

This study described the identification of an E2 ubiquitin-conjugating enzyme gene, OsUBC12, as causal gene for regulation low-temperature germination in rice, through QTL analysis with CSSLs derived from a crossing between indica and japonica rice. The authors further investigated the molecular and biochemical function of UBC12 and discovered that it promotes the ubiquitination and subsequent degradation of SnRK1.1 protein which is a positive regulator of the ABA signaling pathway. Overall, this study provided valuable information for the fields of plant biology and rice breeding. However, I still have several concerns as listed below.

I could not understand how the authors identified the difference in the transposon insertion between IR64 and Koshihikari. While it appears the insertion in the Koshihikari genome was anticipated through examination of widely accessible databases such as the Rice Genome Annotation Project, what of the deletion in the IR64 genome?

The authors claimed that a transposon insertion affected the germination traits. However, they did not provide any information about other polymorphisms in the promoter region of OsUBC12. They should

prove and deliberate the effect of the other polymorphisms on the expression of OsUBC12 and germination.

L190; Since, the germination rate of IR64 at 30 °C is close to 60 % as shown in Fig. S5a, the relative germination fold changes at 30 °C cannot exceed those at 15 °C under these experimental conditions. The authors should consider re-evaluating their claim or conducting further experimentation under different conditions.

It is interesting that the function of OsUBC12 is temperature dependent. However, there is no experiment and discussion about the mechanisms underlying the temperature-dependent regulation of OsUBC12. The authors should test whether temperature affects the activity of OsUBC12 and subsequently provide a thorough discussion of their findings.

L438; The authors claimed that "This finding is consistent with the notion that japonica is adapted to the low temperatures occurring at high latitudes and higher elevations, while indica is adapted to low-latitude regions". However, Fig. 2a showed that all of GJ-trp, whose distribution almost overlaps with that of indica varieties (Sweeney MT, et al. PLoS Genet, (2007); doi.org/10.1371/journal.pgen.0030133), have the transposon insertion in the OsUBC12 promoter, suggesting that this insertion is not related to adaptation to the low temperatures occurring at high latitudes. They should more precisely discuss the relationship between adaptation and this gene.

L383; Since they claimed that OsUBC12 preferentially functions at 15 °C, but performs only a fine-tuning at 30 °C, it is an overstatement to refer to it as a "master regulator of seed germination."

Reviewer #3:

Remarks to the Author:

This work presented an E2 ubiquitin-conjugating enzyme-encoding gene OsUBC12 can increase low-temperature germinability (LTG) in japonica, owing to increased expression of the gene caused by a transposon insertion in its promoter. The work also revealed the mechanism by which E2 ubiquitin-conjugating enzyme directly interacted with and promoted degradation of OsSnRK1.1 protein kinase that positively regulates ABA sensitivity, and authors proposed that the OsUBC12 gene has breeding values in LTG improvement for indica rice. In general, the work was finished with rather complete data including positional cloning, gene function verification, biochemical assays, and natural variation exploitation for application, which has been adopted as routine experiments for cloning an agriculturally important genes in crops. However, a few points may be strengthened to highlight the novelty and /or practical values of gene and its working mechanism.

1. It has been well known that ABA inhibits seed germination, and many core signaling components in the ABA signaling, including SnRK1.1, have been verified for their roles in seed germination, and numerous reports suggest that one of the most common 'brakes' that weaken the ABA signaling is degradation of the core signaling components by ubiquitin-mediated degradation pathways. For this case, authors need to highlight, and convince the community, what is new for an E2 ubiquitin ligase-mediated SnRK1.1 degradation in the ABA signaling (seed germination) regulation compared to our known or easily deductive knowledge.

2. The statement that the transposon-insertion variation has been selected in japonica rice for LTG is questionable. If it has been selected for LTG, how to explain that 1) the OsUBC12 gene also functions in promoting seed germination under normal/favorite temperature conditions; 2) the allele has been introgressed into several indica rice varieties (seems not an intentionally selection of LTG). Only a small portion of the cultivated and wild rice germplasms were checked for the variation, which cannot support the evolutionary conclusion. How about the tropical japonica rice, in which LTG seems not a selective trait?

3. Whether the genomic region carrying the japonica OsUBC12 locus (with the transposon insertion) has practical LTG breeding value needs more convincing supporting data. The fact that the allele has

been introgressed into the modern elite indica lines such as Y58S and J4155S is not a supporting evidence because it seemed to be an unintended introgression of the allele.

Reviewer #4:

Remarks to the Author:

Zhang et al. identified a transposon insertion in the OsUBC12 promoter that increases UBC12 expression and germination rate at low temperature in japonica rice compared to indica. They generated *ubc12* knockout mutants and overexpression (OE) lines and showed that UBC12 decreases ABA sensitivity and promotes low-temperature germination (LTG). They showed that UBC12 (E2 ligase) has *in vitro* ubiquitination activity and interacts with OsSnRK1.1 kinase. They conducted *in vitro* degradation assays and concluded that UBC12 destabilize SnRK1. SnRK1-OE lines show opposite ABA sensitivity and germination rates at low temperature compared to UBC12-OE lines. They suggest that OsUBC12 may negatively regulate ABA signaling by promoting the proteasomal degradation OsSnRK1.1, thus accelerating LTG.

The findings are interesting and point to a possible role of introgression of this genomic region into indica varieties to improve cold tolerance during germination. While the role of UBC12 in LTG was well characterized (although I have comments about their germination assays), the mechanism by which UBC12 promotes LTG by inhibiting ABA signaling through SnRK1 was not demonstrated. The authors have shown that mutants affected in UBC12 expression (*ubc12*) or UBC12-dependent ubiquitination (UBC12-OE C92A) have altered ABA sensitivity, and that UBC12-OE and SnRK1-OE lines have opposite ABA sensitivity. Although SnRK1 and UBC12 interact *in vitro* (*in vivo* data is necessary), the E3 ligase that may work with UBC12 to degrade SnRK1 was not identified. Hence, direct ubiquitination of SnRK1 through UBC12 can't be shown (see Fernandez et al, 2020, REF 65 as an example of E2-E3 involved in ABA signaling). Therefore, to demonstrate a negative role of UBC12 in SnRK1-dependent ABA signaling during LTG, genetic interaction/co-overexpression studies should be conducted *in vivo*. SnRK1 protein level are expected to be increased in *osubc12* and enhanced cold sensitivity of *osubc12* could be rescued by decreasing SnRK1 expression (for example, by generating *osubc12 ossnrk1* mutants). Also, decreased UBC12-OE sensitivity to cold could be rescued by increasing SnRK1 (for example by generating double UBC12-OE SnRK1-OE). The current data only show opposite regulation of UBC12-OE and SNRK1-OE lines by ABA and cold stress. Furthermore, only SnRK1 overexpression lines were characterized (no loss-of-function studies). Also Phosphorylation of UBC12 by SnRK1 is inconclusive (see comments below).

Major comments:

1) Germination rates are very much dependent on the growth conditions of the developing seeds, which affect seed dormancy. In this study, the IR64 line, which was used as the control in several experiments, was grown in a growth chamber, while F2 populations and overexpression lines were grown under natural conditions (not specified) in different provinces of China (see methods). Therefore, the germination rates can't be compared. In order to compare germination rates and seedling emergence, all genotypes must be grown under the same conditions and seeds must be harvested at the same time. Same comment for gene expression analysis (Fig. 1b).

Fig 1. Both indica and japonica should be used as controls in germination and gene expression studies: (b,d,e) These experiments should be repeated and include the control japonica Koshihikari genotype, while in (f) the indica var IR64 should be used as control. Furthermore, why was japonica cultivar KY131 used to generate *ubc12* crisper mutants, and not Koshihikari and SL2116/7? These experiments would demonstrate that the transposon insertion in UBC12 promoter is the cause of SL2116/7 cold tolerance.

2) L157: Five ORFs were present in the 40kb region, but only UBC12 was discussed. Which other ORFs

were present in the 40kb region? Were these regions analyzed for potential genomic differences with parent lines? Could these regions also be important for LTG?

3) Fig 2: a) This panel is not well explained. All genotypes analyzed should be listed in the legend; example, what is GJ-trp and which subspecies were analyzed? I assume there are 4 included in this group, they should be listed. (b) Do Y58S and JING4155S, which carry the introgressed transposon, show increased germination at 15C like SL2116/7 lines? (c) at least two osubc12 mutant alleles and quantification of mutant phenotype should be shown. Last, germination of the indica and japonica cultivars and lines shown in panels a and b, carrying or lacking the transposon, should be tested to confirm a link between the transposon and cold tolerance. Importantly, is UBC12 transcription regulated by cold and/or ABA?

4) RNAseq. No GO function analysis was performed. Only a few DEGs are shown in Fig 3 and only ABI5 is involved in ABA signaling, therefore it is difficult to make any conclusions. How many ABA related genes were among the DEGs? Please include GO function enrichment analysis and provide a list of DEGs. Fig 3b: I assume +0.4 and -4 refer to log₂ (please specify in legend); why wasn't a more stringent selection criteria applied for selection of DEGs?

5) Pro-pro interaction studies. Fig 5 and L325-27. In vivo interaction between SnRK1.1 and UBC12 should be demonstrated (example co-IP).

6) Phosphorylation and degradation assays. Fig 6 (a) SnRK1 requires activation of the T-loop by SnAK/GRIK kinases (Shen et al., 2009 Plant Phys; Han et al., 2020 Plant Comm) to increase its phosphorylation activity and phosphorylate targets, therefore it can't be concluded that SnRK1 does not phosphorylate UBC12. A positive control (ABI5) could be used to confirm that the phosphorylation assay is working. (Fig 6 c,d) These results are not convincing; please include quantification of three biological replicates. Furthermore, the results should be validated by in vivo experiments, for example by using stable lines coexpressing inducible UBC12 and OE-SnRK1.1, showing that induction of UBC12 triggers degradation/destabilization of SnRK1.

7) For the above reasons, the model shown in Fig S11 is not fully supported (UBC12-SnRK1 phosphorylation/ubiquitination)

8) Methodology is minimal and not enough details are provided for some sections. Plant material: what temperature was used in the growth chambers? Only IR64 was grown in growth chambers? Please provide detailed description of RNAseq, including how the seeds were imbibed/germinated prior to RNA extraction (and how parental plants were grown), RNA quantity/quality analysis, library preparation, RNAseq analysis, DEG analysis, etc. List of DEGs etc should be provided.

9) Genetic background: different genetic backgrounds were used in this study. All figures should note (at least in the legend) which genotype was used as "WT". Example: Which genotype were OE constructs transformed into? Furthermore, why were CRISPR/Cas9 mutants generated in the Kongyun japonica background, and not in the Koshihikari japonica?

10) Statistical analysis should be shown in all graphs (main and supplem figures).

11) Figure legends are minimal or not shown for some supplementary data. Please add or include more details

MINOR

CRISPR/Cas9 mutants: was cas9 removed from these lines?

Fig S8: the conserved ubiquitin domain should be shown by sequence alignment with known and

characterized UBC proteins. As is presented, this figure only shows the presence of a cysteine residue

L114-122: REF 46-49 these are reviews, the original research papers should be cited.

L122-124: other papers link SnRK1 to ABA signaling, only one is cited here.

L383: What makes UBC12 a "master regulator" of seed germination?

REVIEWER COMMENTS

Reviewer #1 (Remarks to the Author):

This is an excellent manuscript that provides a molecular-genetic explanation for the improved germination rates of *japonica* rice varieties at cold temperatures, compared to *indica* varieties that require higher germination temperatures. As such, this work reveals an important step in the domestication of rice varieties in colder agricultural regions. The results are of high quality and presented well. I have some minor comments.

1) 253: “Among the DEGs, *OsUBC12* was significantly down-regulated in *osubc12* mutants compared with WT (Fig. 3b), supporting the reliability of the transcriptome data.”

These mutations are predicted to impact OsUBC12 protein synthesis, not the *OsUBC12* transcript (at least not in theory). Thus, the downregulation of *OsUBC12* transcript in these mutants could be because OsUBC12 promotes *OsUBC12* expression (probably indirectly). Is there anything known about ABA-regulation of this gene? If ABA represses *OsUBC12* expression, then loss of OsUBC12 function may enhance this repression. These data are probably already available and should be incorporated or mentioned if possible.

2) Some rephrasing suggestions:

274: “Taken together, these findings indicate that knockdown of *OsUBC12* enhances ABA signaling, and thus that knockdown of *OsUBC12* enhances ABA signaling.”

Suggestion: remove the part after the comma.

303: “In contrast, *OsUBC12^{C92A}-OE* lines showed decreased insensitivity combining ABA sensitivity phenotypes (Fig. 4h, i) and germination inhibition rate analysis (Fig. 4j, k).”

Suggestion: In contrast, this reduction in ABA sensitivity was less pronounced in the *OsUBC12^{C92A}-OE* lines (Fig. 4h, i, j and k).

321: “OsSnRK1.1-His was not detected in the control sample (GST protein alone) but was pulled down by OsUBC12-GST, suggesting that OsUBC12 directly interacts with OsSnRK1.1 (Fig. 5b).”

Suggestion to rephrase to: OsSnRK1.1-His was pulled down by OsUBC12-GST, but not by the GST-only control, suggesting that OsUBC12 directly interacts with OsSnRK1.1 (Fig. 5b).

356: “We also analyzed *OsSnRK1.1* expression in *osubc12* knockout mutants and transgenic OE lines, and showed that the presence of the functional *OsUBC12* allele does not significant affect *OsSnRK1.1* transcript levels (Fig. S10a, b), indicating that the regulatory effects of OsUBC12 on OsSnRK1.1 likely occur mainly at the

post-translational rather than the transcriptional level.”

Suggestion to rephrase to: We also found that OsSnRK1.1 expression was not significantly altered in *osubc12* knockout mutants and transgenic OsUBC12-OE lines, indicating that the regulatory...

398: “These findings indicate that OsUBC12-regulated LTG may be associated with enhanced ABA signaling.”

Suggestion to rephrase to: These findings indicate that OsUBC12-promoted LTG is associated with suppression of ABA signaling.

297: “In contrast, the germination of *OsUBC12^{C92A}-OE* lines at 15 °C was more similar to that of WT (Fig. 4b), whereas only a slight effect was observed at 30 °C (Fig. S5c).”

Suggestion: delete part after the comma.

3) Some phrasing errors:

281: “To determine whether OsUBC12 has ubiquitin-conjugating enzyme activity and its active site, we mutated its active-site cysteine residue to alanine (OsUBC12C92A) and performed an in vitro ubiquitin thioester formation assay.”

Delete: ...and its active site...

348: “Notably, OsSnRK1.1 degradation was significantly inhibited the proteasome inhibitor MG132, irrespective of whether protein extracts from WT or *osubc12* mutants were assayed (Fig. 6c, bottom panel).”

...inhibited by the proteasome inhibitor....

4) Throughout the manuscript the authors have used unnecessary cautionary phrasing. Some examples are copied below:

132: “Molecular and biochemical analyses showed that OsUBC12 may negatively regulate ABA signaling by promoting the proteasomal degradation of OsSnRK1.1, thus accelerating LTG.”

...OsUBC12 negatively regulates ABA signaling....

243: “These results suggest that japonica OsUBC12 locus introgressions into indica might have been used to improve the low-temperature germinability of indica rice.”

.... introgressions into indica have been used to....

375: “Taken together, our results suggest that OsUBC12 may negatively regulate ABA signaling by promoting the degradation of OsSnRK1.1, thus accelerating LTG.”

...our results show that OsUBC12 negatively regulates...

Reviewer #2 (Remarks to the Author):

This study described the identification of an E2 ubiquitin-conjugating enzyme gene, OsUBC12, as causal gene for regulation low-temperature germination in rice, through QTL analysis with CSSLs derived from a crossing between *indica* and *japonica* rice. The authors further investigated the molecular and biochemical function of UBC12 and discovered that it promotes the ubiquitination and subsequent degradation of SnRK1.1 protein which is a positive regulator of the ABA signaling pathway. Overall, this study provided valuable information for the fields of plant biology and rice breeding. However, I still have several concerns as listed below.

I could not understand how the authors identified the difference in the transposon insertion between IR64 and Koshihikari. While it appears the insertion in the Koshihikari genome was anticipated through examination of widely accessible databases such as the Rice Genome Annotation Project, what of the deletion in the IR64 genome?

The authors claimed that a transposon insertion affected the germination traits. However, they did not provide any information about other polymorphisms in the promoter region of *OsUBC12*. They should prove and deliberate the effect of the other polymorphisms on the expression of *OsUBC12* and germination.

L190; Since, the germination rate of IR64 at 30 °C is close to 60 % as shown in Fig. S5a, the relative germination fold changes at 30 °C cannot exceed those at 15 °C under these experimental conditions. The authors should consider re-evaluating their claim or conducting further experimentation under different conditions.

It is interesting that the function of OsUBC12 is temperature dependent. However, there is no experiment and discussion about the mechanisms underlying the temperature-dependent regulation of OsUBC12. The authors should test whether temperature affects the activity of OsUBC12 and subsequently provide a thorough discussion of their findings.

L438; The authors claimed that “This finding is consistent with the notion that *japonica* is adapted to the low temperatures occurring at high latitudes and higher elevations, while *indica* is adapted to low-latitude regions”. However, Fig. 2a showed that all of GJ-trp, whose distribution almost overlaps with that of *indica* varieties (Sweeney MT, et al. PLoS Genet, (2007); doi.org/10.1371/journal.pgen.0030133), have the transposon insertion in the *OsUBC12* promoter, suggesting that this insertion is not related to adaptation to the low temperatures occurring at high latitudes. They should more precisely discuss the relationship between adaptation and this gene.

L383; Since they claimed that OsUBC12 preferentially functions at 15 °C, but performs only a fine-tuning at 30 °C, it is an overstatement to refer to it as a "master regulator of seed germination."

Reviewer #3 (Remarks to the Author):

This work presented an E2 ubiquitin-conjugating enzyme-encoding gene *OsUBC12* can increase low-temperature germinability (LTG) in *japonica*, owing to increased expression of the gene caused by a transposon insertion in its promoter. The work also revealed the mechanism by which E2 ubiquitin-conjugating enzyme directly interacted with and promoted degradation of OsSnRK1.1 protein kinase that positively regulates ABA sensitivity, and authors proposed that the *OsUBC12* gene has breeding values in LTG improvement for *indica* rice. In general, the work was finished with rather complete data including positional cloning, gene function verification, biochemical assays, and natural variation exploitation for application, which has been adopted as routine experiments for cloning an agriculturally important genes in crops. However, a few points may be strengthened to highlight the novelty and /or practical values of gene and its working mechanism.

1. It has been well known that ABA inhibits seed germination, and many core signaling components in the ABA signaling, including SnRK1.1, have been verified for their roles in seed germination, and numerous reports suggest that one of the most common ‘brakes’ that weaken the ABA signaling is degradation of the core signaling components by ubiquitin-mediated degradation pathways. For this case, authors need to highlight, and convince the community, what is new for an E2 ubiquitin ligase-mediated SnRK1.1 degradation in the ABA signaling (seed germination) regulation compared to our known or easily deductive knowledge.

2. The statement that the transposon-insertion variation has been selected in *japonica* rice for LTG is questionable. If it has been selected for LTG, how to explain that 1) the *OsUBC12* gene also functions in promoting seed germination under normal/favorite temperature conditions; 2) the allele has been introgressed into several indica rice varieties (seems not an intentionally selection of LTG). Only a small portion of the cultivated and wild rice germplasms were checked for the variation, which cannot support the evolutionary conclusion. How about the tropical japonica rice, in which LTG seems not a selective trait?

3. Whether the genomic region carrying the *japonica OsUBC12* locus (with the transposon insertion) has practical LTG breeding value needs more convincing supporting data. The fact that the allele has been introgressed into the modern elite indica lines such as Y58S and J4155S is not a supporting evidence because it seemed to be an unintended introgression of the allele.

Reviewer #4 (Remarks to the Author):

Zhang et al. identified a transposon insertion in the *OsUBC12* promoter that increases *UBC12* expression and germination rate at low temperature in *japonica* rice compared to *indica*. They generated *ubc12* knockout mutants and overexpression (OE) lines and showed that UBC12 decreases ABA sensitivity and promotes low-temperature germination (LTG). They showed that UBC12 (E2 ligase) has *in vitro* ubiquitination activity and interacts with OsSnRK1.1 kinase. They conducted *in vitro* degradation assays and concluded that UBC12 destabilize SnRK1. *SnRK1-OE* lines show opposite ABA sensitivity and germination rates at low temperature compared to *UBC12-OE* lines. They suggest that OsUBC12 may negatively regulate ABA signaling by promoting the proteasomal degradation OsSnRK1.1, thus accelerating LTG.

The findings are interesting and point to a possible role of introgression of this genomic region into indica varieties to improve cold tolerance during germination. While the role of UBC12 in LTG was well characterized (although I have comments about their germination assays), the mechanism by which UBC12 promotes LTG by inhibiting ABA signaling through SnRK1 was not demonstrated. The authors have shown that mutants affected in *UBC12* expression (*ubc12*) or UBC12-dependent ubiquitination (UBC12-OE C92A) have altered ABA sensitivity, and that *UBC12-OE* and *SnRK1-OE* lines have opposite ABA sensitivity. Although SnRK1 and UBC12 interact *in vitro* (*in vivo* data is necessary), the E3 ligase that may work with UBC12 to degrade SnRK1 was not identified. Hence, direct ubiquitination of SnRK1 through UBC12 can't be shown (see Fernandez et al, 2020, REF 65 as an example of E2-E3 involved in ABA signaling). Therefore, to demonstrate a negative role of UBC12 in SnRK1-dependent ABA signaling during LTG, genetic interaction/co-overexpression studies should be conducted *in vivo*. SnRK1 protein level are expected to be increased in *osubc12* and enhanced cold sensitivity of *osubc12* could be rescued by decreasing SnRK1 expression (for example, by generating *osubc12 ossnrk1* mutants). Also, decreased *UBC12-OE* sensitivity to cold could be rescued by increasing SnRK1 (for example by generating double *UBC12-OE SnRK1-OE*). The current data only show opposite regulation of *UBC12-OE* and *SNRK1-OE* lines by ABA and cold stress. Furthermore, only *SnRK1* overexpression lines were characterized (no loss-of-function studies). Also Phosphorylation of UBC12 by SnRK1 is inconclusive (see comments below).

Major comments:

- 1) Germination rates are very much dependent on the growth conditions of the developing seeds, which affect seed dormancy. In this study, the IR64 line, which was used as the control in several experiments, was grown in a growth chamber, while F2 populations and overexpression lines were grown under natural conditions (not specified) in different provinces of China (see methods). Therefore, the germination rates can't be compared. In order to compare germination rates and seedling emergence, all genotypes must be grown under the same conditions and seeds must be

harvested at the same time. Same comment for gene expression analysis (Fig. 1b).

Fig 1. Both *indica* and *japonica* should be used as controls in germination and gene expression studies: (b,d,e) These experiments should be repeated and include the control *japonica* Koshihikari genotype, while in (f) the *indica* var IR64 should be used as control. Furthermore, why was *japonica* cultivar KY131 used to generate *ubc12* crispr mutants, and not Koshihikari and SL2116/7? These experiments would demonstrate that the transposon insertion in UBC12 promoter is the cause of SL2116/7 cold tolerance.

2) L157: Five ORFs were present in the 40kb region, but only *UBC12* was discussed. Which other ORFs were present in the 40kb region? Were these regions analyzed for potential genomic differences with parent lines? Could these regions also be important for LTG?

3) Fig 2: a) This panel is not well explained. All genotypes analyzed should be listed in the legend; example, what is GJ-trp and which subspecies were analyzed? I assume there are 4 included in this group, they should be listed. (b) Do Y58S and JING4155S, which carry the introgressed transposon, show increased germination at 15C like SL2116/7 lines? (c) at least two *osubc12* mutant alleles and quantification of mutant phenotype should be shown. Last, germination of the *indica* and *japonica* cultivars and lines shown in panels a and b, carrying or lacking the transposon, should be tested to confirm a link between the transposon and cold tolerance. Importantly, is *UBC12* transcription regulated by cold and/or ABA?

4) RNAseq. No GO function analysis was performed. Only a few DEGs are shown in Fig 3 and only *ABI5* is involved in ABA signaling, therefore it is difficult to make any conclusions. How many ABA related genes were among the DEGs? Please include GO function enrichment analysis and provide a list of DEGs. Fig 3b: I assume +0.4 and -4 refer to log2 (please specify in legend); why wasn't a more stringent selection criteria applied for selection of DEGs?

5) Pro-pro interaction studies. Fig 5 and L325-27. *In vivo* interaction between SnRK1.1 and UBC12 should be demonstrated (example co-IP).

6) Phosphorylation and degradation assays. Fig 6 (a) SnRK1 requires activation of the T-loop by SnAK/GRIK kinases (Shen et al., 2009 Plant Phys; Han et al., 2020 Plant Comm) to increase its phosphorylation activity and phosphorylate targets, therefore it can't be concluded that SnRK1 does not phosphorylate UBC12. A positive control (*ABI5*) could be used to confirm that the phosphorylation assay is working. (Fig 6 c,d) These results are not convincing; please include quantification of three biological replicates. Furthermore, the results should be validated by *in vivo* experiments, for example by using stable lines coexpressing inducible UBC12 and OE-SnRK1.1, showing that induction of UBC12 triggers degradation/destabilization of SnRK1.

7) For the above reasons, the model shown in Fig S11 is not fully supported (UBC12-SnRK1 phosphorylation/ubiquitination)

8) Methodology is minimal and not enough details are provided for some sections. Plant material: what temperature was used in the growth chambers? Only IR64 was grown in growth chambers? Please provide detailed description of RNAseq, including how the seeds were imbibed/germinated prior to RNA extraction (and how parental plants were grown), RNA quantity/quality analysis, library preparation, RNAseq analysis, DEG analysis, etc. List of DEGs etc should be provided.

9) Genetic background: different genetic backgrounds were used in this study. All figures should note (at least in the legend) which genotype was used as “WT”. Example: Which genotype were OE constructs transformed into? Furthermore, why were CRISPR/Cas9 mutants generated in the Kongyun *japonica* background, and not in the Koshihikari *japonica*?

10) Statistical analysis should be shown in all graphs (main and supplem figures).

11) Figure legends are minimal or not shown for some supplementary data. Please add or include more details

MINOR

CRISPR/Cas9 mutants: was cas9 removed from these lines?

Fig S8: the conserved ubiquitin domain should be shown by sequence alignment with known and characterized UBC proteins. As is presented, this figure only shows the presence of a cysteine residue

L114-122: REF 46-49 these are reviews, the original research papers should be cited.

L122-124: other papers link SnRK1 to ABA signaling, only one is cited here.

L383: What makes UBC12 a “master regulator” of seed germination?

Dear reviewers,

Re: Manuscript ID: NCOMMS-23-05556A

Title: A transposon insertion in the *OsUBC12* promoter enhances cold tolerance during germination in *japonica* rice (*Oryza sativa*)

We highly appreciate your insightful suggestions and comments, all of which are very helpful to the improvement of our manuscript. We have studied these comments carefully and have made correction which we hope meet with approval. All amendments are highlighted with colour highlighting in the revised version. Below, we have listed our point-by-point responses.

Thanks for your time and consideration again. And we are looking forward to your positive response.

Yours sincerely,

Jun Fang, Ph. D, Researcher

August 3, 2023

Response to reviewer's comments:

Reviewer #1 (Remarks to the Author):

This is an excellent manuscript that provides a molecular-genetic explanation for the improved germination rates of *japonica* rice varieties at cold temperatures, compared to *indica* varieties that require higher germination temperatures. As such, this work reveals an important step in the domestication of rice varieties in colder agricultural regions. The results are of high quality and presented well. I have some minor comments.

1) 253: “Among the DEGs, *OsUBC12* was significantly down-regulated in *osubc12* mutants compared with WT (Fig. 3b), supporting the reliability of the transcriptome data.”

These mutations are predicted to impact OsUBC12 protein synthesis, not the *OsUBC12* transcript (at least not in theory). Thus, the downregulation of *OsUBC12* transcript in these mutants could be because OsUBC12 promotes *OsUBC12* expression (probably indirectly). Is there anything known about ABA-regulation of this gene? If ABA represses *OsUBC12* expression, then loss of OsUBC12 function may enhance this repression. These data are probably already available and should be incorporated or mentioned if possible.

Response: Thanks for your meticulous reading and professional suggestion. We strongly agree with your viewpoint that the knockout mutations are predicted to impact OsUBC12 protein synthesis, not the OsUBC12 transcript. Therefore, we have re-conducted the RNA-seq of both WT and *osubc12* mutant, and found that *OsUBC12* was still significantly down-regulated in *osubc12* mutants compared with WT (Fig. S14). Moreover, the transcription levels of *OsUBC12* was significantly lower in *osubc12* mutants compared to WT by qRT-PCR analysis, consistent with the RNA-seq data (Fig. S15a). We have also investigated the expression kinetics of *OsUBC12* in response to ABA. As shown in Fig. 3a, the *OsUBC12* gene was up-regulated by ABA. The above results indicate that the knockout of *OsUBC12* can indeed lead to down-regulation of *UBC12* expression, but its mechanism may be more complex. We have amended the description as follows: “ Among the DEGs, *OsUBC12* was significantly down-regulated in *osubc12* mutants compared with WT (Fig. S14). The transcription levels of *OsUBC12* in *osubc12* mutants analyzed by

qRT-PCR also verified this points (Fig. S15a), suggesting that the knockout of *UBC12* in *osubc12* mutants may not only theoretically affect the synthesis of *UBC12* protein, but also reduce the transcription level of *UBC12*.”(as shown in the manuscript in Page 10, line 294-line 299)

2) Some rephrasing suggestions:

274: “Taken together, these findings indicate that knockdown of *OsUBC12* enhances ABA signaling, and thus that knockdown of *OsUBC12* enhances ABA signaling.”

Suggestion: remove the part after the comma.

Response: Thanks for your suggestion. We have removed the part after the comma in the manuscript in Page 10, line 316-line 318.

303: “In contrast, *OsUBC12^{C92A}-OE* lines showed decreased insensitivity combining ABA sensitivity phenotypes (Fig. 4h, i) and germination inhibition rate analysis (Fig. 4j, k).”

Suggestion: In contrast, this reduction in ABA sensitivity was less pronounced in the *OsUBC12^{C92A}-OE* lines (Fig. 4h, i, j and k).

Response: Thanks for your suggestion. This sentence has been modified to “In contrast, this reduction in ABA sensitivity was less pronounced in the *OsUBC12^{C92A}-OE* lines (Fig. 4h, i, j and k)” in the manuscript in Page 11, line 345-line 347.

321: “OsSnRK1.1-His was not detected in the control sample (GST protein alone) but was pulled down by OsUBC12-GST, suggesting that OsUBC12 directly interacts with OsSnRK1.1 (Fig. 5b).”

Suggestion to rephrase to: OsSnRK1.1-His was pulled down by OsUBC12-GST, but not by the GST-only control, suggesting that OsUBC12 directly interacts with OsSnRK1.1 (Fig. 5b).

Response: Thanks for your suggestion. This sentence has been modified to “OsSnRK1.1-His was pulled down by OsUBC12-GST, but not by the GST-only control, suggesting that OsUBC12 directly interacts with OsSnRK1.1 (Fig. 5b)” in the manuscript in Page 12, line 362-line 364.

356: “We also analyzed *OsSnRK1.1* expression in *osubc12* knockout mutants and transgenic OE lines, and showed that the presence of the functional *OsUBC12* allele does not significantly affect *OsSnRK1.1* transcript levels (Fig. S10a, b), indicating that the regulatory effects of OsUBC12 on OsSnRK1.1 likely occur mainly at the post-translational rather than the transcriptional level.”

Suggestion to rephrase to: We also found that OsSnRK1.1 expression was not significantly altered in *osubc12* knockout mutants and transgenic *OsUBC12-OE* lines, indicating that the regulatory...

Response: Thanks for your suggestion. This sentence has been modified to “We also found that OsSnRK1.1 expression was not significantly altered in *osubc12* knockout mutants and transgenic OsUBC12-OE lines (Fig. 6f, g), indicating that the regulatory effects of OsUBC12 on OsSnRK1.1 likely occur mainly at the post-translational rather than the transcriptional level.” in the manuscript in **Page 13, line 413-line 416**.

398: “These findings indicate that OsUBC12-regulated LTG may be associated with enhanced ABA signaling.”

Suggestion to rephrase to: These findings indicate that OsUBC12-promoted LTG is associated with suppression of ABA signaling.

Response: Thanks for your suggestion. This sentence has been modified to “These findings indicate that OsUBC12-promoted LTG is associated with suppression of ABA signaling.” in the manuscript in **Page 15, line 472-line 474**.

297: “In contrast, the germination of *OsUBC12^{C92A}-OE* lines at 15 °C was more similar to that of WT (Fig. 4b), whereas only a slight effect was observed at 30 °C (Fig. S5c).”

Suggestion: delete part after the comma.

Response: Thanks for your suggestion. We have deleted the part after the comma in the manuscript in **Page 11, line 340-line 341**.

3) Some phrasing errors:

281: “To determine whether OsUBC12 has ubiquitin-conjugating enzyme activity and its active site, we mutated its active-site cysteine residue to alanine (OsUBC12C92A) and performed an in vitro ubiquitin thioester formation assay.”

Delete: ...and its active site...

Response: Thanks for your suggestion. We have deleted the four words (and its active site) in the manuscript in **Page 11, line 325**.

348: “Notably, OsSnRK1.1 degradation was significantly inhibited the proteasome inhibitor MG132, irrespective of whether protein extracts from WT or *osubc12* mutants were assayed (Fig. 6c, bottom panel).”

...inhibited by the proteasome inhibitor....

Response: Thanks for your meticulous reading. This sentence has been amended to “Notably, OsSnRK1.1 degradation was significantly inhibited by the proteasome inhibitor MG132, irrespective of whether protein extracts from WT or *osubc12* mutants were assayed (Fig. 6c).” in the manuscript in **Page 13, line 402-line 404**.

4) Throughout the manuscript the authors have used unnecessary cautionary phrasing. Some examples are copied below:

132: “Molecular and biochemical analyses showed that OsUBC12 may negatively regulate ABA signaling by promoting the proteasomal degradation of OsSnRK1.1, thus accelerating LTG.”

...OsUBC12 negatively regulates ABA signaling....

Response: Thanks for your suggestion. We have removed the unnecessary cautionary phrasing in this sentence, and amended the description as follows: “Molecular and biochemical analyses showed that OsUBC12 negatively regulate ABA signaling by promoting the proteasomal degradation of OsSnRK1.1, thus accelerating LTG.” (as shown in the manuscript in **Page 5, line 133-line 135**). In addition, we have also checked and corrected other issues which are similar to the mistakes.

243: “These results suggest that japonica OsUBC12 locus introgressions into indica might have been used to improve the low-temperature germinability of indica rice.”

.... introgressions into indica have been used to....

Response: Thanks for your suggestion. We have removed the unnecessary cautionary phrasing in this sentence, and amended the description as follows: “These results suggest that *japonica OsUBC12* locus introgressions into *indica* have been used to

improve the low-temperature germinability of *indica* rice.” (as shown in the manuscript in Page 9, line 277-line 279).

375: “Taken together, our results suggest that OsUBC12 may negatively regulate ABA signaling by promoting the degradation of OsSnRK1.1, thus accelerating LTG.”

...our results show that OsUBC12 negatively regulates...

Response: Thanks for your suggestion. We have removed the unnecessary cautionary phrasing in this sentence, and amended the description as follows: “Taken together, our results suggest that OsUBC12 negatively regulate ABA signaling by promoting the degradation of OsSnRK1.1, thus accelerating LTG.” (as shown in the manuscript in Page 14, line 444-line 446). In addition, we have also checked and corrected other issues which are similar to the mistakes, such as the Page 2, line 41 and Page 14, line 430 of our revised manuscript.

Reviewer #2 (Remarks to the Author):

This study described the identification of an E2 ubiquitin-conjugating enzyme gene, OsUBC12, as causal gene for regulation low-temperature germination in rice, through QTL analysis with CSSLs derived from a crossing between *indica* and *japonica* rice. The authors further investigated the molecular and biochemical function of UBC12 and discovered that it promotes the ubiquitination and subsequent degradation of SnRK1.1 protein which is a positive regulator of the ABA signaling pathway. Overall, this study provided valuable information for the fields of plant biology and rice breeding. However, I still have several concerns as listed below.

I could not understand how the authors identified the difference in the transposon insertion between IR64 and Koshihikari. While it appears the insertion in the Koshihikari genome was anticipated through examination of widely accessible databases such as the Rice Genome Annotation Project, what of the deletion in the IR64 genome?

Response: Thanks for your meticulous reading and elaborate comments. Fine mapping was carried out for a cross between SL2117 and IR64, and the candidate gene was narrowed down to an ~40-kb region between markers M5-22.59 and M5-22.63. We used the genome sequence of Nipponbare (from Rice Genome Annotation Project Database, <http://rice.uga.edu/>) as a reference, combined with pan-genome data of 33 genetically diverse rice accessions (including Koshihikari and IR64, from Rice Resource Center Database, <http://ricerc.sicau.edu.cn/>) (Qin et al. 2021, Cell, <https://doi.org/10.1016/j.cell.2021.04.046>), and found that the locus *LOC_Os05g38550*, encoding an E2 ubiquitin-conjugating enzyme OsUBC12, is localized to this region, and the -542 site of its promoter was inserted a transposon (*LOC_Os05g38540*; about 6 kb) in *japonica* Koshihikari background compared to that in *indica* IR64 (Fig. 1a, bottom panel). In addition, PCR and its product sequencing results also verified the insertion of the transposon. We have added the related descriptions to the manuscript in **Page 6, line 168-line 178**.

The authors claimed that a transposon insertion affected the germination traits. However, they did not provide any information about other polymorphisms in the promoter region of *OsUBC12*. They should prove and deliberate the effect of the other polymorphisms on the expression of *OsUBC12* and germination.

Response: Thanks for your professional comments. Moreover, besides the presence or absence of the transposon, we also analyzed other polymorphisms in the promoter region of *OsUBC12* (Fig. S7), and performed a dual-luciferase reporter assay. We cloned *pOsUBC12^{IR64}* (2 kb) or *pOsUBC12^{Koshihikari}* (7.9 kb, containing the inserted transposon) to drive firefly luciferase, and used Renilla luciferase as an internal reference for transfection efficiency (Fig. S8). We also constructed *pOsUBC12^{Koshihikari}ΔTransposon* (*pOsUBC12^{Koshihikari}* but lacking the transposon, the transposon was artificially removed) and *pOsUBC12^{IR64+Transposon}* (*pOsUBC12^{IR64}* but containing the transposon, the transposon was artificially inserted into the same position as the *pOsUBC12^{Koshihikari}*) together to directly further elucidate whether the transposon insertion changes the transcriptional activity of the *OsUBC12* promoter rather than other polymorphisms (Fig. S8). Compared with the rice protoplasts transfected with *p35S:REN-pOsUBC12^{IR64}:LUC*, those transfected with *p35S:REN-pOsUBC12^{Koshihikari}:LUC* and *pOsUBC12^{IR64+Transposon}* displayed significantly increased promoter activity (greater LUC/REN ratios), and the promoter activity of *pOsUBC12^{Koshihikari}ΔTransposon* has no significant difference only because of the loss of the transposon (Fig. 1c). These results confirm that the insertion of the transposon in *pOsUBC12^{Koshihikari}* could up-regulate the expression of *OsUBC12*, while other polymorphisms in the promoter region of *OsUBC12* do not directly affect the expression. We have added the descriptions to the manuscript in **Page 7, line 187-line 206**.

L190; Since, the germination rate of IR64 at 30 °C is close to 60 % as shown in Fig. S5a, the relative germination fold changes at 30 °C cannot exceed those at 15 °C under these experimental conditions. The authors should consider re-evaluating their claim or conducting further experimentation under different conditions.

Response: Thanks for your meticulous reading and professional comments. We have removed the original Fig.1e (The relative germination fold changes of SL2116 and SL2117 after 60 h at 15 °C or 24 h at 30 °C). The claim-“with higher relative germination fold changes at 15 °C (Fig. 1e)” have also been removed.

It is interesting that the function of *OsUBC12* is temperature dependent. However, there is no experiment and discussion about the mechanisms underlying the

temperature-dependent regulation of OsUBC12. The authors should test whether temperature affects the activity of OsUBC12 and subsequently provide a thorough discussion of their findings.

Response: Thanks for your professional comments. Considering the temperature-dependent function of OsUBC12 and its biochemical properties as an E2 ubiquitin-conjugating enzyme, we analyzed the effect of low temperature on the post-translational level of UBC12. The results showed that low temperature could promote the accumulation of OsUBC12 protein *in vitro* and *in vivo* (Fig. S22a, b). At the transcriptional level, *OsUBC12* is induced by low temperature. (Fig. 3a). These observations further indicate the priority regulatory mechanism of UBC12 for low-temperature germination. We have reorganized and added these description for discussion. (as shown in the manuscript in Page 15, line 464-line 470).

L438; The authors claimed that “This finding is consistent with the notion that *japonica* is adapted to the low temperatures occurring at high latitudes and higher elevations, while *indica* is adapted to low-latitude regions”. However, Fig. 2a showed that all of GJ-trp, whose distribution almost overlaps with that of *indica* varieties (Sweeney MT, et al. PLoS Genet, (2007); doi.org/10.1371/journal.pgen.0030133), have the transposon insertion in the *OsUBC12* promoter, suggesting that this insertion is not related to adaptation to the low temperatures occurring at high latitudes. They should more precisely discuss the relationship between adaptation and this gene.

Response: Thanks for pointing this out. *Japonica* varieties has a wide geographic distribution globally, including both tropical and more temperate regions. Specifically, *temperate japonica* varieties tend to be found in temperate regions, *tropical japonica* varieties are most often found in tropical regions. Our analysis show that the transposon insertion conferring low-temperate germinability trait is found in high frequency in both *tropical* and *temperate japonica*, but not other subgroups. The result indicates that the insertion occurred in the common ancestor of *tropical* and *temperate japonica*, and may have facilitated the adaptation to temperate climate in *temperate japonica*. Now we have cited the literature and revised our manuscript to reflect these thoughts as follows:

“Natural variation analysis revealed that transposon insertion in the *OsUBC12* promoter mainly occurred in the *japonica* varieties which are planted in both tropical and cooler temperate regions globally⁵⁷, and that the transposon is absent in

wild and *indica* accessions which are mostly cultivated in tropical regions (Fig. 2a). Indeed, the *japonica* subpopulation in general has a greater low-temperature germinability than *indica*, and most alleles with increased low-temperature germinability were found in *japonica*^{4,13,14,78,79}. This finding is consistent with the notion that the *japonica* allele with the transposon insertion conferring low-temperature germinability may have facilitated *temperate japonica* varieties to adapt to cooler climate¹¹, and the *japonica* allele could be useful to enhance the low-temperature germinability trait of *indica*.” (as shown in the manuscript in **Page 16, line 513-line 523**).

L383; Since they claimed that OsUBC12 preferentially functions at 15 °C, but performs only a fine-tuning at 30 °C, it is an overstatement to refer to it as a "master regulator of seed germination."

Response: Thanks for your professional comments. The word-“master” has been removed in the manuscript in **Page 14, line 451**.

Reviewer #3 (Remarks to the Author):

This work presented an E2 ubiquitin-conjugating enzyme-encoding gene *OsUBC12* can increase low-temperature germinability (LTG) in *japonica*, owing to increased expression of the gene caused by a transposon insertion in its promoter. The work also revealed the mechanism by which E2 ubiquitin-conjugating enzyme directly interacted with and promoted degradation of OsSnRK1.1 protein kinase that positively regulates ABA sensitivity, and authors proposed that the *OsUBC12* gene has breeding values in LTG improvement for *indica* rice. In general, the work was finished with rather complete data including positional cloning, gene function verification, biochemical assays, and natural variation exploitation for application, which has been adopted as routine experiments for cloning an agriculturally important genes in crops. However, a few points may be strengthened to highlight the novelty and /or practical values of gene and its working mechanism.

1. It has been well known that ABA inhibits seed germination, and many core signaling components in the ABA signaling, including SnRK1.1, have been verified for their roles in seed germination, and numerous reports suggest that one of the most common ‘brakes’ that weaken the ABA signaling is degradation of the core signaling components by ubiquitin-mediated degradation pathways. For this case, authors need to highlight, and convince the community, what is new for an E2 ubiquitin ligase-mediated SnRK1.1 degradation in the ABA signaling (seed germination) regulation compared to our known or easily deductive knowledge.

Response: Thanks for your professional comments. We believe our findings are interesting and significant and the innovative content of this paper mainly lay two points:

The first significant finding is the functionary mechanism of *OsUBC12* in low-temperature germination. Seed germination are complex adaptive traits of higher plants that are influenced by a large number of genes and environmental factors. Although studies of genetics and physiology have shown the important roles of the plant hormones abscisic acid (ABA) in the regulation of germination, and one of the most common ‘brakes’ that weaken the ABA signaling is degradation of the core signaling components by ubiquitin-mediated degradation pathways. However, the function and mechanism of E2 ubiquitin-conjugating enzymes, the key enzyme of ubiquitin-proteasome system (UPS), in regulating LTG through ABA signaling have not been reported yet. We found that *OsUBC12* negatively regulate ABA signaling by

promoting the degradation of OsSnRK1.1, thus accelerating LTG. Furthermore, we clarified a novel model to explain how OsUBC12 regulates LTG (Fig. S11). According to our model, OsSnRK1.1 functions as a downstream key regulator to enhance ABA responses by upregulating the expression of ABA-signaling-related genes such as *OsABI5* and *OsRAB21*, thus inhibiting LTG. OsUBC12, an E2 enzyme for K48-linked polyubiquitination, recruits and degrades OsSnRK1.1. Compared with *indica* rice, a transposon insertion in the *japonica* *OsUBC12* promoter activates its expression. Increased OsUBC12 levels further promote the degradation of OsSnRK1.1, thereby weakening OsSnRK1.1-regulated ABA signaling and enhancing the low-temperature germinability of *japonica* rice.

The second significance is that we found a loci in *japonica* rice can be used to improve the low-temperature germination ability of *indica* rice. Optimal rice (*Oryza sativa*) germination temperature ranges from 25 °C to 35 °C; temperatures below 17 °C cause cold stress, with low germination rates, germination delay, retarded growth and seedling mortality. Moreover, low-temperature germinability is a prerequisite for modern direct-seeding cultivation, an alternative to conventional transplanting that effectively reduces rice production costs. In general, *Japonica* rice has greater capacity for germination at low temperatures than the *indica* subpopulation. Hybridization between different rice groups has been a common practice in rice breeding, and genomic loci conferring favorable agronomic traits are often introgressed across different rice subgroups. Which loci in *japonica* rice can be used to improve the low-temperature germination ability of *indica* rice? Have they been used in modern breeding? These problems may be of interest to the experts in *Nature Communications* readers across several disciplines including of plant molecular biology and agriculture. Here, we report that *OsUBC12*, encoding an E2 ubiquitin-conjugating enzyme, increases low-temperature germinability in *japonica*, owing to a transposon insertion in its promoter enhances its expression. Natural variation analysis revealed transposon insertion in the *OsUBC12* promoter mainly in the *japonica* lineage, and the transposon is absent in wild and *indica* accessions. Notably, the genomic region carrying the *japonica* *OsUBC12* locus (with the transposon insertion) has been introgressed into the modern elite *indica* two-line male sterile lines Y58S and J4155S, from Nongken 58S, Zhu 1S, Peiai 64S, Guangzhan 63S, Xiangling 628S, Longke 638S, indicating that this *japonica*-derived locus (harboring the transposon insertion) has been introgressed

into modern cultivars of an *indica* genomic background. We also investigated the low-temperature germinability of 50 varieties without transposon insertion and 21 varieties with transposon insertion, which from the majority of the 69 cultivated accessions selected to represent nine rice subpopulations from the 3,000 rice genomes as well as all varieties and sterile lines in Fig2b, confirming that varieties carrying the *japonica* *OsUBC12* locus (transposon insertion) have higher low-temperature germinability than varieties without the locus (Fig. 2c). In brief, our findings provide insight for researchers and breeders about the underlying mechanisms of UBC12 regulating LTG and provide genetic reference points for improving LTG in *indica* rice.

2. The statement that the transposon-insertion variation has been selected in *japonica* rice for LTG is questionable. If it has been selected for LTG, how to explain that 1) the *OsUBC12* gene also functions in promoting seed germination under normal/favorite temperature conditions; 2) the allele has been introgressed into several indica rice varieties (seems not an intentionally selection of LTG). Only a small portion of the cultivated and wild rice germplasms were checked for the variation, which cannot support the evolutionary conclusion. How about the tropical *japonica* rice, in which LTG seems not a selective trait?

Response: 1) Thanks for your meticulous reading and elaborate comments. We agree that *OsUBC12* allele with the transposon-insertion is not necessarily selected in *japonica* for LTG, but we argue that this particular allele may have contributed to the adaptation to temperate climate in *japonica*. We have revised our manuscript accordingly to reflect this. *OsUBC12* promotes seed germination under both normal and cold conditions, which is consistent with its potential role in facilitating the temperate *japonica* to adapt colder environment.

Response: 2) *Japonica* varieties has a wide geographic distribution globally, including both tropical and more temperate regions. Specifically, *temperate japonica* varieties tend to be found in temperate regions, *tropical japonica* varieties are most often found in tropical regions. Our analysis show that the transposon insertion conferring low-temperate germinability trait is found in high frequency in both *tropical* and *temperate japonica*, but not other subgroups. The result indicates that the insertion occurred in the common ancestor of *tropical* and *temperate japonica*, and the low-temperature germinability trait conferred by the insertion may have facilitated

the adaptation to temperate climate in *temperate japonica*. We have revised our manuscript to reflect this. (as shown in the manuscript in Page 16, line 513-line 523).

The cultivated rice population we used in our analysis were selected by another independent research group to represent the genetic diversity of the 3000-rice population, and it covers all subgroups in cultivated rice. Therefore, we consider the panel is highly representative of rice genetic diversity and have adequate power to clarify the distribution of the transposon insertion on *OsUBC12* in different subgroups. Moreover, we have improved and re-analyzed the natural variation in *pOsUBC12* among rice populations. Previously, we analyzed 75 individuals and only 62 individuals information were obtained. Now, all 75 individuals information have been obtained (as shown in Fig. 2a and the manuscript in Page 8, line 233-line 236).

Nevertheless, we also extended our analyses using the high quality rice genome assembly dataset published in Shang et al. 2022 Cell Res. We called the presence absence variation of the transposon in 197 Asian domesticated rice accessions with subgroup information. The result is consistent with the result obtained with the dataset from Zhang et al. 2022: the transposon is enriched in japonica rice (28/58), and absent in aus group, and only found in one accession of 135 accessions of *indica* variety (as shown in Fig. S12 and the manuscript in Page 8, line 238-line 244).

3. Whether the genomic region carrying the *japonica OsUBC12* locus (with the transposon insertion) has practical LTG breeding value needs more convincing supporting data. The fact that the allele has been introgressed into the modern elite *indica* lines such as Y58S and J4155S is not a supporting evidence because it seemed to be an unintended introgression of the allele.

Response: Thanks for your professional comments. To analyze the relationship between the *japonica OsUBC12* locus (transposon insertion) and LTG, we investigated the low-temperature germinability of 50 varieties without transposon insertion and 21 varieties with transposon insertion, which from the majority of the 69 cultivated accessions selected to represent nine rice subpopulations from the 3,000 rice genomes as well as all varieties and sterile lines in Fig2b, confirming that varieties carrying the *japonica OsUBC12* locus (transposon insertion) have higher low-temperature germinability than varieties without the locus (Fig. 2c). (as shown in Fig 2c and the manuscript in Page 9, line 266-line 273).

Reviewer #4 (Remarks to the Author):

Zhang et al. identified a transposon insertion in the *OsUBC12* promoter that increases *UBC12* expression and germination rate at low temperature in *japonica* rice compared to *indica*. They generated *ubc12* knockout mutants and overexpression (OE) lines and showed that UBC12 decreases ABA sensitivity and promotes low-temperature germination (LTG). They showed that UBC12 (E2 ligase) has *in vitro* ubiquitination activity and interacts with OsSnRK1.1 kinase. They conducted *in vitro* degradation assays and concluded that UBC12 destabilize SnRK1. *SnRK1-OE* lines show opposite ABA sensitivity and germination rates at low temperature compared to *UBC12-OE* lines. They suggest that OsUBC12 may negatively regulate ABA signaling by promoting the proteasomal degradation OsSnRK1.1, thus accelerating LTG.

The findings are interesting and point to a possible role of introgression of this genomic region into *indica* varieties to improve cold tolerance during germination. While the role of UBC12 in LTG was well characterized (although I have comments about their germination assays), the mechanism by which UBC12 promotes LTG by inhibiting ABA signaling through SnRK1 was not demonstrated. The authors have shown that mutants affected in *UBC12* expression (*ubc12*) or UBC12-dependent ubiquitination (UBC12-OE C92A) have altered ABA sensitivity, and that *UBC12-OE* and *SnRK1-OE* lines have opposite ABA sensitivity. Although SnRK1 and UBC12 interact *in vitro* (*in vivo* data is necessary), the E3 ligase that may work with UBC12 to degrade SnRK1 was not identified. Hence, direct ubiquitination of SnRK1 through UBC12 can't be shown (see Fernandez et al, 2020, REF 65 as an example of E2-E3 involved in ABA signaling). Therefore, to demonstrate a negative role of UBC12 in SnRK1-dependent ABA signaling during LTG, genetic interaction/co-overexpression studies should be conducted *in vivo*. SnRK1 protein level are expected to be increased in *osubc12* and enhanced cold sensitivity of *osubc12* could be rescued by decreasing SnRK1 expression (for example, by generating *osubc12 ossnrk1* mutants). Also, decreased *UBC12-OE* sensitivity to cold could be rescued by increasing SnRK1 (for example by generating double *UBC12-OE SnRK1-OE*). The current data only show opposite regulation of *UBC12-OE* and *SNRK1-OE* lines by ABA and cold stress. Furthermore, only *SnRK1* overexpression lines were characterized (no loss-of-function studies). Also Phosphorylation of UBC12 by SnRK1 is inconclusive (see comments below).

Response: Thanks for your professional comments and insightful suggestions. **Firstly**, we are very sorry for our imprecise and non-exhaustive descriptions. In each germination experiment of our study, the control and experimental groups were grown under the same conditions, and both control and experimental groups seeds were collected at 45 days after flowering, air-dried, and stored at 45 °C for three days to break dormancy as described by as described by Yoshida *et al.*³⁷ and Fujino *et al.*³. The growth conditions and sampling times of the control lines involved in each experiment in this study, including gene expression analysis assay, were consistent with their related CSSLs, mutants, transgenic lines or varieties. We have added the related descriptions to **the Methods section of manuscript in Page 18, line 575-line 581**. **Moreover**, we have performed the Co-IP assays to validate the interaction between OsUBC12 and OsSnRK1.1 *in vivo*. Co-IP assays indicated that OsSnRK1.1-Myc fusion proteins could be immunoprecipitated with OsUBC12-Flag in *N. benthamiana* transiently expressing OsSnRK1.1-Myc and OsUBC12-Flag, while OsSnRK1.1-Myc could not be immunoprecipitation with negative control GFP-Flag (Fig. 5d), also confirmed the OsUBC12-OsSnRK1.1 interaction *in vivo*. Collectively, the four independent assays demonstrate that OsUBC12 directly interacts with OsSnRK1.1 both *in vitro* and *in vivo* (**as shown in the manuscript in Page 12, line 366-line 372**). *In vivo* degradation experiment in the rice protoplast system also has been supplemented and suggested that OsUBC12-Flag obviously promoted the degradation of OsSnRK1.1-Flag compared to empty-Flag (EV) and OsSnRK1.1-Flag co-expression, in contrast, this degradation was less pronounced in the presence of OsUBC12C92A-Flag (Fig. 6e). These results further confirm that OsUBC12 promotes OsSnRK1.1 degradation *in vivo*, and requires its active-site cysteine residue to do so (**as shown in the manuscript in Page 13, line 399-line 413**). To further reveal the functionality of OsUBC12 and the negative role of UBC12 in SnRK1-dependent ABA signaling during LTG, we performed the genetic interaction analysis by generating *OsSnRK1.1* homozygous knockout mutants (*ossnrk1.1-1*, *ossnrk1.1-2*, and *ossnrk1.1-3*) (Fig. S20) and *osubc12snrk1.1* double mutants (*osubc12snrk1.1-1*, *osubc12snrk1.1-2*, and *osubc12snrk1.1-3*) in the KY131 background (Fig. S21). Germination tests displayed that *ossnrk1.1* mutants showed clearly increased germination rates at low temperature (15° C) (Fig. 7a), but only a slight increment in germination at 30° C (Fig. S9f), and knockout of *OsSnRK1.1*

rescued the increased cold sensitive phenotype of *osubc12* during LTG (Fig. 7a). The genetic relationship between *OsUBC12* and *OsSnRK1.1* in ABA sensitivity was analyzed too. Similarly, knockout of *OsSnRK1.1* rescued the increased ABA sensitive phenotype of *osubc12* (Fig. 7b-g), indicating that OsSnRK1.1 acts downstream of OsUBC12 to regulate LTG and ABA responses. **Finally**, we have supplemented the phosphorylation assay according to your professional suggestion. The results displayed that OsSnRK1.1 and OsABI5 had auto-phosphorylation activity, and OsGRIK1 could further activate and phosphorylate OsSnRK1.1. In the presence of both OsGRIK1 and OsSnRK1.1, OsABI5 could be activated and phosphorylated by SnRK1, while OsUBC12 remained unable to be phosphorylated by OsSnRK1.1 (Fig. S18b) (as shown in the manuscript in Page 12, line 381-line 391).

Below, we have listed our point-by-point responses for your professional comments.

Major comments:

1) Germination rates are very much dependent on the growth conditions of the developing seeds, which affect seed dormancy. In this study, the IR64 line, which was used as the control in several experiments, was grown in a growth chamber, while F2 populations and overexpression lines were grown under natural conditions (not specified) in different provinces of China (see methods). Therefore, the germination rates can't be compared. In order to compare germination rates and seedling emergence, all genotypes must be grown under the same conditions and seeds must be harvested at the same time. Same comment for gene expression analysis (Fig. 1b).

Fig 1. Both *indica* and *japonica* should be used as controls in germination and gene expression studies: (b,d,e) These experiments should be repeated and include the control *japonica* Koshihikari genotype, while in (f) the *indica* var IR64 should be used as control. Furthermore, why was *japonica* cultivar KY131 used to generate *ubc12* crispr mutants, and not Koshihikari and SL2116/7? These experiments would demonstrate that the transposon insertion in UBC12 promoter is the cause of SL2116/7 cold tolerance.

Response: Thanks for your meticulous reading and elaborate comments. First, we are very sorry for our imprecise and non-exhaustive descriptions. In each germination

experiment of our study, the control and experimental groups were grown under the same conditions, and both control and experimental groups seeds were collected at 45 days after flowering, air-dried, and stored at 45 °C for three days to break dormancy as described by as described by Yoshida *et al.*³⁷ and Fujino *et al.*³. The growth conditions and sampling times of the control lines involved in each experiment in this study, including gene expression analysis assay, were consistent with their related CSSLs, mutants, transgenic lines or varieties. We have added the related descriptions to **the Methods section of manuscript in Page 18, line 575-line 581.**

Furthermore, we have re-conducted germination and gene expression assay using *indica* IR64 and *japonica* Koshihikari rice as controls in Fig 1b, d. We have also re-conducted germination assay using *indica* IR64 and *japonica* KY131 rice as controls in Fig 1f. KY131 is a high-quality rice with a large planting area in our cold *Japonica* rice area in Northeast China, and its genotype is consistent with Koshihikari, both of which carry the *japonica* *OsUBC12* locus (transposon insertion), so we used KY131 as the genetic background to generate *ubc12* crispr mutants for further experiments. At the same time, to provide more evidence about *japonica* *OsUBC12* locus and cold tolerance, we have also generated *OsUBC12* homozygous knockout mutants (*osubc12-ks1* and *osubc12-ks2*) using a clustered regularly interspaced short palindromic repeat (CRISPR)/CRISPR-associated nuclease 9 (Cas9) system in Koshihikari background (Fig. S10), and supplemented the germination of *osubc12-ks* at 15 ° C (Fig. 1e) and 30 ° C (Fig S9b). (as shown in the manuscript in Page 7, line 208-line 218)

2) L157: Five ORFs were present in the 40kb region, but only *UBC12* was discussed. Which other ORFs were present in the 40kb region? Were these regions analyzed for potential genomic differences with parent lines? Could these regions also be important for LTG?

Response: Thanks for your meticulous reading. In addition to *UBC12* (LOC_Os05g38550) and the transposon gene (LOC_Os05g38540; about 6 kb) inserted at its promoter-542 site in *japonica* Koshihikari background compared to that in *indica* IR64 (Fig. 1a, bottom panel), there are three other genes in this region, including *LOC_Os05g38510*, *LOC_Os05g38520* and *LOC_Os05g38530*. We have supplemented the analysis of potential potential genomic differences in these three

genes between parent lines. Of the remaining three genes, *LOC_Os05g38510* and *LOC_Os05g38530* have 98.19% and 99.58%, genomic homologies between Koshihikari and IR64, respectively, with nearly no much variation (Fig. S4, S5), whereas the variation in *LOC_Os05g38520* was all in the introns (Fig. S6). We have added the related descriptions to **the manuscript in Page 6, line 168-line 182**. Therefore, we hypothesized that the transposon insertion in *pOsUBC12^{Koshihikari}* may affect the expression level of *OsUBC12*, and thus cause the change of germinability. And the hypothesis was confirmed by a series of molecular and functional analysis.

3) Fig 2: a) This panel is not well explained. All genotypes analyzed should be listed in the legend; example, what is GJ-trp and which subspecies were analyzed? I assume there are 4 included in this group, they should be listed. (b) Do Y58S and JING4155S, which carry the introgressed transposon, show increased germination at 15C like SL2116/7 lines? (c) at least two *osubc12* mutant alleles and quantification of mutant phenotype should be shown. Last, germination of the *indica* and *japonica* cultivars and lines shown in panels a and b, carrying or lacking the transposon, should be tested to confirm a link between the transposon and cold tolerance. Importantly, is *UBC12* transcription regulated by cold and/or ABA?

Response: Fig 2: a) Thanks for your meticulous reading and elaborate comments. We have perfected the relevant figure captions (**as shown in Fig. 2a and the manuscript in Page 32, line 1085-line 1093**), and provided the list of varieties and their associated subgroup, haplotype information used in Fig. 2a (**as shown in Supplemental Table S2**).

Response: Fig 2: b) Because the varieties and sterile lines in 2b do not have a strict genetic background control lines like the chromosome segment substitution lines (CSSLs), in order to analyze the relationship between the *japonica OsUBC12* locus (transposon insertion) and LTG, we have investigated the low-temperature germinability of 50 varieties without transposon insertion and 21 varieties with transposon insertion, which from the majority of the 69 cultivated accessions selected to represent nine rice subpopulations from the 3,000 rice genomes as well as all varieties and sterile lines in Fig2b, confirming that varieties carrying the *japonica OsUBC12* locus (transposon insertion) have higher low-temperature germinability than varieties without the locus (Fig. 2c). (**as shown in the manuscript in Page 8, line 266-line 273**).

Response: Fig 2: c) we have re-conducted the simulated rice direct seeding experiment with three *osubc12* mutant alleles, and provided the emergence rate (as shown in Fig 2d).

Last, as mentioned above, to confirm a link between the transposon and cold tolerance, we have investigated the low-temperature germinability of 50 varieties without transposon insertion and 21 varieties with transposon insertion, which from the majority of the 69 cultivated accessions selected to represent nine rice subpopulations from the 3,000 rice genomes as well as all varieties and sterile lines in Fig2b, confirming that varieties carrying the *japonica OsUBC12* locus (transposon insertion) have higher low-temperature germinability than varieties without the locus (Fig. 2c). (as shown in the manuscript in Page 8, line 266-line 273).

Finally, we have investigated the expression kinetics of *OsUBC12* in response to cold and ABA. As shown in Fig. 3a, the transcription of *OsUBC12* was induced by both cold and ABA, implying that *OsUBC12* may be involved in responding to ABA and cold stress (as shown in the manuscript in Page 9, line 282-line 287).

4) RNAseq. No GO function analysis was performed. Only a few DEGs are shown in Fig 3 and only *ABI5* is involved in ABA signaling, therefore it is difficult to make any conclusions. How many ABA related genes were among the DEGs? Please include GO function enrichment analysis and provide a list of DEGs. Fig 3b: I assume +0.4 and -4 refer to log2 (please specify in legend); why wasn't a more stringent selection criteria applied for selection of DEGs?

Response: Thanks for your professional comments. We have performed the GO enrichment analysis of significantly differentially expressed genes in *osubc12* mutants vs. WT (KY131) seeds detected by RNA-seq analysis (as shown in Fig S13b). Moreover, we have provided the list of DEGs (as shown in Supplementary Data 1) and ABA related genes (as shown in Supplementary Data 1).

Response: Fig 3b: The color scale represents the log2 (FPKM+1) and then standardized by “scale()” in R script. We have re-conducted the Heat map analysis with a more stringent selection criteria (as shown in Fig S14).

To explore whether *OsUBC12* is involved in regulating ABA signaling, we have first analyzed the expression kinetics of *OsUBC12* in response to cold and ABA. As shown in Fig. 3a, the transcription of *OsUBC12* was induced by both cold and ABA,

implying that OsUBC12 may be involved in responding to ABA and cold stress. Then, we referenced the the results of RNA-seq. Many ABA-related genes showed differential expression between WT and *osubc12* mutants (Fig. S14, Supplementary Data 1). Notably, the expression of *OsABI5*, a core components of ABA signaling which also functions as a negative regulator of seed germination ⁶⁹, was up-regulated in *osubc12* mutants (Fig. S14, Supplementary Data 1). We then examined the expression of *OsABI5* as well as *OsRAB21* which is a commonly used marker gene for ABA response ⁷⁰ in WT and *osubc12* mutants by qRT-PCR analysis. Both *OsABI5* and *OsRAB21* expression was significantly higher in *osubc12* mutants compared to WT (Fig. 3b, c).

We also studied the effect of ABA on the germination of WT and *osubc12* mutant seeds by quantifying germination rates in response to 0, 1 or 2 μ M ABA (Fig. 3d-f). In the absence of ABA, the *osubc12* mutant lines exhibited delayed germination (Fig. 3d). Treatment with 1 or 2 μ M exogenous ABA produced a more pronounced inhibitory effect on *osubc12* mutants than on WT (Fig. 3e, f). To better quantify the ABA sensitivity of WT and *osubc12* mutants, we analyzed their ABA-mediated germination inhibition rates. After 1 or 2 μ M ABA treatment, the germination inhibition rate was significantly higher in *osubc12* mutants in WT, indicating that *OsUBC12* knockdown increases ABA sensitivity (Fig. 3g, h). Taken together, these findings indicate that knockdown of *OsUBC12* enhances ABA signaling.

5) Pro-pro interaction studies. Fig 5 and L325-27. *In vivo* interaction between SnRK1.1 and UBC12 should be demonstrated (example co-IP).

Response: Thanks for your professional comments. First, we have made several attempts to perform Co-IP assays in rice protoplasts, but only OsUBC12-Flag was expressed in protoplasts, whereas OsSnRK1.1-Myc could not be detected. Therefore, we conducted Co-IP assays using *N. benthamiana* system. OsUBC12-Flag and OsSnRK1.1-Myc can both be expressed and detected, but OsSnRK1.1-Myc has two bands detected in tobacco expression (as shown in Response Fig 1), and it was hypothesized that OsSnRK1.1 might contain phosphorylated forms or be affected by tobacco homologous genes in tobacco. Co-IP assays indicated that OsSnRK1.1-Myc fusion proteins could be immunoprecipitated with OsUBC12-Flag in *N. benthamiana* transiently expressing OsSnRK1.1-Myc and OsUBC12-Flag, while

OsSnRK1.1-Myc could not be immunoprecipitation with negative control GFP-Flag (Fig. 5d), also confirmed the OsUBC12-OsSnRK1.1 interaction *in vivo*. Collectively, the four independent assays demonstrate that OsUBC12 directly interacts with OsSnRK1.1 both *in vitro* and *in vivo* (as shown in the manuscript in Page 12, line 366-line 372).

Response Fig 1:

M: Thermo PageRuler Prestained Protein Ladder (10-180KD); H₂O: Protein extracts of *N. benthamiana* transiently expressing H₂O; 1-10: Protein extracts of *N. benthamiana* transiently expressing OsSnRK1.1-Myc

Response Fig 1: The expression of OsSnRK1.1-Myc in in *N. benthamiana*

6) Phosphorylation and degradation assays. Fig 6 (a) SnRK1 requires activation of the T-loop by SnAK/GRIK kinases (Shen et al., 2009 Plant Phys; Han et al., 2020 Plant Comm) to increase its phosphorylation activity and phosphorylate targets, therefore it can't be concluded that SnRK1 does not phosphorylate UBC12. A positive control (ABI5) could be used to confirm that the phosphorylation assay is working. (Fig 6 c,d) These results are not convincing; please include quantification of three biological replicates. Furthermore, the results should be validated by *in vivo* experiments, for example by using stable lines coexpressing inducible UBC12 and OE-SnRK1.1, showing that induction of UBC12 triggers degradation/destabilization of SnRK1.

Response: Thanks for your professional comments.

Response: Fig 6 (a) Phosphorylation assays: we have supplemented the phosphorylation assay according to your professional suggestion. The results displayed that OsSnRK1.1 and OsABI5 had auto-phosphorylation activity, and OsGRIK1 could further activate and phosphorylate OsSnRK1.1. In the presence of both OsGRIK1 and OsSnRK1.1, OsABI5 could be activated and phosphorylated by SnRK1, while OsUBC12 remained unable to be phosphorylated by OsSnRK1.1 (Fig.

S18b) (as shown in the manuscript in Page 12, line 381-line 390).

Response: Fig 6 c,d Degradation assays: we have re-conducted the cell-free degradation assays which include quantification of three biological replicates (as shown in Fig 6b, c, d). Furthermore, we have conducted *in vivo* degradation experiment in the rice protoplast system. The results suggest that OsUBC12-Flag obviously promoted the degradation of OsSnRK1.1-Flag compared to empty-Flag (EV) and OsSnRK1.1-Flag co-expression, in contrast, this degradation was less pronounced in the presence of OsUBC12C92A-Flag (Fig. 6e). These results confirm that OsUBC12 promotes OsSnRK1.1 degradation *in vitro* and *in vivo*, and requires its active-site cysteine residue to do so (as shown in the manuscript in Page 13, line 399-line 413).

7) For the above reasons, the model shown in Fig S11 is not fully supported (UBC12-SnRK1 phosphorylation/ubiquitination)

Response: Thanks for your professional comments. We have supplemented the above evidence to improve the model.

8) Methodology is minimal and not enough details are provided for some sections. Plant material: what temperature was used in the growth chambers? Only IR64 was grown in growth chambers? Please provide detailed description of RNAseq, including how the seeds were imbibed/germinated prior to RNA extraction (and how parental plants were grown), RNA quantity/quality analysis, library preparation, RNAseq analysis, DEG analysis, etc. List of DEGs etc should be provided.

Response: Thanks for your professional comments. We have perfected the Methods section of our revised manuscript and provided exhaustive details. The growth conditions of the plant materials and their controls used in the study, such as light, temperature, etc., also have been provided in detail (as shown in the manuscript in Page 18, line 565-line 581).

In addition, we have provided detailed description of RNAseq (as shown in the manuscript in Page 21, line 655-line 674), and also provided the list of DEGs (as shown in Supplementary Data 1).

9) Genetic background: different genetic backgrounds were used in this study. All figures should note (at least in the legend) which genotype was used as “WT”.

Example: Which genotype were OE constructs transformed into? Furthermore, why were CRISPR/Cas9 mutants generated in the Kongyun *japonica* background, and not in the Koshihikari *japonica*?

Response: Thanks for your professional comments. All Figure Captions of our revised manuscript have labeled the genetic background of the “WT”. Moreover, the genetic backgrounds of both *osubc12-ks* mutants, *osubc12* mutants, *ossnrk1.1* mutants, *osubc12snrk1.1* double mutants and overexpression lines have been described (as shown in the Methods section of manuscript in Page 20, line 639-line 642). Furthermore, as mentioned above, KY131 is a high-quality rice with a large planting area in our cold *Japonica* rice area in Northeast China, and its genotype is consistent with Koshihikari, both of which carry the *japonica OsUBC12* locus (transposon insertion), so we used KY131 as the genetic background to generate *ubc12* crispr mutants for further experiments. At the same time, to provide more evidence about *japonica OsUBC12* locus and cold tolerance, we have also generated *OsUBC12* homozygous knockout mutants (*osubc12-ks1* and *osubc12-ks2*) using a clustered regularly interspaced short palindromic repeat (CRISPR)/CRISPR-associated nuclease 9 (Cas9) system in Koshihikari background (Fig. S10), and supplemented the germination of *osubc12-ks* at 15 ° C (Fig. 1e) and 30 ° C (Fig S9b) (as shown in the manuscript in Page 7, line 208-line 218).

10) Statistical analysis should be shown in all graphs (main and supplem figures).

Response: Thanks for your professional comments. We have supplemented statistical analysis to the supplementary figures, and completed the required statistical analysis for the graphs.

11) Figure legends are minimal or not shown for some supplementary data. Please add or include more details

Response: Thanks for your meticulous reading. We have perfected the supplementary figure captions with sufficient details.

MINOR

CRISPR/Cas9 mutants: was cas9 removed from these lines?

Response: Thanks for your professional comments. All CRISPR/Cas9 mutants in our

manuscript are homozygous mutants with removed cas9 (Response Fig 2).

M: DL2000 plus DNA Marker. EV:pYLCRISPR/Cas9Pubi-H

Response Fig 2 The PCR of *HPT* marker gene in cas9 (pYLCRISPR/Cas9Pubi-H)

Fig S8: the conserved ubiquitin domain should be shown by sequence alignment with known and characterized UBC proteins. As is presented, this figure only shows the presence of a cysteine residue

Response: Thanks for your professional comments. We have provided the protein structure and sequence alignment of OsUBC12 (as shown in Fig. S16a, b).

L114-122: REF 46-49 these are reviews, the original research papers should be cited.

Response: Thanks for your meticulous reading. We have cited the relevant original research papers (as shown in the manuscript in Page 4, line 117-line 122).

L122-124: other papers link SnRK1 to ABA signaling, only one is cited here.

Response: Thanks for your meticulous reading. We have supplemented the relevant papers linking SnRK1 to ABA signaling (as shown in the manuscript in Page 5, line 122-line 125).

L383: What makes UBC12 a “master regulator” of seed germination?

Response: Thanks for your meticulous reading. As suggested by Reviewer 2, the word-“master” has been removed and “master regulator of seed germination” has been modified to “regulator of seed germination” in the manuscript in Page 14, line 451.

Reviewers' Comments:

Reviewer #1:

Remarks to the Author:

The authors have addressed all my concerns.

Reviewer #2:

Remarks to the Author:

Authors have conducted additional analyses and experiments and added the appropriate explanations in addressing my questions and concerns. The revised manuscript was improved. I have no further comments on the revised manuscript.

Reviewer #3:

Remarks to the Author:

The revision has responded my questions, but this reviewer still has concerns as follows:

1. The mechanistic novelty of this work. SnRK1 has been reported for its role in positively regulating ABA signaling which is well known for its role in seed germination, ubiquitination of rice SnRK1 mediated by an E2 (OsUBC12), which works harder under low temperature, is the key finding in this work in terms of mechanistic insight of ABA signaling regulation. Authors just carefully explained the novelty in the rebuttal (response to reviewers). At least, some insightful discussion should be included to highlight the significance of the finding if Editor and other reviewers think the novelty is acceptable to this journal. Some repeated description of the results in Discussion may be deleted or simplified to save space to discuss the significance of the findings.

2. The potential application of OsUBC12 in rice, especially in indica, has not completely convinced this reviewer, and this part needs more logical and data-supported clarification.

First, the potential application of OsUBC12 locus (with the transposon insertion) in indica LTG breeding have been repeatedly emphasized but was not supported with data. For examples, In abstract Line 39 : "Notably, the genomic region carrying the japonica OsUBC12 locus (with the transposon insertion) has been introgressed into the modern elite indica two-line male sterile lines Y58S and J4155S". Line 48 (abstract): "...possible genomic contributions of introgressions of the japonica OsUBC12 locus to trait improvements of indica rice cultivars and provide genetic reference points for improving LTG in indica rice". At the end of Introduction Line 131: "Notably, the japonica OsUBC12 locus harboring the inserted transposon has been introgressed into the modern elite indica two-line male sterile lines Y58S and J4155S." Line 137 "...provide avenues to improve the LTG of indica rice via molecular breeding." Although in the second part of the results, authors provided examples that the Japonica allele (with the transposon insertion) from Nongken58S was detected in several Indica two-line MS lines with parental relatedness to Nongken58S, this only suggests existence of the allele introgression by Indica X Japonica breeding, but not evidence for LTG selection of the allele. By the way, according to the description of the pedigree relationships of Nongken58S with the Indica two-line MS lines (line 253-260), the lines Y58S and J4155S have no pedigree relationship with Nongken58S, unless you clarify that Zhu1S is derived from Guangzhan 63S that has pedigree relationship with Nongken58S. Second, existence of the strong allele for LTG in Indica, which may be introgressed by linkage or hitchhiking effect, is not critical for making the conclusion. To prove the allele is indeed valuable for LTG breeding, a near-isogenic line (NIL) with the strong allele introduced to an Indica background should be generated for testing LTG. Or a NIL with the strong allele introduced to a Japonica background with the weak allele can improve LTG. Otherwise, the conclusion for the LTG breeding value of the gene should be largely attenuated.

Third, for sure, the significant difference in LTG between a group with 50 varieties without the transposon insertion and a group of 21 varieties with transposon insertion is a suggestive result for

the potential application of the gene (transposon Indel variation), but this result is rather predictable if the functional contribution of the gene to LTG has been confirmed to be the transposon insertion. By the way, separating the two groups into two subgroup pairs (Indica: transposons + vs -; Japonica: transposons + vs -) for LTG comparison may provide more suggestive information.

3. Natural variation of the transposon insertion needs further clarification. Line 231: "...we investigated a rice diversity panel consisting of 69 cultivated accessions selected to represent nine rice subpopulations from the 3,000 rice genomes and six wild accessions." (also mentioned in Line 269-270). How could 69 cultivated accessions represent nine rice subpopulations and 'confidently' (a word used by author in Line 233) lead to claim that the transposon is absent in all wild and indica accessions? Although the diversity panel size for the checking the presence/absence of the transposon is not very critical, a panel of 69 accessions is too small to represent anything not even for representing the nine rice subpopulations and six wild relatives. Who knows there may be more interesting (or stronger alleles) in the cultivated and wild rice? Authors then checked the presence/absence of the transposon in a panel containing 197 Asian cultivated rice to support the conclusion, but they still need to rephrase the representativeness of the 69-accession panel, or just integrate the two panel for description.

Reviewer #4:

Remarks to the Author:

The authors have addressed most of the concerns raised in my previous review, thank you. I have only a few additional questions/comments:

1) how were the samples for the RNAseq collected? were the seeds germinated at low temp? How many days after germination were the samples collected? Please provide details in the methods or results.

2) Go analysis revealed that "OsUBC12 may be involved in a variety of biological and physiological processes" however, no further discussion of what these processes may be is provided. In light of SnRK1 role as energy regulator during stress, it would be interesting to further analyze these pathways to determine if there is any connection between UBC12 and SnRNK1 roles in the regulation of metabolic pathways.

3) Fig 2c, 4b need statistical analysis.

4) Interaction studies in *N. benthamiana* provide evidence that UBC12-SnRK1 interaction occurs "in planta", not "in vivo". To conclude that the interaction occurs in vivo, Co-IP needs to be performed in rice. Please revise the conclusion in the abstract and the rest of the manuscript (L371, 373; 494).

5) Fig 6 and text (L408, 413; L745; L1182): Degradation was done in protoplasts, not in vivo. Please correct.

6) Fig S18. Results from Phosphorylation assays are not clear. There is a band (boxed in light blue) in the positive control (ABI5) in the presence of SnRK1.1, and SnRK1+GRIK, but also in the absence of the kinases (and in the absence of ABI5?). Furthermore, how can ABI5 have autophosphorylation activity in vitro (L388)? In contrast, there are bands in UBC12 only in the presence of SnRK1, and SnRK1+GRIK, although weak. Why was it concluded that UBC12 was not phosphorylated? It may be better to blot with anti-His and anti-GST to detect phosphorylated and non-phosphorylated ABI5 and UBC12, respectively. See Han et al., 2020 (Plant Communications) Fig 3B for example of clear phosphorylation of a SnRK1 substrate. Alternatively, the authors may consider removing the phosphorylation part of this paper, since it doesn't add much to the story.

7) L402-404 "Compared to WT extract, the degradation rate of OsSnRK1.1 was significantly decreased in osubc12 (Fig 6b), OsSnRK1.1 degradation was significantly inhibited by the proteasome inhibitor MG132" Fig 6c. To claim significant differences one needs to provide statistical analysis. Please remove "significantly" or provide statistical analysis.

Germination assays in supplem material between Fig S9 and S10 don't have Figure number and legend

REVIEWER COMMENTS

Reviewer #1 (Remarks to the Author):

The authors have addressed all my concerns.

Reviewer #2 (Remarks to the Author):

Authors have conducted additional analyses and experiments and added the appropriate explanations in addressing my questions and concerns. The revised manuscript was improved. I have no further comments on the revised manuscript.

Reviewer #3 (Remarks to the Author):

The revision has responded my questions, but this reviewer still has concerns as follows:

1. The mechanistic novelty of this work. SnRK1 has been reported for its role in positively regulating ABA signaling which is well known for its role in seed germination, ubiquitination of rice SnRK1 mediated by an E2 (OsUBC12), which works harder under low temperature, is the key finding in this work in terms of mechanistic insight of ABA signaling regulation. Authors just carefully explained the novelty in the rebuttal (response to reviewers). At least, some insightful discussion should be included to highlight the significance of the finding if Editor and other reviewers think the novelty is acceptable to this journal. Some repeated description of the results in Discussion may be deleted or simplified to save space to discuss the significance of the findings.

2. The potential application of OsUBC12 in rice, especially in indica, has not completely convinced this reviewer, and this part needs more logical and data-supported clarification.

First, the potential application of OsUBC12 locus (with the transposon insertion) in indica LTG breeding have been repeatedly emphasized but was not supported with data. For examples, In abstract Line 39: “Notably, the genomic region carrying the japonica OsUBC12 locus (with the transposon insertion) has been introgressed into the modern elite indica two-line male sterile lines Y58S and J4155S”. Line 48 (abstract): “...possible genomic contributions of introgressions of the japonica OsUBC12 locus to trait improvements of indica rice cultivars and provide genetic

reference points for improving LTG in indica rice”. At the end of Introduction Line 131: “Notably, the japonica OsUBC12 locus harboring the inserted transposon has been introgressed into the modern elite indica two-line male sterile lines Y58S and J4155S.” Line 137 “...provide avenues to improve the LTG of indica rice via molecular breeding.” Although in the second part of the results, authors provided examples that the Japonica allele (with the transposon insertion) from Nongken58S was detected in several Indica two-line MS lines with parental relatedness to Nongken58S, this only suggests existence of the allele introgression by Indica X Japonica breeding, but not evidence for LTG selection of the allele. By the way, according to the description of the pedigree relationships of Nongken58S with the Indica two-line MS lines (line 253-260), the lines Y58S and J4155S have no pedigree relationship with Nongken58S, unless you clarify that Zhu1S is derived from Guangzhan 63S that has pedigree relationship with Nongken58S.

Second, existence of the strong allele for LTG in Indica, which may be introgressed by linkage or hitchhiking effect, is not critical for making the conclusion. To prove the allele is indeed valuable for LTG breeding, a near-isogenic line (NIL) with the strong allele introduced to an Indica background should be generated for testing LTG. Or a NIL with the strong allele introduced to a Japonica background with the weak allele can improve LTG. Otherwise, the conclusion for the LTG breeding value of the gene should be largely attenuated.

Third, for sure, the significant difference in LTG between a group with 50 varieties without the transposon insertion and a group of 21 varieties with transposon insertion is a suggestive result for the potential application of the gene (transposon Indel variation), but this result is rather predictable if the functional contribution of the gene to LTG has been confirmed to be the transposon insertion. By the way, separating the two groups into two subgroup pairs (Indica: transposons + vs -; Japonica: transposons + vs -) for LTG comparison may provide more suggestive information.

3. Natural variation of the transposon insertion needs further clarification. Line 231: “...we investigated a rice diversity panel consisting of 69 cultivated accessions selected to represent nine rice subpopulations from the 3,000 rice genomes and six wild accessions.” (also mentioned in Line 269-270). How could 69 cultivated accessions represent nine rice subpopulations and ‘confidently’ (a word used by author in Line 233) lead to claim that the transposon is absent in all wild and indica accessions? Although the diversity panel size for the checking the presence/absence of

the transposon is not very critical, a panel of 69 accessions is too small to represent anything not even for representing the nine rice subpopulations and six wild relatives. Who knows there may be more interesting (or stronger alleles) in the cultivated and wild rice? Authors then checked the presence/absence of the transposon in a panel containing 197 Asian cultivated rice to support the conclusion, but they still need to rephrase the representativeness of the 69-accession panel, or just integrate the two panel for description.

Reviewer #4 (Remarks to the Author):

The authors have addressed most of the concerns raised in my previous review, thank you. I have only a few additional questions/comments:

1) how were the samples for the RNAseq collected? were the seeds germinated at low temp? How many days after germination were the samples collected? Please provide details in the methods or results.

2) Go analysis revealed that “OsUBC12 may be involved in a variety of biological and physiological processes” however, no further discussion of what these processes may be is provided. In light of SnRK1 role as energy regulator during stress, it would be interesting to further analyze these pathways to determine if there is any connection between UBC12 and SnRNK1 roles in the regulation of metabolic pathways.

3) Fig 2c, 4b need statistical analysis.

4) Interaction studies in *N. benthamiana* provide evidence that UBC12-SnRK1 interaction occurs “in planta”, not “in vivo”. To conclude that the interaction occurs in vivo, Co-IP needs to be performed in rice. Please revise the conclusion in the abstract and the rest of the manuscript (L371, 373; 494).

5) Fig 6 and text (L408, 413; L745; L1182): Degradation was done in protoplasts, not in vivo. Please correct.

6) Fig S18. Results from Phosphorylation assays are not clear. There is a band (boxed in light blue) in the positive control (ABI5) in the presence of SnRK1.1, and SnRK1+GRIK, but also in the absence of the kinases (and in the absence of ABI5?). Furthermore, how can ABI5 have autophosphorylation activity in vitro (L388)? In contrast, there are bands in UBC12 only in the presence of SnRK1, and SnRK1+GRIK, although weak. Why was it concluded that UBC12 was not phosphorylated? It may be better to blot with anti-His and anti-GST to detect

phosphorylated and non-phosphorylated ABI5 and UBC12, respectively. See Han et al., 2020 (Plant Communications) Fig 3B for example of clear phosphorylation of a SnRK1 substrate. Alternatively, the authors may consider removing the phosphorylation part of this paper, since it doesn't add much to the story.

7) L402-404 “Compared to WT extract, the degradation rate of OsSnRK1.1 was significantly decreased in osubc12 (Fig 6b), OsSnRK1.1 degradation was significantly inhibited by

the proteasome inhibitor MG132” Fig 6c. To claim significant differences one needs to provide statistical analysis. Please remove “significantly” or provide statistical analysis.

Germination assays in supplem material between Fig S9 and S10 don't have Figure number and legend

Dear reviewers,

Re: Manuscript ID: NCOMMS-23-05556A

Title: A transposon insertion in the *OsUBC12* promoter enhances cold tolerance during germination in *japonica* rice (*Oryza sativa*)

We highly appreciate your insightful suggestions and comments, all of which are very helpful to the improvement of our manuscript. We have studied these comments carefully and have made corrections accordingly. Besides, evolutionary biologist Hongru Wang (Agricultural Genomics Institute at Shenzhen, Chinese Academy of Agricultural Sciences), the second author of our manuscript, has reevaluated and revised the Natural variations and potential application part of this paper. All amendments are shown with track changes or colour highlighting in the revised version. Below, we have listed our point-by-point responses.

Thanks for your time and consideration again. And we are looking forward to your positive response.

Yours sincerely,

Jun Fang, Ph. D, Researcher

September 22, 2023

Response to reviewer's comments:

Reviewer #1 (Remarks to the Author):

The authors have addressed all my concerns.

Response: Thank you for your support and recognition of our manuscript. We really appreciate your efforts in reviewing the manuscript.

Reviewer #2 (Remarks to the Author):

Authors have conducted additional analyses and experiments and added the appropriate explanations in addressing my questions and concerns. The revised manuscript was improved. I have no further comments on the revised manuscript.

Response: Thank you for your support and recognition of our manuscript. We really appreciate your efforts in reviewing the manuscript.

Reviewer #3 (Remarks to the Author):

The revision has responded my questions, but this reviewer still has concerns as follows:

Response: Thank you for your support and recognition of our manuscript. Your careful suggestion and insightful comments have further improved our revised manuscripts and highlighted the novelty of the gene and its working mechanism. We really appreciate your efforts in reviewing our manuscript. Our point-by-point responses are detailed below.

1. The mechanistic novelty of this work. SnRK1 has been reported for its role in positively regulating ABA signaling which is well known for its role in seed germination, ubiquitination of rice SnRK1 mediated by an E2 (OsUBC12), which works harder under low temperature, is the key finding in this work in terms of mechanistic insight of ABA signaling regulation. Authors just carefully explained the novelty in the rebuttal (response to reviewers). At least, some insightful discussion should be included to highlight the significance of the finding if Editor and other reviewers think the novelty is acceptable to this journal. Some repeated description of the results in Discussion may be deleted or simplified to save space to discuss the significance of the findings.

Response: Thank you for your professional suggestion. We are very sorry for our negligence. We have added some description of the mechanistic novelty to **the Discussion section in our revised manuscript (as shown in the revised manuscript in Page 16, line 495-line 504)**, and have simplified some repeated description of the results in **Discussion section** to save space to discuss the significance of our findings **(as shown in the revised manuscript in Page 16, line 492, line 493)**.

2. The potential application of OsUBC12 in rice, especially in indica, has not completely convinced this reviewer, and this part needs more logical and data-supported clarification.

First, the potential application of OsUBC12 locus (with the transposon insertion) in indica LTG breeding have been repeatedly emphasized but was not supported with data. For examples, In abstract Line 39: “Notably, the genomic region carrying the japonica OsUBC12 locus (with the transposon insertion) has been introgressed into the modern elite indica two-line male sterile lines Y58S and J4155S”. Line 48 (abstract): “...possible genomic contributions of introgressions of the japonica OsUBC12 locus to trait improvements of indica rice cultivars and provide genetic reference points for improving LTG in indica rice”. At the end of Introduction Line 131: “Notably, the japonica OsUBC12 locus harboring the inserted transposon has been introgressed into the modern elite indica two-line male sterile lines Y58S and J4155S.” Line 137 “...provide avenues to improve the LTG of indica rice via molecular breeding.” Although in the second part of the results, authors provided examples that the Japonica allele (with the transposon insertion) from Nongken58S was detected in several Indica two-line MS lines with parental relatedness to Nongken58S, this only suggests existence of the allele introgression by Indica X Japonica breeding, but not evidence for LTG selection of the allele. By the way, according to the description of the pedigree relationships of Nongken58S with the Indica two-line MS lines (line 253-260), the lines Y58S and J4155S have no pedigree relationship with Nongken58S, unless you clarify that Zhu1S is derived from Guangzhan 63S that has pedigree relationship with Nongken58S.

Response: Thank you for your careful comments. We are very sorry for our overemphasize on the potential application of *OsUBC12* locus (with the transposon

insertion). We now have toned down our description.

As shown in the Abstract of our revised manuscript in Page 2, line 38-line 42,

“Notably, the genomic region carrying the japonica OsUBC12 locus (with the transposon insertion) has been introgressed into the modern elite indica two-line male sterile lines Y58S and J4155S.” has been modified to “The variation detection in eight representative two-line male sterile lines suggested the existence of this allele introgression by *indica-japonica* hybridization breeding, and varieties carrying the *japonica* OsUBC12 locus (transposon insertion) have higher low-temperature germinability than varieties without the locus.”

As shown in the Abstract of our revised manuscript in Page 2, line 49-line 50,

“…possible genomic contributions of introgressions of the japonica OsUBC12 locus to trait improvements of indica rice cultivars and provide genetic reference points for improving LTG in indica rice” has been modified to “These findings shed light on the underlying mechanisms of UBC12 regulating LTG and provide genetic reference points for improving LTG in *indica* rice.”

As shown in the end of Introduction of our revised manuscript in Page 5, line

131-line 132, “Notably, the japonica OsUBC12 locus harboring the inserted transposon has been introgressed into the modern elite indica two-line male sterile lines Y58S and J4155S.” has been modified to “And this allele introgression has been applied in *indica-japonica* hybridization breeding.”

As shown in the end of Introduction of our revised manuscript in Page 5, line

136-line 137, “…provide avenues to improve the LTG of indica rice via molecular breeding.” has been modified to “provide a potential genetic locus that may be applied to improve the LTG of *indica* rice.”

In addition, we have also checked other issues related to overemphasize in our revised manuscript.

Second, existence of the strong allele for LTG in Indica, which may be introgressed by linkage or hitchhiking effect, is not critical for making the conclusion. To prove the allele is indeed valuable for LTG breeding, a near-isogenic line (NIL) with the strong allele introduced to an Indica background should be generated for testing LTG. Or a NIL with the strong allele introduced to a Japonica background with the weak allele can improve LTG. Otherwise, the conclusion for the LTG breeding value of the gene

should be largely attenuated.

Response: Thank you for your professional comments. We strongly agree with your viewpoint on generating the near-isogenic line (NIL) for testing LTG to further analyze the LTG breeding valuable of the allele. We have started to supplement and construct these NILs by consecutive selective backcrosses. However, as you know, the obtain of NILs requires multiple continuous backcross, which takes a very long time. We also supplemented the LTG test of SL2016, the CSSL with IR64 introgression (IR64 introgression segments containing the IR64 *OsUBC12* locus) in an Koshihikari background and whose *UBC12* transcript levels were significantly lower than those of the control Koshihikari (Fig. S13b, c). The results show results showed that the low-temperature germinability of SL2016 was significantly lower than that of Koshihikari (Fig. S13d). Based on our current experimental results, the expression analysis of two reciprocal CSSLs, including the CSSL with Koshihikari introgression in an IR64 background (SL2116 and SL2117) and the CSSL with IR64 introgression in an Koshihikari background (SL2016), the dual-luciferase reporter assay, the LTG analysis of *OsUBC12* homozygous knockout mutants, the LTG analysis of two reciprocal CSSLs, the LTG analysis of *OsUBC12*-transgenic lines, the natural variations of *OsUBC12*, the variation detection in eight representative two-line male sterile lines, the LTG phenotype between the group with 50 varieties without the transposon insertion and the group of 21 varieties with transposon insertion, and the simulated rice direct seeding experiment comprehensively suggested this allele does indeed have potential LTG breeding value. In addition, we will publish and report back to you and the community after obtaining LTG analysis of the relevant NILs, and we have added this part of the work to **the end of Discussion of our revised manuscript (as shown in the revised manuscript in Page 18, line 556-line 560)**.

Third, for sure, the significant difference in LTG between a group with 50 varieties without the transposon insertion and a group of 21 varieties with transposon insertion is a suggestive result for the potential application of the gene (transposon Indel variation), but this result is rather predictable if the functional contribution of the gene to LTG has been confirmed to be the transposon insertion. By the way, separating the two groups into two subgroup pairs (Indica: transposons + vs -; Japonica: transposons + vs -) for LTG comparison may provide more suggestive information.

Response: Thank you for your careful suggestion. In this experiment, considering

that the sample size of *indica* with transposon insertion was too small (only Y58S and Jing4155S), we only divided *japonica* into two groups for LTG comparison. The results showed that the low-temperature germinability of varieties carrying the *japonica OsUBC12* locus in *japonica* was indeed higher than that of varieties without the locus (Fig. S13a). We have added the related descriptions to **the Results section of the revised manuscript in Page 9, line 273-line 276**.

3. Natural variation of the transposon insertion needs further clarification. Line 231: “...we investigated a rice diversity panel consisting of 69 cultivated accessions selected to represent nine rice subpopulations from the 3,000 rice genomes and six wild accessions.” (also mentioned in Line 269-270). How could 69 cultivated accessions represent nine rice subpopulations and ‘confidently’ (a word used by author in Line 233) lead to claim that the transposon is absent in all wild and *indica* accessions? Although the diversity panel size for the checking the presence/absence of the transposon is not very critical, a panel of 69 accessions is too small to represent anything not even for representing the nine rice subpopulations and six wild relatives. Who knows there may be more interesting (or stronger alleles) in the cultivated and wild rice? Authors then checked the presence/absence of the transposon in a panel containing 197 Asian cultivated rice to support the conclusion, but they still need to rephrase the representativeness of the 69-accession panel, or just integrate the two panel for description.

Response: Thank you for your reminding and careful suggestion. We agree that the “representativeness” claim is too strong given the small number of accessions. We now have toned down our description and integrated the two panel for description as follows:

“To study the natural variation in *pOsUBC12* among rice populations, we investigated a rice diversity panel consisting of 69 cultivated accessions selected from nine rice subpopulations of the 3,000 rice genomes and six wild accessions⁶¹. We determined the presence/absence state for all the accessions and found that the transposon is absent in all wild and *indica* accessions in the panel, but show high frequency in *japonica* accessions (21/24) (Fig. 2a). Furthermore, we extended our analyses using the high quality rice genome assembly dataset published in Shang et al.⁶². We called the presence absence variation of the transposon in 197 Asian domesticated rice accessions with subgroup information. The result is consistent with the result obtained

with the dataset from Zhang et al. ⁶¹: the transposon is enriched in *japonica* rice (28/58), and absent in aus group, and only found in one accession of 135 accessions of *indica* variety (Fig. S12). These results suggest that the transposon insertion occurred in the *japonica* lineage (Fig. 2a, S12). The fixation index (Fst) among these subpopulations was 0.976, suggestive of significant genetic divergence among them (Fig. 2a).” (as shown in the revised manuscript in Page 8, line 230-line 244)

We have done our best to improve the manuscript, and hope that the revision could address your concern and receive your approval and support.

Reviewer #4 (Remarks to the Author):

The authors have addressed most of the concerns raised in my previous review, thank you. I have only a few additional questions/comments:

Response: Thank you for your support and recognition of our manuscript. Your careful suggestion and insightful comments have further improved our revised manuscript. We really appreciate your efforts in reviewing our manuscript. Our point-by-point responses are detailed below.

1) how were the samples for the RNAseq collected? were the seeds germinated at low temp? How many days after germination were the samples collected? Please provide details in the methods or results.

Response: Thank you for your careful suggestion. Three independent *osubc12* mutant and wild-type (WT, KY131) seeds were grown and harvested from the field at the experimental stations of the Northeast Institute of Geography and Agroecology, Chinese Academy of Sciences, in Harbin, Heilongjiang province, China. Seeds were soaked in sterile water at 30 ° C for 12h and then drained, followed by RNA extraction for RNA-seq analysis. We have added the descriptions to **the methods section of the revised manuscript in Page 21, line 655-line 659.**

2) Go analysis revealed that “OsUBC12 may be involved in a variety of biological and physiological processes” however, no further discussion of what these processes may be is provided. In light of SnRK1 role as energy regulator during stress, it would

be interesting to further analyze these pathways to determine if there is any connection between UBC12 and SnRK1 roles in the regulation of metabolic pathways.

Response: Thank you for your professional comments. We agree that it would be interesting to further analyze the energy metabolism pathways influenced by UBC12-OsSnRK1 to determine the connection, role, and mechanism of UBC12 and OsSnRK1 in the regulation of metabolic pathways. We have added the descriptions to **the Discussion section of the revised manuscript in Page 17, line 547-line 550.**

3) Fig 2c, 4b need statistical analysis.

Response: Thanks for your professional suggestion. We have supplemented statistical analysis to the Fig 2c and Fig 4b of our revised manuscript.

4) Interaction studies in *N. benthamiana* provide evidence that UBC12-SnRK1 interaction occurs “in planta”, not “in vivo”. To conclude that the interaction occurs in vivo, Co-IP needs to be performed in rice. Please revise the conclusion in the abstract and the rest of the manuscript (L371, 373; 494).

Response: Thanks for your professional comments. We have amended this conclusion, and revised the conclusion in the rest of the manuscript. **(as shown in the revised manuscript in L379, L381, L485)**

5) Fig 6 and text (L408, 413; L745; L1182): Degradation was done in protoplasts, not in vivo. Please correct.

Response: Thanks for your professional comments. We have amended this description, and corrected the description in the rest of the manuscript. **(as shown in the revised manuscript in L399, L404, L460, L744, L1179)**

6) Fig S18. Results from Phosphorylation assays are not clear. There is a band (boxed in light blue) in the positive control (ABI5) in the presence of SnRK1.1, and SnRK1+GRIK, but also in the absence of the kinases (and in the absence of ABI5?). Furthermore, how can ABI5 have autophosphorylation activity in vitro (L388)? In contrast, there are bands in UBC12 only in the presence of SnRK1, and SnRK1+GRIK, although weak. Why was it concluded that UBC12 was not phosphorylated? It may be better to blot with anti-His and anti-GST to detect

phosphorylated and non-phosphorylated ABI5 and UBC12, respectively. See Han et al., 2020 (Plant Communications) Fig 3B for example of clear phosphorylation of a SnRK1 substrate. Alternatively, the authors may consider removing the phosphorylation part of this paper, since it doesn't add much to the story.

Response: Thank you for your careful suggestion. We have removed the phosphorylation part of this paper, and improved the model at the same time (**as shown in Fig S23 of the revised manuscript**).

7) L402-404 “Compared to WT extract, the degradation rate of OsSnRK1.1 was significantly decreased in osubc12 (Fig 6b), OsSnRK1.1 degradation was significantly inhibited by the proteasome inhibitor MG132” Fig 6c. To claim significant differences one needs to provide statistical analysis. Please remove “significantly” or provide statistical analysis.

Response: Thank you for your professional comments. According to your careful suggestion, We removed the “significantly” in these statements (**as shown in the revised manuscript in Page 13, L392-L395**).

Germination assays in supplem material between Fig S9 and S10 don't have Figure number and legend

Response: Thank you for your careful comments. We have added the corresponding Figure number and legend between Fig S9 and S10 in Supplementary information.

We have done our best to improve the manuscript, and hope that the revision could address your concern and receive your approval and support.

Reviewers' Comments:

Reviewer #3:

Remarks to the Author:

Authors have carefully answered my concerns and attenuated the conclusions on the selection and potential application of the locus for LTG. A CSSL with the weak allele introgressed into japonica variety Koshihikari showed decreased LTG, partially addressed my concerns.

However, several sentences still emphasize that the allele has been applied in breeding, with which I cannot agree. For example,

Line 130-131 : "And this allele introgression has been applied in indica-japonica hybridization breeding"; Line 285-286: ..."japonica OsUBC12 locus introgressions has been applied in indica-japonica hybridization breeding". "has been applied" is misleading. Actually, this allele or this allele introgression has not been directly applied, at least no evidence suggests that this allele has been intentionally selected or applied. It just exists or has been indirectly introgressed by chance in some lines derived from indica-japonica hybridization.

Reviewer #4:

Remarks to the Author:

Reviewer #4 (Remarks to the Author):

1) how were the samples for the RNAseq collected? were the seeds germinated at low temp? How many days after germination were the samples collected? Please provide details in the methods or results.

Response: Thank you for your careful suggestion. Three independent *osubc12* mutant and wild-type (WT, KY131) seeds were grown and harvested from the field at the experimental stations of the Northeast Institute of Geography and Agroecology, Chinese Academy of Sciences, in Harbin, Heilongjiang province, China. Seeds were soaked in sterile water at 30 °C for 12h and then drained, followed by RNA extraction for RNA-seq analysis. We have added the descriptions to the methods section of the revised manuscript in Page 21, line 655-line 659.

> The fact that seeds used for RNAseq were germinated at 30C should be included also in the results, as it was an important missing detail in the paper. In light of this, it is now unclear to this reviewer why transcriptomic analysis was conducted in wt and mutant seeds germinated at 30C instead of low temperature (15C), since in this paper the authors aimed to explore the mechanism by which OsUBC12 regulates LTG. In L292-298 the authors wrote: "To explore the molecular mechanism by which OsUBC12 regulates LTG, we first analyzed the expression kinetics of OsUBC12 in response to cold and ABA. As shown in Fig. 3a, the transcription of OsUBC12 was induced by both cold and ABA, implying that OsUBC12 may be involved in responding to ABA and cold stress. Then, we examined differences in gene expression in *osubc12* mutant plants compared to WT using transcriptome deep sequencing (RNA-seq)." However, the authors conducted the experiment at 30C. Furthermore, the RNAseq does not support the conclusion made in the paper that "OsUBC12-promoted LTG is associated with suppression of ABA signaling". Indeed, Go enrichment analysis does not show enrichment in ABA signaling genes, or low temperature signaling genes. I was hoping the authors would explain how the pathways that are enriched in the Go enrichment analysis linked LTG to SnRK1-mediated ABA signaling (see question 2 below), instead the authors only included my sentence in the discussion (L547-551).

2) Go analysis revealed that "OsUBC12 may be involved in a variety of biological and physiological processes" however, no further discussion of what these processes may be is provided. In light of SnRK1 role as energy regulator during stress, it would be interesting to further analyze these pathways to determine if there is any connection between UBC12 and SnRNK1 roles in the regulation of metabolic pathways.

Response: Thank you for your professional comments. We agree that it would be interesting to further analyze the energy metabolism pathways influenced by UBC12-OsSnRK1 to determine the connection, role, and mechanism of UBC12 and OsSnRK1 in the regulation of metabolic pathways. We have added the descriptions to the Discussion section of the revised manuscript in Page 17, line 547-line 551.

> Well, the authors have not addressed this point, they have merely re-stated my comment in the paper, with no further analysis shown. It is still unclear how degradation of SnRK1 by UBC12 alters sensitivity to low temperature. See also comment about RNAseq above.

3) Fig 2c, 4b need statistical analysis.

Response: Thanks for your professional suggestion. We have supplemented statistical analysis to the Fig 2c and Fig 4b of our revised manuscript.

>Thank you

4) Interaction studies in *N. benthamiana* provide evidence that UBC12-SnRK1 interaction occurs "in planta", not "in vivo". To conclude that the interaction occurs in vivo, Co-IP needs to be performed in rice. Please revise the conclusion in the abstract and the rest of the manuscript (L371, 373; 494).

Response: Thanks for your professional comments. We have amended this conclusion, and revised the conclusion in the rest of the manuscript. (as shown in the revised manuscript in L379, L381, L485)

>Thank you

5) Fig 6 and text (L408, 413; L745; L1182): Degradation was done in protoplasts, not in vivo. Please correct.

Response: Thanks for your professional comments. We have amended this description, and corrected the description in the rest of the manuscript. (as shown in the revised manuscript in L399, L404, L460, L744, L1179)

>Thank you

6) Fig S18. Results from Phosphorylation assays are not clear. There is a band (boxed in light blue) in the positive control (ABI5) in the presence of SnRK1.1, and SnRK1+GRIK, but also in the absence of the kinases (and in the absence of ABI5?). Furthermore, how can ABI5 have autophosphorylation activity in vitro (L388)? In contrast, there are bands in UBC12 only in the presence of SnRK1, and SnRK1+GRIK, although weak. Why was it concluded that UBC12 was not phosphorylated? It may be better to blot with anti-His and anti-GST to detect phosphorylated and non-phosphorylated ABI5 and UBC12, respectively. See Han et al., 2020 (Plant Communications) Fig 3B for example of clear phosphorylation of a SnRK1 substrate. Alternatively, the authors may consider removing the phosphorylation part of this paper, since it doesn't add much to the story.

Response: Thank you for your careful suggestion. We have removed the phosphorylation part of this paper, and improved the model at the same time (as shown in Fig S23 of the revised manuscript).

>Thank you

7) L402-404 "Compared to WT extract, the degradation rate of OsSnRK1.1 was significantly decreased in *osubc12* (Fig 6b), OsSnRK1.1 degradation was significantly inhibited by the proteasome inhibitor MG132" Fig 6c. To claim significant differences one needs to provide statistical analysis. Please remove "significantly" or provide statistical analysis.

Response: Thank you for your professional comments. According to your careful suggestion, We removed the "significantly" in these statements (as shown in the revised manuscript in Page 13, L392-L395).

>Thank you

Germination assays in supplement material between Fig S9 and S10 don't have Figure number and legend

Response: Thank you for your careful comments. We have added the corresponding Figure number and legend between Fig S9 and S10 in Supplementary information.

>Thank you

We have done our best to improve the manuscript, and hope that the revision could address your concern and receive your approval and support.

>Aside from my comments above, I agree with reviewer#1 about condensing the discussion and highlighting the novelty of this work. Also, the new paragraph added in the discussion is not very clear (L497-505”).

Dear reviewers,

Re: Manuscript ID: NCOMMS-23-05556B

Title: A transposon insertion in the *OsUBC12* promoter enhances cold tolerance during germination in *japonica* rice (*Oryza sativa*)

We highly appreciate the insightful suggestions and comments from four reviewers, all of which are very helpful to the improvement of our manuscript. We have studied these comments carefully and have made correction which we hope the revision could address your concern and meet with approval. All amendments are highlighted with colour highlighting in the revised version. Below, we have listed our point-by-point responses.

Thanks for your contribution in reviewing our manuscript again. And we are looking forward to your positive response.

Yours sincerely,

Jun Fang, Ph. D, Researcher

December 21, 2023

Response to reviewer's comments:

Reviewer #3 (Remarks to the Author):

Authors have carefully answered my concerns and attenuated the conclusions on the selection and potential application of the locus for LTG. A CSSL with the weak allele introgressed into japonica variety Koshihikari showed decreased LTG, partially addressed my concerns.

However, several sentences still emphasize that the allele has been applied in breeding, with which I cannot agree. For example,

Line 130-131: “And this allele introgression has been applied in *indica-japonica* hybridization breeding”; Line 285-286: ...“*japonica OsUBC12* locus introgressions has been applied in *indica-japonica* hybridization breeding”. “has been applied” is misleading. Actually, this allele or this allele introgression has not been directly applied, at least no evidence suggests that this allele has been intentionally selected or applied. It just exists or has been indirectly introgressed by chance in some lines derived from *indica-japonica* hybridization.

Response: Thanks for your professional guidance and careful suggestion. **As shown in Page 5, line 131-line 134 of the revised manuscript**, “And this allele introgression has been applied in *indica-japonica* hybridization breeding” has been modified to “And the variation detection analysis suggested the existence of this allele introgression by *indica-japonica* hybridization breeding, while varieties carrying the *japonica OsUBC12* locus (transposon insertion) have higher low-temperature germinability than varieties without the locus” .

As shown in Page 10, line 288-line 289 of the revised manuscript, “*japonica OsUBC12* locus introgressions has been applied in *indica-japonica* hybridization breeding” has been modified to “*japonica OsUBC12* locus introgressions exists in *indica-japonica* hybridization breeding” .

Moreover, to further verify whether *OsUBC12* could affect the *OsSnRK1.1* protein level *in vivo*, we have also prepared anti-*OsSnRK1.1* antibody by Abmart Shanghai Co.,Ltd, and performed western blotting to measure *OsSnRK1.1* protein levels in WT, *osubc12* mutants, *OsUBC12-OE* and *OsUBC12^{C92A}-OE* transgenic lines. The results showed that *OsSnRK1.1* protein levels were clearly increased in *osubc12* mutants and obviously decreased in *OsUBC12-OE* lines compared to WT, whereas

the reduction of OsSnRK1.1 protein levels in *OsUBC12^{C92A}-OE* lines was not pronounced (as shown in **Response Fig 1 below**, or **Page 13, line 404-line 412 and Fig. 6e of the revised manuscript**).

Response Fig 1 (Namely Fig. 6e in the revised manuscript):

Fig. 6e OsSnRK1.1 protein levels of WT, *osubc12* mutants, *OsUBC12-OE* and *OsUBC12^{C92A}-OE* transgenic lines.

Total proteins extracted from the seeds of WT, *osubc12* mutants, *OsUBC12-OE* and *OsUBC12^{C92A}-OE* transgenic lines, was subjected to precipitates with anti-OsSnRK1.1 or anti-OsActin respectively by western blotting analysis. OsActin was used as the loading control. OsSnRK1.1 protein levels were visualized by western blotting using an anti-OsSnRK1.1 antibody. The anti-OsSnRK1.1 antibody was prepared by Abmart Shanghai Co.,Ltd.

We hope that the revision could address your concern. Thanks again for your contribution in reviewing our manuscript, all of which are very helpful to the improvement of our manuscript.

Reviewer #4 (Remarks to the Author):

1) how were the samples for the RNAseq collected? were the seeds germinated at low temp? How many days after germination were the samples collected? Please provide details in the methods or results.

Response: Thank you for your careful suggestion. Three independent *osubc12* mutant and wild-type (WT, KY131) seeds were grown and harvested from the field at the experimental stations of the Northeast Institute of Geography and Agroecology, Chinese Academy of Sciences, in Harbin, Heilongjiang province, China. Seeds were soaked in sterile water at 30 °C for 12h and then drained, followed by RNA extraction for RNA-seq analysis. We have added the descriptions to the methods section of the revised manuscript in Page 21, line 655-line 659.

> The fact that seeds used for RNAseq were germinated at 30C should be included also in the results, as it was an important missing detail in the paper. In light of this, it is now unclear to this reviewer why transcriptomic analysis was conducted in wt and mutant seeds germinated at 30C instead of low temperature (15C), since in this paper the authors aimed to explore the mechanism by which OsUBC12 regulates LTG. In L292-298 the authors wrote: “To explore the molecular mechanism by which OsUBC12 regulates LTG, we first analyzed the expression kinetics of OsUBC12 in response to cold and ABA. As shown in Fig. 3a, the transcription of OsUBC12 was induced by both cold and ABA, implying that OsUBC12 may be involved in responding to ABA and cold stress. Then, we examined differences in gene expression in *osubc12* mutant plants compared to WT using transcriptome deep sequencing (RNA-seq).” However, the authors conducted the experiment at 30C. Furthermore, the RNAseq does not support the conclusion made in the paper that “OsUBC12-promoted LTG is associated with suppression of ABA signaling”. Indeed, Go enrichment analysis does not show enrichment in ABA signaling genes, or low temperature signaling genes. I was hoping the authors would explain how the pathways that are enriched in the Go enrichment analysis linked LTG to SnRK1-mediated ABA signaling (see question 2 below), instead the authors only included my sentence in the discussion (L547-551).

Response: Thank you for your professional comments and suggestions. We have re-conducted the RNA-seq of both WT and *osubc12* mutant seeds germinated at low temperature (15 ° C) (as shown in Response Fig 2 below, which is also namely

Fig. S14 of the revised manuscript). We identified 3526 differentially expressed genes (DEGs) (2063 up-regulated and 1463 down-regulated) in *osubc12* mutants compared to the WT using a 1.2-fold change in expression and p-value < 0.05 as the threshold (**Response Fig 2a below or Fig. S14a, Supplementary Data 1 of the revised manuscript**). Gene ontology (GO) enrichment analysis revealed that OsUBC12 may be involved in a variety of biological and physiological processes, such as response to abiotic stimulus, response to temperature stimulus, response to cold, abscisic acid-activated signaling pathway, response to abscisic acid, protein binding ect (**Response Fig 2b below or Fig. S14b, Supplementary Data 1 of the revised manuscript**). We have added the related descriptions to **the Results section of the revised manuscript in Page 10, line 298-line 308**.

Go enrichment analysis of DEGs in *osubc12* mutants vs. WT not only showed the enrichment of low-temperature related genes (**Response Fig 2b below or Fig. S14b of the revised manuscript**), but also the enrichment of ABA signaling genes (**Response Fig 2b below or Fig. S14b of the revised manuscript**), further supporting the conclusion that OsUBC12-promoted LTG is associated with suppression of ABA signaling. The descriptions have been added to **the revised manuscript in Page 15, line 471-line 474**, and we have explained the corresponding results in the revised manuscript as follows:

“Considering the temperature-dependent function of OsUBC12 and its biochemical properties as an E2 ubiquitin-conjugating enzyme, we analyzed the effect of low temperature on the post-translational level of UBC12. The results showed that low temperature could promote the accumulation of OsUBC12 protein *in vitro* and in protoplasts (Fig. S22a, b). At the transcriptional level, *OsUBC12* is induced by both cold and ABA. (Fig. 3a). These observations further indicate the priority regulatory mechanism of UBC12 for low-temperature germination. Moreover, Go enrichment analysis of DEGs in *osubc12* mutants vs. WT not only showed the enrichment of low-temperature related genes (Fig. S14b), but also the enrichment of ABA signaling genes (Fig. S14b). The expression levels of *OsABI5* and *OsRAB21* were higher in *osubc12* mutants than in WT (Fig. 3b, c). Knockdown of *OsUBC12* also increased ABA sensitivity (Fig. 3d-h). These findings indicate that OsUBC12-promoted LTG is associated with suppression of ABA signaling. The results obtained by overexpression analysis of transgenic rice lines showed an agreement with that of *osubc12* mutants, and blocking the 92th cysteine site of OsUBC12 could significantly suppress its

regulatory effect on germination (Fig. 4b-k). Our results thus provide evidence that OsUBC12 accelerates LTG by repressing ABA signaling via its conserved ubiquitination function. ” (as shown in Page 15, line 465-line 482 of the revised manuscript).

In addition, raw RNA sequencing data are available at the NCBI Sequence ReadArchive (SRA) under accession PRJNA1050330 (as shown in the Data availability section of the revised manuscript in Page 24, line 789-line 790), and we have provided detailed description of RNA-seq in the methods section of the revised manuscript in Page 21, line 675-line 695.

Response Fig 2 (Namely Fig. S14 in the revised manuscript):

Fig. S14 Volcano plots, GO enrichment and Heat map analysis of the significantly differentially expressed genes in *osubc12* mutants vs. WT (KY131) seeds germinated at low temperature (15 °C) detected by RNA-seq analysis.

(a) Volcano plots of the significantly differentially expressed genes in *osubc12* mutants vs. WT (KY131) seeds germinated at low temperature (15 °C). (b) GO enrichment analysis of the significantly differentially expressed genes in *osubc12* mutants vs. WT (KY131) germinated at low temperature (15 °C). (c) Heat map of microarray expression profiles for ABA-related genes in *osubc12* mutants vs. WT (KY131) germinated at low temperature (15 °C). The color scale represents the $\log_2(\text{FPKM}+1)$ and then standardized by “scale()” in R script.

2) Go analysis revealed that “OsUBC12 may be involved in a variety of biological and physiological processes” however, no further discussion of what these processes may be is provided. In light of SnRK1 role as energy regulator during stress, it would be interesting to further analyze these pathways to determine if there is any connection between UBC12 and SnRNK1 roles in the regulation of metabolic pathways.

Response: Thank you for your professional comments. We agree that it would be interesting to further analyze the energy metabolism pathways influenced by UBC12-OsSnRK1 to determine the connection, role, and mechanism of UBC12 and OsSnRK1 in the regulation of metabolic pathways. We have added the descriptions to the Discussion section of the revised manuscript in Page 17, line 547-line 551.

> Well, the authors have not addressed this point, they have merely re-stated my comment in the paper, with no further analysis shown. It is still unclear how degradation of SnRK1 by UBC12 alters sensitivity to low temperature. See also comment about RNAseq above.

Response: Thanks for your professional guidance and careful suggestion. We have re-conducted the RNA-seq of both WT and *osubc12* mutant seeds germinated at low temperature (15 ° C), and further analyzed the related metabolic pathways. Go enrichment analysis of DEGs in *osubc12* mutants vs. WT revealed that OsUBC12 also may be involved in some metabolic pathways, including carbohydrate metabolic process, sucrose metabolic process, starch metabolic process, and lipid metabolism process ect (as shown in Response Fig 2b, which is also namely Fig. S14 of the revised manuscript), suggesting the connection between OsUBC12 and OsSnRK1.1 from the other side, and OsSnRK1.1 may be the important target protein of OsUBC12. We have explained the corresponding results as follows:

“Moreover, in addition to regulating ABA signaling, SnRK1 also functions as an energy regulator integrating carbohydrate, starch, and lipid metabolism ect⁸³⁻⁸⁵, while Go enrichment analysis of DEGs in *osubc12* mutants vs. WT revealed that OsUBC12 also may be involved in some metabolic pathways, including carbohydrate metabolic process, sucrose metabolic process, starch metabolic process, and lipid metabolism process ect (Fig. S14b). This also suggests the connection between OsUBC12 and OsSnRK1.1 from the other side, and OsSnRK1.1 may be the important target protein of OsUBC12.” (as shown in Page 16, line 513-line 523 of the revised manuscript).

Furthermore, to further verify whether OsUBC12 could affect the OsSnRK1.1 protein level *in vivo*, we have also prepared anti-OsSnRK1.1 antibody by Abmart Shanghai Co.,Ltd, and performed western blotting to measure OsSnRK1.1 protein levels in WT, *osubc12* mutants, *OsUBC12-OE* and *OsUBC12^{C92A}-OE* transgenic lines. The results showed that OsSnRK1.1 protein levels were clearly increased in *osubc12* mutants and obviously decreased in *OsUBC12-OE* lines compared to WT, whereas the reduction of OsSnRK1.1 protein levels in *OsUBC12^{C92A}-OE* lines was not pronounced (as shown in Response Fig below, or Page 13, line 404-line 412 and Fig. 6e of the revised manuscript).

Namely Fig. 6e in the revised manuscript:

Fig. 6e OsSnRK1.1 protein levels of WT, *osubc12* mutants, *OsUBC12-OE* and *OsUBC12^{C92A}-OE* transgenic lines.

Total proteins extracted from the seeds of WT, *osubc12* mutants, *OsUBC12-OE* and *OsUBC12^{C92A}-OE* transgenic lines, was subjected to precipitates with anti-OsSnRK1.1 or anti-OsActin respectively by western blotting analysis. OsActin was used as the loading control. OsSnRK1.1 protein levels were visualized by western blotting using an anti-OsSnRK1.1 antibody. The anti-OsSnRK1.1 antibody was prepared by Abmart Shanghai Co.,Ltd.

Based on our current experimental results: at the post-translational level, low temperature could promote the accumulation of OsUBC12 protein (Fig. S22a, b). At the transcriptional level, *OsUBC12* was induced by both cold and ABA. (Fig. 3a). GO enrichment analysis of DEGs in *osubc12* mutants vs. WT showed the link between OsUBC12-promoted LTG and the suppression of ABA signaling (Fig. S14b). The expression levels of *OsABI5* and *OsRAB21* were higher in *osubc12* mutants than in WT (Fig. 3b, c). Knockdown of *OsUBC12* also increased ABA sensitivity (Fig. 3d-h). These findings indicate that OsUBC12-promoted LTG is associated with suppression of ABA signaling. The results obtained by overexpression analysis of transgenic rice lines showed an agreement with that of *osubc12* mutants, and blocking the 92th cysteine site of OsUBC12 could significantly suppress its regulatory effect on germination (Fig. 4b-k). Our results thus provide evidence that OsUBC12 accelerates LTG by repressing ABA signaling via its conserved ubiquitination function. Then, we demonstrated that OsUBC12 interacts with OsSnRK1.1 (Fig. 5), promoting its degradation (Fig. 6, S17). Further analysis revealed that OsSnRK1.1 inhibits LTG by enhancing ABA signaling (Fig. S19), and acts downstream of OsUBC12 (Fig. 7). Overall, OsUBC12 negatively regulate ABA signaling by promoting OsSnRK1.1 degradation, thereby accelerating LTG (for details, see **the Results or Discussion section of the revised manuscript**).

3) Fig 2c, 4b need statistical analysis.

Response: Thanks for your professional suggestion. We have supplemented statistical analysis to the Fig 2c and Fig 4b of our revised manuscript.

>Thank you

Response: Thanks for your contribution in reviewing our manuscript.

4) Interaction studies in *N. benthamiana* provide evidence that UBC12-SnRK1 interaction occurs “in planta”, not “in vivo”. To conclude that the interaction occurs in vivo, Co-IP needs to be performed in rice. Please revise the conclusion in the abstract and the rest of the manuscript (L371, 373; 494).

Response: Thanks for your professional comments. We have amended this conclusion, and revised the conclusion in the rest of the manuscript. (as shown in the revised manuscript in L379, L381, L485)

>Thank you

Response: Thanks for your contribution in reviewing our manuscript.

5) Fig 6 and text (L408, 413; L745; L1182): Degradation was done in protoplasts, not in vivo. Please correct.

Response: Thanks for your professional comments. We have amended this description, and corrected the description in the rest of the manuscript. (as shown in the revised manuscript in L399, L404, L460, L744, L1179)

>Thank you

Response: Thanks for your contribution in reviewing our manuscript.

6) Fig S18. Results from Phosphorylation assays are not clear. There is a band (boxed in light blue) in the positive control (ABI5) in the presence of SnRK1.1, and SnRK1+GRIK, but also in the absence of the kinases (and in the absence of ABI5?). Furthermore, how can ABI5 have autophosphorylation activity in vitro (L388)? In contrast, there are bands in UBC12 only in the presence of SnRK1, and SnRK1+GRIK, although weak. Why was it concluded that UBC12 was not phosphorylated? It may be better to blot with anti-His and anti-GST to detect phosphorylated and non-phosphorylated ABI5 and UBC12, respectively. See Han et al., 2020 (Plant Communications) Fig 3B for example of clear phosphorylation of a SnRK1 substrate. Alternatively, the authors may consider removing the phosphorylation part of this paper, since it doesn't add much to the story.

Response: Thank you for your careful suggestion. We have removed the phosphorylation part of this paper, and improved the model at the same time (as shown in Fig S23 of the revised manuscript).

>Thank you

Response: Thanks for your contribution in reviewing our manuscript.

7) L402-404 “Compared to WT extract, the degradation rate of OsSnRK1.1 was significantly decreased in osubc12 (Fig 6b), OsSnRK1.1 degradation was significantly inhibited by the proteasome inhibitor MG132” Fig 6c. To claim significant differences one needs to provide statistical analysis. Please remove “significantly” or provide statistical analysis.

Response: Thank you for your professional comments. According to your careful suggestion, We removed the “significantly” in these statements (as shown in the

revised manuscript in Page 13, L392-L395).

>Thank you

Response: Thanks for your contribution in reviewing our manuscript.

Germination assays in supplem material between Fig S9 and S10 don't have Figure number and legend

Response: Thank you for your careful comments. We have added the corresponding Figure number and legend between Fig S9 and S10 in Supplementary information.

>Thank you

Response: Thanks for your contribution in reviewing our manuscript.

We have done our best to improve the manuscript, and hope that the revision could address your concern and receive your approval and support.

>Aside from my comments above, I agree with reviewer#1 about condensing the discussion and highlighting the novelty of this work. Also, the new paragraph added in the discussion is not very clear (L497-505”).

Response: Thank you for your careful suggestion. We have refined the discussion, and perfected the paragraph on the novelty of this work (**as shown in Page 16, line 507-line 515 of the revised manuscript**).

Additionally, we have removed the discussion (L547-551: “in light of SnRK1 role as energy regulator during stress, it would be interesting to further analyze the energy metabolism pathways influenced by UBC12-OsSnRK1 to determine the connection, role, and mechanism of UBC12 and OsSnRK1 in the regulation of metabolic pathways.”), and added the description as follows:

“SnRK1s are serine/threonine protein kinases which requires activation of the T-loop by SnAK/GRIK kinases to increase its phosphorylation activity and phosphorylate targets^{89,90}, it would be interesting to further study the cross-talk between OsSnRK1-mediated phosphorylation and OsUBC12-mediated ubiquitination.” (**as shown in the Discussion section of the revised manuscript in Page 18, line 571-line 575**).

We hope that the revision could address your concern. Thanks again for your contribution in reviewing our manuscript, all of which are very helpful to the improvement of our manuscript.

Reviewers' Comments:

Reviewer #1:

Remarks to the Author:

Reviewer #4 did a good job at helping the authors to further clarify their research results and pointed out some missing data. There were three comments left for the authors to address.

1) The reviewer has (correctly) questioned the temperature conditions at which the original transcriptomics analyses were done. The authors have responded by performing a transcriptomics experiment at this more relevant temperature, which has further strengthened the original conclusions.

2) The reviewer also insisted on a more wider consideration of the various roles of UBC12 in physiological processes other than ABA regulation, which is indeed of interest. The authors have responded to this by adding additional comments to the manuscript. However, they have maintained their focus on the ABA part of UBC12 control, and I believe this is justified, also based on the new data sets that have been added. In light of the known role of ABA in stress-controlled germination, the authors appear to be correct in zooming in on this facet of UBC12-dependent regulation. Moreover, the authors have added in vivo data showing that SNRK1.1 accumulation is indeed altered in *ubc12* mutants and UBC12 overexpression lines, and these changes correspond well with the other data presented in their manuscript. Considering the important role of SNRK1.1 in ABA regulation, and the essential role of ABA in stress-regulated seed germination, it appears that the authors are correct in focusing on this part of the UBC12-controlled physiological processes. Indeed, the ABA hypersensitivity observed in *ubc12* mutants corresponds well with a role of UBC12 in ABA-regulated seed germination.

3) Finally, the reviewer has suggested that the authors modify the manuscript to clarify further the novelty of their findings in the discussion section. The additions made have addressed this suggestion, and although some of the phrasing is up for improvement, this can be readily addressed during the potential text editing phase of the manuscript.

In conclusion, the authors have addressed the reviewer comments well and provided new data that further strengthens the main conclusions of the manuscript. This work makes an important contribution to our understanding of a key rice domestication event.

Reviewer #3:

Remarks to the Author:

I have no more comments on the revision.

Response to reviewer's comments:

Reviewer #1 (Remarks to the Author):

Reviewer #4 did a good job at helping the authors to further clarify their research results and pointed out some missing data. There were three comments left for the authors to address.

1) The reviewer has (correctly) questioned the temperature conditions at which the original transcriptomics analyses were done. The authors have responded by performing a transcriptomics experiment at this more relevant temperature, which has further strengthened the original conclusions.

Response: Thank you for your support and recognition of our manuscript. We really appreciate the contribution of you and Reviewer #4 in reviewing the manuscript.

2) The reviewer also insisted on a more wider consideration of the various roles of UBC12 in physiological processes other than ABA regulation, which is indeed of interest. The authors have responded to this by adding additional comments to the manuscript. However, they have maintained their focus on the ABA part of UBC12 control, and I believe this is justified, also based on the new data sets that have been added. In light of the known role of ABA in stress-controlled germination, the authors appear to be correct in zooming in on this facet of UBC12-dependent regulation. Moreover, the authors have added in vivo data showing that SNRK1.1 accumulation is indeed altered in *ubc12* mutants and UBC12 overexpression lines, and these changes correspond well with the other data presented in their manuscript. Considering the important role of SNRK1.1 in ABA regulation, and the essential role of ABA in stress-regulated seed germination, it appears that the authors are correct in focusing on this part of the UBC12-controlled physiological processes. Indeed, the ABA hypersensitivity observed in *ubc12* mutants corresponds well with a role of UBC12 in ABA-regulated seed germination.

Response: Thank you for your support and recognition of our manuscript. We really appreciate the contribution of you and Reviewer #4 in reviewing the manuscript.

3) Finally, the reviewer has suggested that the authors modify the manuscript to clarify further the novelty of their findings in the discussion section. The additions made have addressed this suggestion, and although some of the phrasing is up for improvement, this can be readily addressed during the potential text editing phase of the manuscript.

Response: Thank you for your support and recognition of our manuscript. We really appreciate the contribution of you and Reviewer #4 in reviewing the manuscript.

In conclusion, the authors have addressed the reviewer comments well and provided new data that further strengthens the main conclusions of the manuscript. This work makes an important contribution to our understanding of a key rice domestication event.

Response: Thanks again for your contribution in reviewing our manuscript.

Reviewer #3 (Remarks to the Author):

I have no more comments on the revision.

Response: Thank you for your support and recognition of our manuscript. We really appreciate your contribution in reviewing the manuscript.